Manuscript prepared for Earth Surf. Dynam.
with version 2015/04/24 7.83 Copernicus papers of the LATEX class copernicus.cls.
Date: 6 October 2016

# Reconstruction of North American drainage basins and river discharge since the Last Glacial Maximum

Andrew D. Wickert[1]

[1]Department of Earth Sciences and Saint Anthony Falls Laboratory, University of Minnesota, Minneapolis, MN, USA

*Correspondence to:* A. D. Wickert (awickert@umn.edu)

**Abstract.** Over the last glacial cycle, ice sheets and the resultant glacial isostatic adjustment (GIA) rearranged river systems. As these riverine threads that tied the ice sheets to the sea were stretched, severed, and restructured, they also shrank and swelled with the pulse of meltwater inputs and time-varying drainage basin areas, and sometimes delivered enough meltwater to the oceans in the right places to influence global climate. Here I present a general method to compute past river flow paths, drainage basin geometries, and river discharges, by combining models of past ice-sheets, glacial isostatic adjustment, and climate. The result is a time series of synthetic paleohydrographs and drainage basin maps from the Last Glacial Maximum to present for nine major drainage basins – the Mississippi, Rio Grande, Colorado, Columbia, Mackenzie, Hudson Bay, Saint Lawrence, Hudson, and Susquehanna/Chesapeake Bay. These are based on five published reconstructions of the North American ice sheets. I compare these maps with drainage reconstructions and discharge histories based on a review of observational evidence, including river deposits and terraces, isotopic records, mineral provenance markers, glacial moraine histories, and evidence of ice stream and tunnel valley flow directions. The sharp boundaries of the reconstructed past drainage basins complement the flexurally smoothed GIA signal that is more often used to validate ice-sheet reconstructions, and provide a complementary framework to reduce nonuniqueness in model reconstructions of the North American ice sheet complex.

## 1 Introduction

At the time of Last Glacial Maximum (LGM) ice extent, ca. 30 ka to 19.5 ka (Clark et al., 2009; Lambeck et al., 2014), North American drainage configurations and river discharge were dramatically different from those at present. Ice advance rerouted rivers (e.g., Ridge, 1997; Curry, 1998; Blumentritt et al., 2009), and ice streams fed temporarily-enlarged drainage basins (Dyke and Prest, 1987; Patterson, 1998; Margold et al., 2014, 2015). The growth of the ice-sheets changed atmospheric circulation (Kutzbach and Wright, 1985; COHMAP members, 1988; Bromwich et al., 2004; Kim et al., 2008; Ullman et al., 2014) by generating up to 3.5–4 km of high-albedo ice-surface topography (e.g., Peltier, 2004; Tarasov et al., 2012; Peltier et al., 2015; Ivanović et al., 2016a) that

influenced global climate and therefore patterns of drainage-basin-scale water balance and river discharge to the oceans.

Reconstructions of past ice-sheet thickness have proliferated (e.g., Tushingham and Peltier, 1991; Marshall and Clarke, 1999; Lambeck et al., 2002; Peltier, 2004; Tarasov and Peltier, 2006; Tarasov et al., 2012; Gregoire et al., 2012; Argus et al., 2014; Peltier et al., 2015), but are evaluated using limited (Tarasov and Peltier, 2006; Tarasov et al., 2012) to no (all others) geologic evidence for past drainage patterns. Most of these (all but Marshall and Clarke, 1999, and Gregoire et al., 2012) were tuned or calibrated to terminal moraine positions and records of glacial isostatic adjustment, with the latter being a response with a characteristic two-dimensional flexural half-wavelength of 500–850 km over the high-flexural-rigidity Canadian Shield provence that was covered by the ice sheet (Vening Meinesz, 1931; Kirby and Swain, 2009; Tesauro et al., 2012). G12 (Gregoire et al., 2012), which is discussed in detail here, was built to roughly match ice-sheet outlines and evolve more freely. Additional geological data can help to constrain past ice-sheet geometries (see Stokes et al., 2015, who review and motivate data–model integration efforts), but until now, surface water flow paths have not been used to critically evaluate the suite of proposed past ice-sheets. Flow-routing calculations (e.g., Metz et al., 2011; Schwanghart and Scherler, 2014) are deterministic and can be used to produce time-series of drainage pathways predicted by each reconstructed ice-sheet and its associated glacial isostatic adjustment (GIA) history, forming the necessary links between each reconstructed ice-sheet and measurable geomorphic and sedimentary records. Furthermore, the discrete boundaries of drainage basins are sensitive to ice geometry in a way that is distinct from the broad and more diffuse GIA response, and therefore can provide a partly-independent check on the accuracy of any given ice-sheet reconstruction (Wickert et al., 2013).

On a global scale, climate change during deglaciation may have been forced by meltwater delivery to the oceans, especially at the time of the Younger Dryas (e.g., Broecker et al., 1989). The global climate system is highly sensitive to the locations of meltwater inputs to the oceans (e.g., Tarasov and Peltier, 2005; Murton et al., 2010; Condron and Winsor, 2012; Carlson and Clark, 2012). Geologic data provide a timeline of meltwater delivery to one basin or another (e.g., Ridge, 1997; Licciardi et al., 1999; Clark et al., 2001; Flower et al., 2004; Knox, 2007; Rittenour et al., 2007; Carlson et al., 2009; Breckenridge and Johnson, 2009; Murton et al., 2010; Hoffman et al., 2012; Williams et al., 2012; Breckenridge, 2015), with some types of data, such as oxygen isotopes, being able to be analyzed to produce reconstructions of meltwater discharge (Moore et al., 2000; Carlson et al., 2007a; Carlson, 2009; Obbink et al., 2010; Wickert et al., 2013). The relationships between geological data in different parts of the continent must be connected by conservation of ice-sheet mass and glaciological continuity of ice flow. Continental-scale meltwater routing calculations that connect data and models address both of these concerns by using quantified ice-sheet thickness and ensuring that water that is not sent to one basin is sent to another.

In general, there exists a lack of recognition of the importance of Pleistocene drainage rearrangement on river systems. In spite of the broad knowledge that river systems and drainage basins have changed significantly since the LGM (Upham, 1883; Bell, 1889; Clayton and Moran, 1982; Dyke and Prest, 1987; Wright, 1987; Teller, 1990a, b; Marshall and Clarke, 1999; Patterson, 1997, 1998; Licciardi et al., 1999; Overeem et al., 2005; Tarasov and Peltier, 2006; Wickert, 2014; Ullman et al., 2014; Margold et al., 2014; Ullman et al., 2015; Margold et al., 2015), many studies do not include any clear picture of drainage basin evolution. This results in representations of the modern drainage network wholly or partially in place of glacial-stage drainage basins (e.g., Blum et al., 2000; Sionneau et al., 2008; Montero-Serrano et al., 2009; Sionneau et al., 2010; Kujau et al., 2010; Lewis and Teller, 2006; Tripsanas et al., 2007, 2014; Rittenour et al., 2003, 2005, 2007; Knox, 2007) that help to propagate a lack of consciousness about the continental-scale hydrologic changes that occurred, even in cases when the study acknowledges in the text that past drainage pathways were different. The best current Cenozoic history of Mississippi River drainage does not attempt to place its northern drainage basin boundary during the Pleistocene (Galloway et al., 2011). Model-based studies likewise either test a range of possible drainage basin areas (Overeem et al., 2005) or assume that the rivers have remained unchanged (Roberts et al., 2012). In the former case, this can lead to less-precise calculations of sediment transport and deposition. In the latter case, this violates the underlying assumptions used to compute erosion and infer spatially-distributed uplift. Paleoclimate general circulation model (GCM) reconstructions (e.g., Liu et al., 2009; He et al., 2013) maintain static drainage basins in spite of climate sensitivity to patterns of meltwater routing (Condron and Winsor, 2012), and this has just begun to be addressed (Ivanović et al., 2014, 2016a, b). All of these scenarios highlight the need for an improved community-wide understanding of the ice-age effects on the continental-scale river systems of North America that provides the necessary backdrop to understand the present-day rivers, their origins, and their evolution.

Here, I focus primarily on nine North American river basins. These are (counterclockwise from North) the Mackenzie River, Columbia River, Colorado River, Rio Grande, Mississippi River, Susquehanna River/Chesapeake Bay, Hudson River, Saint Lawrence River, and Hudson Bay/Hudson Strait. These catchments include those that were dramatically and directly impacted by Quaternary glaciation and those that were far from the ice sheets, and together constitute a reasonably representative set of rivers from which the continental-scale hydrologic impact of glaciation on North American drainage basins can be surmised.

The present work is inspired by the dual and connected needs for (1) an appropriate mapping tool for data–model intercomparison and (2) maps of the past drainage basins of North America to inform geologic studies. I first present a standardized and automated method to extract past drainage pathways and river discharges from combinations of ice-sheet, GIA (Mitrovica and Milne, 2003; Kendall et al., 2005), and climate (Liu et al., 2009; He, 2011) reconstructions. I then apply this method to five ice-sheet models: the old but still-relevant ICE-3G (Tushingham and Peltier, 1991);

the "industry standard" ICE-5G (Peltier, 2004); the ANU model, developed by a separate research group (Lambeck et al., 2002), the ice-physics-based "G12" model (Gregoire et al., 2012), and the new ICE-6G (Argus et al., 2014; Peltier et al., 2015). I then compare the resultant drainage basin evolution and synthetic paleohydrographs against onshore and offshore geologic data, including data-driven reconstructions of North American drainage basins.

## 2  Methods: Automated Drainage Reconstruction

River systems in North America have responded to synchronous and linked changes in climate, ice extent, and sea level, with the latter including GIA-induced variability. Addressing this connected climate–ice–solid-Earth system is necessary to accurately reconstruct drainage. Ice advances and retreats blocked drainage pathways and rerouted rivers (Tight, 1903; Ver Steeg, 1946; Bluemle, 1972; Teller, 1973; Licciardi et al., 1999; Teller et al., 2002; Stanford, 2010; Carlson and Clark, 2012; Prince and Spotila, 2013). Ice-sheet retreat left behind large isostatic depressions that held proglacial lakes, such as the massive Glacial Lake Agassiz (e.g., Breckenridge, 2015), and strongly affected drainage patterns (e.g, Fisher and Souch, 1998). Even the subtle (∼20–60 m) forebulges of continental ice-sheets may have played major roles in drainage rearrangement (Anderson, 1988; Stanford et al., 2016). This is especially true of the generally low-relief North American cratonal interior, where ice-sheets and GIA have been significant drivers of large-scale topographic slopes.

The drainage basin and paleohydrologic reconstructions are a combination of four inputs: (1) a digital elevation model (DEM) with high enough resolution to resolve the valleys of major river systems, (2) a prescribed ice-sheet history, (3) a model of the GIA response to that ice history, and (4) a past water balance based on GCM outputs that have been offset to fit modern data at the present time step (Fig. 1). These factors together allow us to generate synthetic river discharge histories that can be compared with geologic data.

This analysis produced three major end products. The first is a gridded, spatially-distributed map of estimated river discharges to the sea since the LGM, expressed as centennial to millennial means depending on the resolution of the input ice-sheet model. The second is a coarsened version of the first, designed for coupling with GCMs to investigate the feedbacks between ice melt, ocean circulation, and climate (see Ivanović et al., 2014, 2016b). The third is a set of major drainage basin extents and a time-series of their discharges into the ocean. Here, I focus on this third product, as considering these specific drainage basins simplifies an effect spread across tens of thousands of pixels into nine discrete entities for which there exists a sufficient geologic record to compare model outputs and data.

The drainage analyses were performed using GRASS GIS (Neteler et al., 2012; GRASS Development Team, 2015), a free and open-source GIS software that has efficient data management, computational, and hydrologic tools. Any series of operations within GRASS GIS may be scripted,

and the Python source code to automate these analyses has been released under the GNU General Public License (GPL) version 3 and made available to the public (Section 7).

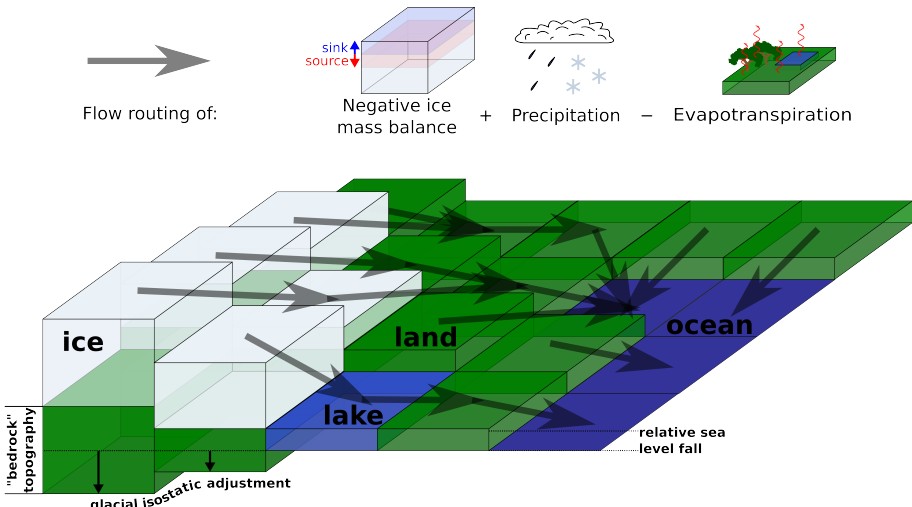

**Figure 1.** Meltwater and meteoric (precipitation minus evapotranspiration) water are routed to the coast, across a surface formed by the sum of present-day topography, glacial isostatic adjustment, and ice sheet thickness.

### 2.1 Surface elevations for flow-routing

Water and ice flow downslope following a routing surface, $Z_r$, that is given by:

$$Z_r = Z_{m,b} + H_i + \Delta Z_b \tag{1}$$

$Z_{m,b}$ is the reference (i.e., modern) land-surface elevation (subscript "$m,b$" stands for "modern glacier bed" or "modern bare (i.e., ice-free) ground"), $H_i$ is the ice thickness, and $\Delta Z_b$ represents changes between the past and modern land-surface (glacier-bed) elevation. Together, $Z_{m,b} + \Delta Z_b$ give the past glacier bed elevation field. When this ($Z_{m,b} + \Delta Z_b$) surface is summed with ice thickness, $H_i$, the resulting routing surface, $Z_r$, combines ice-free land-surface elevations and the el-
evations of the surface of the ice sheet. The ice-sheet surface is chosen to drive the flow-routing calculations because this is what generates the driving stress for ice flow (cf. Cuffey and Paterson, 2010, section 8.2.1), and therefore sets ice divides and meltwater pathways. If one wanted to compute subglacial water routing instead, an alternate approach would be to drive flow by the subglacial pressure gradient to simulate meltwater routing through subglacial streams and tunnel channels (e.g.,
Wright, 1973), and this could be accomplished by multiplying $H_i$ by $\rho_i/\rho_w$, where the latter are the densities of ice and water, respectively (e.g., Livingstone et al., 2015).

### 2.1.1 DEM and grid resolution

$Z_b$ is modern land-surface elevation. The primary modern elevation data set for flow routing is the GEBCO_08 30-arcsecond DEM (British Oceanographic Data Centre, 2010). GEBCO provides the
highest resolution combined topography and bathymetry data sets currently available. 30 arcseconds corresponds to a north–south cell size of 927 m, and a east–west cell size that ranges from 895 m at 15°N, the southern extent of the analysis region, to 81 m at 85°N, the northern extent. The GEBCO_08 grid is used in all areas except beneath the Greenland Ice Sheet, where the subglacial topography is interpolated from the etopo1 1-arcminute global topographic data set (Amante and
Eakins, 2009).

This sub-kilometer resolution over the ice-free Earth is required to resolve flow inside the km-scale valleys of the major rivers that drain North America, and is therefore critical to accurately calculate river networks. More coarsely-gridded drainage routing can send water from the modern Upper Missouri River basin, for example, towards the Arctic Ocean. However, even these high-
resolution calculations exhibit some small deviations from mapped drainage basin extents. Examples of this, such as in the Arrowhead region of Minnesota where the Saint Louis River is believed to have progressively captured Upper Mississippi drainage near Savanna Portage (van Hise and Leith, 1911), indicate that still-higher resolution elevation data will continue to improve flow-routing calculations near headwaters streams, a scale-dependence given by the relationship between channel geometry,
river discharge, and drainage area (Leopold and Maddock, 1953).

Raster grid cell area, $A_c$, is a function of latitude, and this is an important conversion factor for changing scalar quantities in cells, such as ice thickness, into a measure of volume. Using the simplifying assumption that Earth is perfectly spherical,

$$A_c = \left( R_\oplus \frac{\pi}{180°} \delta\theta \right) \left( R_\oplus \sin\left(\theta\right) \frac{\pi}{180°} \delta\varphi \right) \tag{2}$$

Here, $R_\oplus$ is the mean radius of Earth, $\theta$ is colatitude (in degrees), and $\varphi$ is east-longitude (i.e., degrees east of Greenwich). $\delta\theta$ is cell north–south extent and $\delta\varphi$ is cell east-west extent; in the simulations presented here, both of these extents equal 30 arcseconds. These areas are computed at a latitude, $\lambda$, corresponding to the north–south midpoint of the cell, with the implicit assumption that the cells are small enough that the north–south difference in cell size can be approximated to be
linear. For the GEBCO_08 30-arcsecond grid that is used here, this approximation generates errors of <0.3%.

### 2.1.2 Ice-sheet history

$H_i$, ice-sheet thickness, is provided by pre-generated ice-sheet reconstructions. These are produced at a coarser resolution than the 30-arcsecond flow-routing grid, and therefore are interpolated using
an iterative nearest-neighbor approach to remove stepwise discontinuities. This smoothing works successively from 1/3 the native resolution of the ice-sheet reconstruction to 30 arcseconds, and

is necessary because some coarse ice-sheet reconstructions contain multi-cell interior plateaus and several-hundred-meter cliffs at their termini that would otherwise introduce artefacts in the flow-routing calculations. For a 1-degree grid, this requires 5 iterations and leads to smoothing over a half-degree distance (∼40–50 km) outside of a given cell. While this somewhat smears the ice-sheet boundary (see, e.g., "ICE-6G" vs. "From data" in Figures 7 and 8), it does not significantly change its fit to the published ice-margin chronology (Dyke et al., 2003; Dyke, 2004).

I calculate deglacial drainage using five ice models, four of which were created based on geo-physical data, and one of which was generated using a numerical model of ice-sheet physics. In the former, ice-sheet thickness evolution was determined by inverting for global sea level data by combining (1) the total water mass contained within the ice sheets, causing global mean sea level fall, and (2) calculations of the GIA process, which produces deviations from mean sea-level history. Most notable among these deviations is the local deformation of the lithosphere and mantle due to the weight of the ice, which leads to postglacial rebound following ice retreat. Ongoing postglacial rebound is a key indicator of the presence of major past ice-sheets, but postglacial rebound rates are a function of both time-evolving past ice-sheet thickness and solid Earth (lithosphere and mantle) rhe-ology. That this one rate is a function of two only partly-constrained variables introduces an element of uncertainty into GIA-based inversions (Mitrovica, 1996) such as the ICE-$n$G and ANU models. Ice physics, on the other hand, are most sensitive to the paleoclimate forcing and basal conditions (e.g., topography, roughness, sedimentary cover, geothermal heat flux) (Tarasov and Peltier, 2006; Cuffey and Paterson, 2010). The four geophysically-based ice-sheet reconstructions investigated here, listed with their associated solid Earth rheology models, are ICE-3G/VM1 (Tushingham and Peltier, 1991), ANU (Lambeck et al., 2002), ICE-5G/VM2 (Peltier, 2004), and ICE-6G_C/VM5a (Argus et al., 2014; Peltier et al., 2015). The one ice-physics-based reconstruction considered is that of (Gregoire et al., 2012), which was allowed to evolve more freely than the tuned ice-physics-based model reconstructions of Tarasov and Peltier (2006) and Tarasov et al. (2012).

ICE-3G (Tushingham and Peltier, 1991) was one of the earliest widely-used ice-sheet reconstruc-tions. It was constructed to constrain ice thickness as much as possible with measurements of glacial isostatic adjustment from radiocarbon-dated sea-level markers. While it does not include enough ice to reproduce the full magnitude of the observed LGM global sea level fall (Austermann et al., 2013; Lambeck et al., 2014), it provides a counterpoint to the more recent ice-sheet reconstructions, all of which include a much more massive North American ice-sheet complex (Figure 2).

ICE-5G (Peltier, 2004) has been often considered the "industry standard" global ice-sheet and GIA reconstruction. It was also built by combining geological data and geophysical inversions, and is an update to ICE-3G. It contains a very thick north–south-oriented ice dome over western Canada (Argus and Peltier, 2010) that is not located where glacial geological evidence would place the Kee-watin dome. This causes its meltwater discharge to the coast to fail to match isotopic records from the Mississippi River drainage basin (Wickert et al., 2013), and in any case, is glaciologically implau-

sible as the ice-sheet would slide and thin over the soft Western Interior Seaway sediments east of the Canadian Rockies. Nevertheless, ICE-5G remains widely-used and matches many observations of GIA and sea level (Peltier, 2004).

ICE-6G (here, shorthand for ICE-6G_C) (Argus et al., 2014; Peltier et al., 2015) is the successor to ICE-5G. It has been shown to improve agreement with present-day GIA measurements, and was built to match geologic evidence for the locations of the ice domes and ice-sheet outlines (Patterson, 1998; Dyke et al., 2003; Dyke, 2004). As ICE-6G is so new, it has not been extensively tested outside of the group that developed it (Argus and Peltier, 2010; Argus et al., 2014; Peltier et al., 2015).

Lambeck et al. (2002) developed the Australian National University (ANU) ice-sheet model, also based on field data constraining sea level and GIA. This model differs from ICE-3G and ICE-5G in that it was developed regionally, and these regions were later combined into a more global picture of ice-sheet evolution.

To produce the G12 ice model, Gregoire et al. (2012) simulated North American ice-sheet evolution from the LGM to present using the Glimmer Community Ice-Sheet Model (Glimmer–CISM) (Rutt et al., 2009). Glimmer–CISM is a thermomechanical model of ice deformation and dynamics that uses the shallow ice approximation for its driving stresses. While the shallow ice approximation neglects higher-order (i.e., less important) stresses, it generally is an acceptable approximation to solutions that do include higher-order terms, and its computational efficiency makes it a viable option for long-term simulations (Rutt et al., 2009), such as the Gregoire et al. (2012) LGM–present study. The G12 reconstruction features the collapse of the ice saddle between the Laurentide and Cordilleran Ice Sheets around 11.6 ka. Geological data, on the other hand, place the major phase of saddle collapse during the Bølling-Allerød warm period, ca. 14.5–12.9 ka (Dyke et al., 2003; Dyke, 2004; Williams et al., 2012), and possibly beginning as early as ∼17.0 ka (Catto et al., 1996; Clague and James, 2002; Carlson and Clark, 2012). While Gregoire et al. (2012) shifted their chronology such that peak saddle collapse occurred ∼14.5 ka, during Meltwater Pulse 1A (MWP-1A) (Fairbanks, 1989; Deschamps et al., 2012), I use their model ages as direct geologic age, such that the results presented here are linked in time with the effects of their climate forcings on their ice sheet. Most importantly, the ice dynamics contained within Glimmer–CISM provide a set of tools to match reality that place G12 in a different category than the GIA-based models.

### 2.1.3 Change in land-surface elevation: glacial isostatic adjustment

$\Delta Z_b$, change in land-surface (glacier bed) elevation, is here provided by the combination of factors that change local relative sea level. At the LGM, ice-volume equivalent global mean sea level was 130–135 m lower than present, but local relative sea level was highly variable due to the effects of GIA (Austermann et al., 2013; Lambeck et al., 2014). GIA comprises deformational, gravitational, and rotational effects of ice-age loading. The deformational component is the local flexural isostatic response to changes in ice and water loading. The gravitational component is in response to redistri-

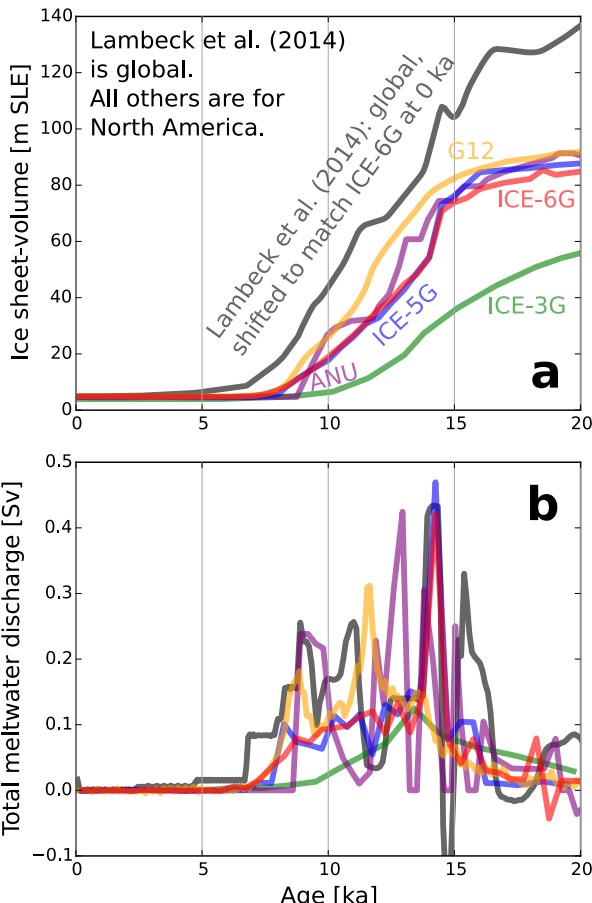

**Figure 2.** Ice volume and meltwater discharge histories over the North American computational region for each of the examined ice-sheet reconstructions (colors), along with changes in global ice volume from Lambeck et al. (2014) (gray). **a.** Cumulative change in ice volume over time. Volumes for the five ice-sheet reconstructions evaluated here are summed over the computational domain, which contains the entire North American ice-sheet complex and part of Greenland. The Lambeck et al. (2014) ice-volume curve is global, but has been shifted to match ICE-6G at the 0 ka (i.e., modern) time step for easier comparison with the deglacial North American curves. These ice volumes are presented in meters sea-level equivalent (SLE), assuming a constant ocean surface area of $3.62 \times 10^8$ km$^2$ (after Marshall and Clarke, 1999). **b.** Total meltwater discharge over the computational domain for each ice-sheet reconstruction. ICE-5G and ICE-6G have a noticeable peak during MWP-1A, 14.65–14.31 ka (Deschamps et al., 2012). Though the time-scale of their numerical model is shifted, Gregoire et al. (2012) also consider their meltwater peak, due to collapse of the ice saddle between the Cordilleran and Laurentide ice sheets, to represent MWP-1A. The global-scale Lambeck et al. (2014) sea-level curve includes a meltwater peak ∼16.5–15 ka, which encompasses Heinrich Event 1B (15.9–15.7 ka) (Gil et al., 2015). and significant ice-sheet melt 12.4–6.7 ka, during the first half of the Holocene. The vertical axis is in Sverdrups, where 1 Sv is $10^6$ m$^3$ s$^{-1}$.

bution of mass (and hence, gravitational potential) across the surface of Earth, and this is amplified by the water self-gravitation feedback (e.g., Mitrovica and Milne, 2003). The rotational component results from perturbations to Earth's moment of inertia due to global mass redistribution, and produces true polar wander that causes the equatorial bulge to migrate, first in the form of seawater and later in the form of solid Earth material. These three components feed back into one another, and

therefore are solved numerically through iteration.

The gravitationally self-consistent sea level theory applied here and its numerical implementation are described by Mitrovica and Milne (2003) and Kendall et al. (2005), respectively, who calculate the GIA effects of changing ice and water masses. The calculations are performed using a pseudo-spectral algorithm that, for this work, is truncated at spherical harmonic degree and order 256. This

satisfies the Nyquist flexural wavenumber for the solid Earth models employed, meaning that the wavelength of the spectral representation is always less than half of the flexural wavelength (Vening Meinesz, 1931) of the lithosphere in the solid Earth model. Satisfaction of this condition allows the solutions to be interpolated to an arbitrarily high resolution without concerns about aliasing.

ICE-3G, ICE-5G, ICE-6G, and ANU were each calibrated to match sea level data based on spe-

cific 1D models of solid Earth viscoelasticity (Tushingham and Peltier, 1991; Lambeck et al., 2002; Peltier, 2004; Argus et al., 2014; Peltier et al., 2015); Argus and Peltier (2010, p. 699) provide a succinct review of the VM$n$ models. I use each ice reconstruction's respective spherically symmetric Earth model to simulate its GIA response to these loads – VM1 for ICE-3G, VM2 for ICE-5G, VM5a for ICE-6G, and an unnamed model for ANU. G12 was built using a simplified model of isostatic

response to glacial loading, and was run only for North America. I pair it with VM2, and *a posteriori* enforce a total global ice-sheet volume (and therefore ice-volume equivalent global mean sea level history) equal to that of ICE-5G. Among these solid Earth models, VM1 is a two-layer model with a viscosity that increases at the transition zone. VM2 includes many layers, with a large increase in viscosity below the 660-km discontinuity and gradually-increasing viscosity with depth thereafter in

the lower mantle; these layers underly a 90 km thick elastic lithosphere in the most recent version of VM2, which is used here (Peltier, 2004). Below 100 km, VM5a is a simplification of VM2; above 100 km, it includes a 60-km thick lithosphere over a 40-km thick layer with a viscosity of $10^{22}$ Pa s. The ANU solid Earth model has a higher viscosity contrast between the upper and lower mantle than VM2, placing it in better agreement with geophysical inferences of the upper mantle–lower

mantle viscosity contrast based on observations of the long-wavelength geoid (Hager and Richards, 1989), and employs a 65 km thick elastic lithosphere. One may note that all of the lithospheric elastic thicknesses are characteristic of cratonal regions; this is because the Laurentide and Fennoscandian ice sheets nucleated and grew across old, thick cratonal lithosphere, and a cratonal value for elastic thickness therefore provides the best fit to near-field data that record GIA, whose local deformational

component is high-amplitude relative to the rotational and gravitational components.

I initiate the GIA modeling of G12 in equilibrium at the LGM, and ANU slightly before the LGM. This is a potential source of error: while the sea-level records of Lambeck et al. (2014) indicate that global ice mass grew only slightly between 30 and 20 ka, providing enough time to approach isostatic equilibrium, this is only $\sim 2$ $e$-folding time-scales towards isostatic equilibrium based on data from the Ångerman River and Hudson Bay (Mitrovica and Forte, 2004; Nordman et al., 2015). Therefore, it is very likely that the North American ice sheet complex was not yet in isostatic equilibrium at the LGM. ICE-5G includes less well-constrained ice-sheet growth since 120 ka. The starting point for ICE-6G is during the last interglacial (MIS 5e), when global mean sea level peaked 5.5–7.5 meters above its present level (Hay et al., 2014) though with some significant regional variability, as indicated by ancient beaches and modeling studies (Hearty et al., 2007; Rohling et al., 2007; Dutton and Lambeck, 2012; Hay et al., 2014; Dutton et al., 2015). ICE-3G was initiated at the last interglacial as well, with ice being grown in the same pattern as it retreated, except more slowly and in reverse.

Our flow-routing calculations neglect geomorphic change for three major reasons. First, rerouting of Lake Agassiz outflow occurred as a precursor to spillway incision (Teller et al., 2002), meaning that ice-sheet geometry and GIA are the primary drivers of drainage reorganization. Second, while process-based landscape evolution models exist (cf. Tucker and Whipple, 2002; Willgoose, 2005; Tucker and Hancock, 2010), the thresholded and highly nonlinear relationships between flow, erosion, sediment transport, and deposition, means that these models may predict a very wide range of potential landscape responses with sufficient uncertainty as to reduce their predictive capability. This would leave changing the DEMs by hand as an option (as was done by Tarasov and Peltier, 2005), but considering incomplete knowledge of past topography and my desire to, at least at this stage, separate the calculations from the field work against which they are checked, I choose to use modern topography while acknowledging that this is an approximation.

## 2.2 Water balance: precipitation, evapotranspiration, and ice melt

Each cell in the flow-routing grid domain can act as a source of water to continental drainage systems. Water balance over each cell is a sum of precipitation gains, $P$, evapotranspiration losses, $ET$, and changes in water-equivalent thickness of the reconstructed ice sheet, $(\rho_i/\rho_w)(\Delta H_i/\Delta t)$. When multiplied by the cell area, $A_c$ (Eq. 2), the result is an expression for volumetric incoming water discharge, $Q_{\text{in}}$ (Eq. 3) that is then routed across the continent.

$$Q_{\text{in}} = A_c \left( P - ET - \frac{\rho_i}{\rho_w} \frac{\Delta H_i}{\Delta t} \right) \tag{3}$$

### 2.2.1 Global precipitation and evapotranspiration

Precipitation and evapotranspiration inputs changed through time (Fig. 3), and precipitation minus evapotranspiration ($P - ET$) provides the amount of meteoric water that can be routed through river systems to the coast. Changes in $P - ET$ through time were calculated by Liu et al. (2009) and

He (2011) using a continuous LGM (22 ka) to present run of the National Center for Atmospheric Research (NCAR) Community Climate System Model version 3 (CCSM3) (Collins et al., 2006). This simulation, called TraCE-21K, was run at 3.5 degree resolution, and used ICE-5G/VM2 for its ice sheet and paleotopography input. Therefore, the only fully self-consistent calculations presented here are from the ICE-5G/VM2 reconstruction. I spherically interpolated (following Dierckx, 1993) the results to 0.25 degree (15 arcminute) resolution to match the assembled modern data products (below).

The low resolution CCSM3 contains systematic biases that produce a ∼1 mm/day positive precipitation anomaly in western North America and a ∼1 mm/day negative precipitation anomaly in eastern North America (Yeager et al., 2006). To correct for this and other possible model biases, I assembled modern precipitation and evapotranspiration fields, computed the difference between each time step of TraCE-21K and its modern time step, and summed the modern compilations with the computed TraCE-21K difference from modern. I could find no combined land–ocean evapotranspiration product, and combined land–ocean precipitation products were coarse, often at 2.5-degree resolution (∼280-km cells at the equator) (Xie and Arkin, 1997; Chen et al., 2002; Adler et al., 2003). Therefore, I generated my own global data product compilations as mean rates from 2000 to 2004, for consistency with prior work (Wickert et al., 2013), at 0.25° resolution. While this study centers on North America, this global climate data product can be used to extend these methods to other regions or the full globe (cf. Ivanović et al., 2014, 2016b). These compiled global grids of mean precipitation and evapotranspiration are publicly available (see Section 7).

I derived the precipitation data by stitching together three data products. The CRU TS3.21 0.5° precipitation product provided precipitation over the global continents except Antarctica (expanded from the work of Harris et al., 2014), and this was smoothed and spherically interpolated (following Dierckx, 1993) to 0.25° resolution. The HOAPS-3 satellite-derived precipitation data product (0.25° resolution) was used for the oceans north of the Antarctic region (Andersson et al., 2010). Gaps of 2° or less between these data sets were filled by an interpolation of them. The remainder of the globe, primarily the Antarctic region, was filled by the CMAP reanalysis data set (Xie and Arkin, 1997), interpolated from 2.5° resolution to 0.25° resolution.

Our modern evapotranspiration rates for the continents (except Greenland and Antarctica) are based on calculations using the MODerate-resolution Imaging Spectroradiometer (MODIS) instruments that are mounted on the Aqua and Terra satellites (Mu et al., 2007, 2011), available at 30-arcsecond resolution. The oceanic evaporation data are from HOAPS-3 and at 0.25° resolution (Andersson et al., 2010). Analysis of the HOAPS-3 data product shows that it has an ∼10% error, with a global 0.2–0.5 mm day$^{-1}$ excess evaporation (Andersson et al., 2011). These data sets were interpolated over the ice-free land surface and global oceans. Evapotranspiration rates for Greenland and Antarctica were taken from the 0-ka time step of the TraCE-21K simulation (Liu et al., 2009; He,

2011), spherically interpolated to 0.25° resolution. Any negative evapotranspiration rates generated by the interpolation were set to 0.

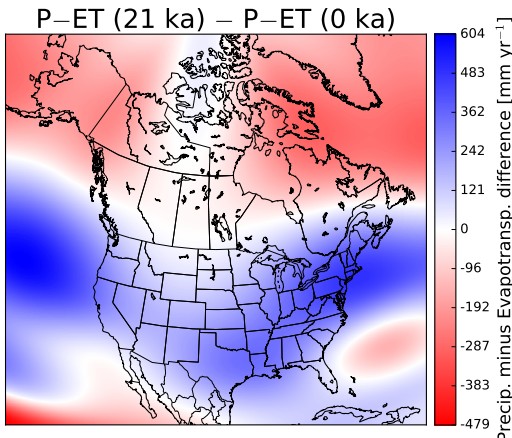

**Figure 3.** Modeled changes precipitation minus evapotranspiration between the LGM and present. These differences were calculated with the TraCE-21K continuous LGM to present paleoclimate run (Liu et al., 2009; He, 2011) of the CCSM3 GCM (Collins et al., 2006). They are combined with ice mass balance to produce water inputs for computed paleodischarges.

### 2.2.2  Ice melt and central differencing

Ice-sheet inputs to runoff are determined by differencing ice-sheet thicknesses at adjacent time steps for each of the ice models. These differences in ice-sheet thickness are converted to water discharges by (1) multiplying them by the DEM cell size (Eq. 2), (2) multiplying them by 0.917 as a conversion factor between ice and water density, and (3) dividing them by the time elapsed (Eq. 3).

Flow routing and drainage basins can be calculated only on time steps when the ice-sheet and GIA
models provide surface topography (Section 2.3, below), but the ice-sheet contribution to runoff requires ice-sheet thickness to be differenced. This requires a choice of whether to allow central differencing and numerical diffusion in space (i.e., flow-routing) or time (i.e., melt rate). Melt rates change more gradually than near-instantaneous drainage rerouting events, and these rerouting events are critical to understanding climate impacts of ice melt (e.g., Broecker et al., 1989; Clark et al.,
2001; Tarasov and Peltier, 2006; Condron and Winsor, 2012). Therefore, I prefer numerical diffusion in time, and allow the gridded melt rate at each flow-routing time step to equal the time-weighted averages of the melt rates calculated before and after it.

Time steps vary by model. ICE-3G (Tushingham and Peltier, 1991) and ANU (Lambeck et al., 2002) use irregular time steps separated by ∼1000 years (ICE-3G) and ∼500 years (ANU). Time
steps for ICE-5G are 1000 years from 32 to 17 ka, and 500 years thereafter (Peltier, 2004). All time steps for ICE-6G (Argus et al., 2014; Peltier et al., 2015) are 500 years. Model outputs from Gregoire

et al. (2012) were provided at 10-year intervals, with GIA computed at uniform 100-year time steps (Wickert et al., 2013), which are the same as those used here for the flow routing calculations. This means that the modeled discharge histories are averaged over 100- to 1000-year time steps and blur short-term geologic events that are responsible for creating major deglacial spillways and deposits (e.g., Kehew and Teller, 1994; Breckenridge, 2007; Murton et al., 2010).

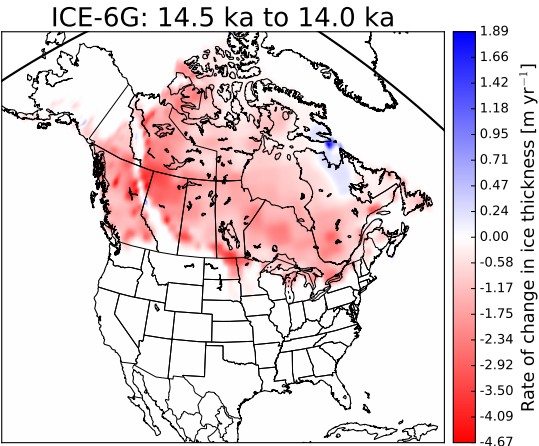

**Figure 4.** Ice-sheet mass balance is a major, and often dominant, input to calculations of North American river discharge during deglaciation (Eq. 3).

### 2.2.3 Neglected terms: groundwater and lakes

While groundwater is a significant reservoir that affects river discharge, I assume that over the millennial time scales of interest to us, groundwater storage changes slowly enough that its effect on river discharge is much less important than precipitation, evapotranspiration, and ice-sheet melt. Although ice sheets can strongly influence groundwater discharge, modeling work by Lemieux et al. (2008) shows that the maximum net deglacial groundwater discharge across the whole North American continent is only ∼2.5% the modern Mississippi River discharge, and that this spike was short-lived.

I do not include proglacial lakes in the continental water storage because the ice-sheet reconstructions discussed here (Tushingham and Peltier, 1991; Lambeck et al., 2002; Peltier, 2004; Gregoire et al., 2012; Peltier et al., 2015) do not. This means that these models implicitly include the lake-water volume within the ice mass. This deficiency affects the ability of the models to match the observed record of ice-marginal geometry, as ice must be used in the models to represent the surface loads of these large lakes in the geologic past, and removes the potential effects of the lakes in driving ice-sheet retreat in mechanistic models. In the present work, water is simply routed across proglacial-lake-producing depressions using a least-cost path algorithm (see Section 2.3).

### 2.3 Continental flow routing and accumulation

After computing the amount of flow each cell contributes to runoff, I used the time-evolving grids of surface elevation to compute drainage patterns across North America. At each time step, I first defined the ocean margin that sets the downstream boundary for terrestrial water flow. I then routed flow down the $Z_r$ (Eq. 1) surface while computing accumulated discharge as (1) meteoric inputs only, (2) ice-sheet inputs only, and (3) combined total discharge. I computed this flow routing and accumulation using "r.watershed", the high-efficiency least-cost path search algorithm of Metz et al. (2011), which is integrated into GRASS GIS (Metz et al., 2011; Neteler et al., 2012; GRASS Development Team, 2015).

#### 2.3.1 Defining continent, ocean, and shore

Shorelines and topographic elevations changed dramatically since the LGM, thus requiring redefinition of the coastline on which river mouths appear at each time step. This could not be done by simply picking regions below sea-level, because this approach would cause flow to "disappear" into regions such as the below-sea-level Caribou sub-basin of Lake Superior and the isostatically-depressed moat around the retreating Laurentide Ice Sheet margin. Therefore, I developed an algorithm to define as "ocean" only those regions that were below sea-level and contiguous with the global ocean. First, I thresholded the topography-plus-ice flow-routing map, $Z_r$, at 0, to define a binary raster map of regions above and below sea level. Then, I converted this into a vector map. Using the topological analysis built into GRASS GIS, I looked only for areas that touched the map edge and were below sea level, and chose these to be ocean. The remaining area was defined as continent and used as the flow-routing surface.

#### 2.3.2 Flow routing and accumulation

The flow-routing calculations produce maps of (1) drainage direction, (2) meteoric water discharge, (3) meltwater discharge, and (4) total water discharge. These maps are spatially distributed, providing 30-arcsecond-resolution time-series of the centennial- to millennial-averaged surface water discharge that is produced by each ice-model–solid-Earth-model combination.

The flow-routing algorithm I used, r.watershed (Metz et al., 2011), employs a least-cost path search algorithm. This sets it apart from other flow-routing tools in that it is capable of routing water through depressions without explicitly flooding them (see, e.g., Schwanghart and Scherler, 2014). This algorithm allows for rapid calculations to be carried out over a landscape that contains moving ice dams and isostatic depressions.

The ability of the Metz et al. (2011) algorithm to route water across local depressions also prevents both natural and artificial (human-built or DEM artefact) depressions in the landscape from impeding the flow routing. This is an advantage because flow-routing calculations without this feature fail

unless they are performed on a hydrologically-corrected DEM (i.e., one in which the elevation is altered to produce the known drainage basins). Using a hydrologically-corrected DEM is a potential source of error in reconstructions of the past because its actual terrain has been artificially changed (e.g., Grimaldi et al., 2007) in a way that may produce unexpected results when combined with surface deformation and ice-sheet thickness.

In spite of these advantages, the Metz et al. (2011) method causes internally drained regions to be amalgamated to drainage basins that flow to the coast. In the drainage calculations performed for this work, the Great Basin is split between the Colorado River and Columbia River drainage basins. In part, this is realistic: outflow from Lake Bonneville entered Columbia River between 18.2±0.3 ka and 16.4±0.2 ka, with limited and/or intermittent outflow continuing to $14.9^{+0.3}_{-0.6}$ ka ($12.6\pm0.15^{14}$C ka) (synthesis of Godsey et al., 2011; McGee et al., 2012) (radiocarbon ages calibrated using Int-Cal13, Reimer et al., 2013, with 2-$\sigma$ error). In part, this does not matter: a hydrologically-closed basin must maintain a state in which precipitation equals evapotranspiration, and as such should not have a major effect on discharge calculations, outside of the aforementioned period of climate change during the end of the Pleistocene. But in part, this is the dissatisfying result of flow-routing calculations that are not coupled to a hydrologic model that can raise and lower the levels of lakes in closed basins, and a fundamental problem in any flow-routing approach that does not include lake and groundwater levels.

This dilemma over the Great Basin highlights a key question regarding how much to adjust the model by hand to fit data, versus how much to just allow the model to run. It would be possible to modify the flow-routing inputs to declare the Great Basin to be closed during part or all of the modeled geologic past. The approach taken here is to leave the flow-routing grid unmodified in order to maintain distance between data and model (see the final paragraph in Section 2.1.3).

## 2.4 Distributed melt delivery to the coast

These methods also allow us to produce grids of meltwater discharge to the coast that are fully-distributed in space and time. These have been converted to GCM-input-appropriate spatial resolutions and used to connect ice melt and global ocean circulation and climate change (Ivanović et al., 2014, 2016b). This connection to continental runoff distinguishes this distributed meltwater approach from "hosing" experiments (e.g., Meissner and Clark, 2006; Condron and Winsor, 2012), in which meltwater additions are prescribed to a patch of ocean to explore the climatic impacts of a major meltwater input event. Fully-distributed GCM meltwater inputs that are tied to paleodrainage histories, when coupled with models that can appropriately handle point sources of fresh water, create a more realistic ice-age ocean that is better-conditioned to respond to changes in meltwater inputs and can be used to connect models of ice-sheet history with their likely impacts on global climate. Code to produce these GCM inputs have been released alongside this publication (see Section 7).

## 2.5 Drainage basins and river discharge

I compute time-evolving drainage patterns with a generalized algorithm that can overcome changes in shoreline geometry, river mouth positions, and difficulties in producing consistent elevation-based flow reconstructions across the very subtle topography of the continental margins. These components are essential to connect glacial-stage rivers to their modern counterparts after 20,000 years of shoreline and topographic change due to evolving sea level and ice-sheet geometry.

First, I create a map of regions where I expect the mouth(s) of each river system to be. Each "river mouth region" (Fig. 5b) is defined by hand to accommodate the changing locations of river mouths with time as a function of sea level, ice cover, and GIA (Figure 6). This is especially important for drainage systems such as Hudson Strait and the Saint Lawrence River, which previously played host to ice streams and now are fully or partially inundated with seawater. The same set of river mouth regions is used for every ice sheet and GIA reconstruction at all time steps.

I then create a set of points at which each of the "major" rivers ($\geq$1000 $m^3$ $s^{-1}$ reconstructed discharge) reaches the coast. Wherever one of these river mouths is located inside a named region (Fig. 5), the program builds a drainage basin. Each drainage basin is labeled with the name of its respective river mouth region (e.g., "Mississippi" or "Columbia"). River mouth regions that contain more than one river mouth produce a named drainage basin that is an amalgamation of the drainage basins of all $\geq$1000 $m^3$ $s^{-1}$ rivers in the basin; this allows us, for example, to combine the Mississippi, Atchafalaya, and Red Rivers into a single broader basin flowing towards Louisiana, and to combine all of the major modern rivers that flow into Hudson Bay into a single basin, even though their previous combined outlet in Hudson Strait is now inundated. These river mouth regions are chosen by hand, and work only if one river mouth does not migrate into a region occupied by another river mouth at a different time. If such a dramatic change occurs, as it does for the Hudson River at 12.5 and 12.0 ka in ICE-6G and for Hudson Bay at 9.6 ka in G12 (these are the only known likely errors), it then becomes challenging to even define the river as being the same, and more reach-scale definitions may become required. If a particular river mouth region contains no river mouths with flow $\geq$1000 $m^3$ $s^{-1}$, the program builds a drainage basin upstream of the highest-discharge coastal cell from the flow accumulation calculation.

After defining the major drainage basins, I sum (1) ice-sheet melt, (2) precipitation minus evapotranspiration, and (3) the sum of (1) and (2) across each basin. Each sum, respectively, provides ice-sheet meltwater discharge ($Q_i$), meteoric (seasonal precipitation minus evapotranspiration) discharge ($Q_m$), and total river discharge ($Q = Q_m + Q_i$). Many of these calculations provide discharges at the modern time step that are close to the modern pre-human-impact (i.e., preindustrial) mean river discharge (Table 1, which also includes measured modern sediment discharges for reference). For some, however, the result differs somewhat due either to (1) the water balance maps not perfectly representing water delivery to the coast and/or (2) the incorporation of the internally-drained Great Basin into the Columbia River and Colorado River basins because the flow-routing

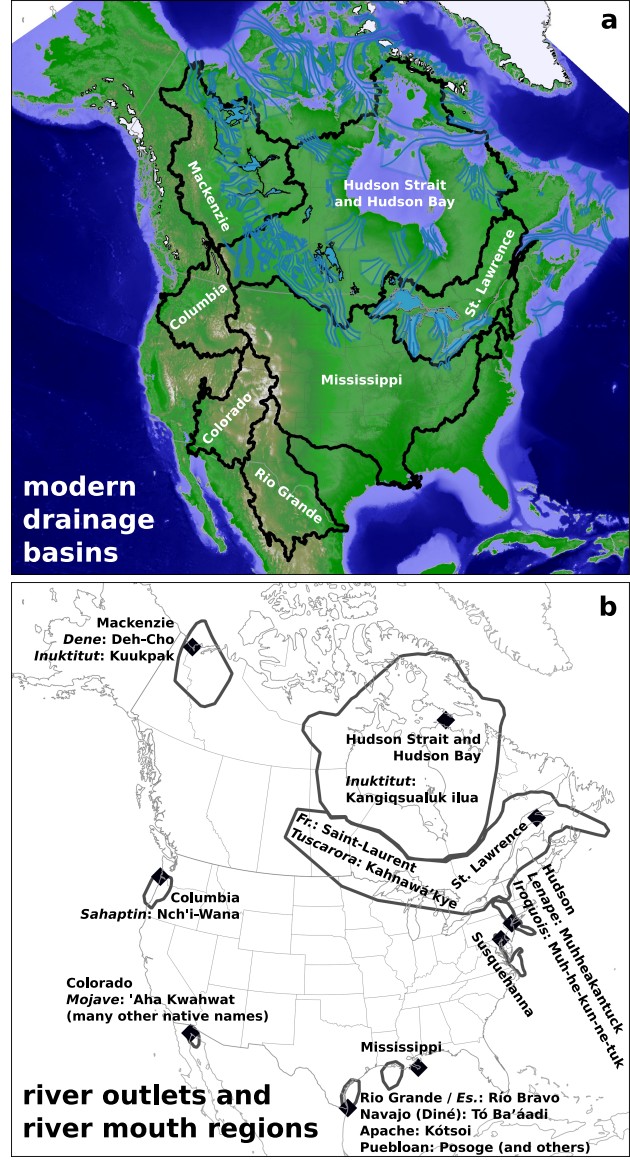

**Figure 5. a.** Modern drainage basin extents for the rivers involved in this study, with ice and lake extents from the 1 ka time step of the maps of Dyke (2004), elevation basemap from GEBCO_08 (British Oceanographic Data Centre, 2010), and modern drainage basins from Lehner et al. (2013). **b.** Modern river mouths alongside the river mouth regions used to define a flow path as part of a named drainage basin (Section 2.5). Names are also given in European non-English languages where they are common, alongside a non-exhaustive set of common names in local languages. Note that the mouth of the Mississippi River lies entirely outside its river mouth region. This is because the topographically-routed flow, computed using the Metz et al. (2011) r.watershed algorithm, follows the course of the Atchafalaya River; the present path of the Mississippi near its mouth does not follow the regional topographic gradient because it is held in its old course by the Old River Control Structure (cf. Harmar and Clifford, 2007). These same river mouth regions are used for all time steps in all models.

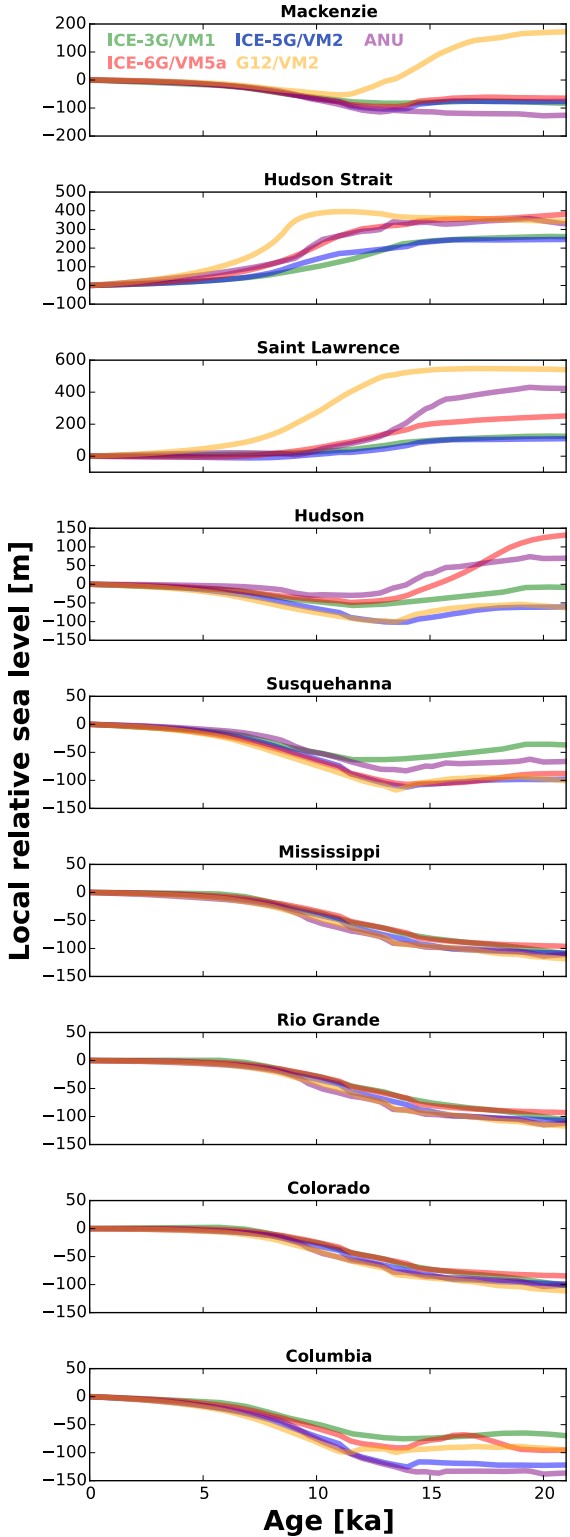

**Figure 6.** Modeled sea-level curves at river mouths (Fig. 5). Rivers mouths formerly near (or under) thick ice have more variable sea-level histories, illustrating the divergent GIA histories result from different ice-sheet and solid Earth models.

algorithm (Metz et al., 2011) was set up to disallow internal drainage. To provide more realistic discharge histories while also being explicit about the raw outputs of the model, I rescaled discharge in each basin to match its modern or preindustrial value with the mean of the computed time steps between 3 ka and present, while also making note on the plots (Figs. 11–16) of the predicted uncorrected modern or preindustrial discharge. This correction may create a systematic underestimate of Columbia River discharge from 18.2±0.3 ka (McGee et al., 2012) to $14.9^{+0.3}_{-0.6}$ ka ($12.6 \pm 0.15^{14}$C ka) (Godsey et al., 2011; Reimer et al., 2013), when waters from Lake Bonneville – part of the Great Basin – flowed into the Columbia River Basin.

## 3  Methods: Data-driven Drainage Reconstruction

A set of entirely data-driven drainage histories provides the necessary counterpoint to evaluate the model-based computed drainage basins and river discharge histories. I developed these in two different ways: as paleogeographic maps of drainage divide positions, and as a simplified river discharge history that defines periods of high and low flow. All radiocarbon ages included in these chronologies were recalibrated to calendar years BP using IntCal13 (Reimer et al., 2013).

I developed a set of data-driven drainage basin boundaries for each of the study basins (Figs. 7–10, lower right panels) based on several lines of evidence. The first is the modern drainage basin outlines of Lehner et al. (2013). The second is the ice-sheet margin chronology of Dyke et al. (2003) and Dyke (2004), which, when lacking independent information, I use as an approximate set of contours of ice-sheet thickness. The ice divides could, however, have moved with time, offsetting them from the chronological contours, and Tarasov et al. (2012) and Gowan (2013) also note that uncertainties exist in this deglaciation chronology. Therefore, a key third set of information comes from the evidence of ice streaming recently completed by Margold et al. (2014, 2015) and the mapped Canadian eskers from Storrar et al. (2013). As there are few chronological controls on when these ice-streams were active, ice divides are simply drawn between diverging ice-streams. While this is a potential source of error, it also may not be such a bad assumption: where there are chronological controls on ice-stream reorganization, these have not been noted to change the continental-scale structure of ice divides (e.g., Stokes et al., 2009; Ross et al., 2009; Ó Cofaigh et al., 2010). Better chronologies often come from direct dating of ice-marginal positions (e.g., Licciardi et al., 1999; Dyke, 2004; Gowan, 2013), fluvial deposits (e.g., Bretz, 1969; Atwater, 1984; Ridge, 1997; Benito and O'Connor, 2003; Knox, 2007; Rittenour et al., 2007), and lacustrine deposits (e.g., Antevs, 1922; Kehew and Teller, 1994; Rayburn et al., 2005, 2007, 2011; Richard et al., 2005; Breckenridge, 2007; Breckenridge et al., 2012), and these provide the fourth layer to define drainage chronologies. The fifth layer of data comes from the offshore stratigraphic record (e.g., Leventer et al., 1982; Andrews and Tedesco, 1992; Andrews et al., 1999; Flower et al., 2004; Carlson et al., 2007a, 2009; Williams et al., 2012; Maccali et al., 2013), which can be compiled into paleohydrographs that may correlate with time-

**Table 1.** Modern drainage basin areas and water ($Q_w$) [cubic meters per second] and sediment ($Q_s$) [$10^6$ metric tons per year] discharges. Many of these were gathered and independently of the Milliman and Farnsworth (2013) compilation, and generally provide good agreement with their numbers.

| River | A [km$^2$] | $Q_w$ [m$^3$ s$^{-1}$] | $Q_s$ [Mt yr$^{-1}$] | Reference(s) |
|---|---|---|---|---|
| Mackenzie | $1.8 \times 10^6$ | 9910 | 128 | Carson et al. (1998) |
| Hudson Strait and Bay | $1.173 \times 10^6$ | 30,900[a] | ~1.1[a,b] | Schneider-Vieira et al. (1994); Bentley (2006) |
| Saint Lawrence[c] | $1.344 \times 10^{6[d]}$; $1.61 \times 10^{6[e]}$ | 12,000[d]; 14,400[e] | 3.6[d] | Holeman (1968); Doyon (1996); Dolgopolova and Isupova (2011); Environment Canada (2013) |
| Hudson River | $0.034 \times 10^6$ | 590 | 0.737 | Wall et al. (2008); Milliman and Farnsworth (2013) |
| Susquehanna[f] | $0.0704 \times 10^6$ | 1082 | 5 | Gross et al. (1978); Kammerer (1990) |
| Mississippi[g] | $3.225 \times 10^6$ | 18,430[g] | 400[h], 210[j] | Kammerer (1990); Milliman and Farnsworth (2013) |
| Rio Grande | $0.870 \times 10^6$ | 570[h,i]; 22[j] | 20[h,i]; 0.66[j] | Gates et al. (2000); Milliman and Farnsworth (2013) |
| Colorado[k] | $0.637 \times 10^6$ | ~665[h,i]; 6.3[j] | 145[h,i], 0.1[j] | U.S. Bureau of Reclamation (1952); van Andel (1964) (as cited by Carriquiry and Sánchez, 1999); Kammerer (1990); Prairie and Callejo (2005); Nowak (2011); Milliman and Farnsworth (2013) |
| Columbia | $0.668 \times 10^6$ | 7500 | 7 | Milliman and Meade (1983); Kammerer (1990) |

[a] This is the sum of river inputs to Hudson Bay from pre-1975 data

[b] Modeled following Syvitski et al. (2003)

[c] Not including modern diversions into or out of the Great Lakes (i.e., preindustrial discharge). These include the Chicago Diversion (out of the Great Lakes basin), and the Ogoki and Long Lake Diversions (into the Great Lakes basin). The drainage basin figures show the modern Great Lakes–Saint Lawrence drainage basin, which includes these diversions; as these are small compared to other sources of possible error, I have not returned the mapped Great Lakes–Saint Lawrence drainage basin to its pre-diversion state.

[d] The Great Lakes–Saint Lawrence drainage system alone

[e] Including inflow into the Saint Lawrence estuary from rivers other than the Saint Lawrence

[f] All drainages to Chesapeake Bay are included in the LGM calculations, even though only the Susquehanna is included at present (and in this table). This is because Chesapeake Bay is a ria, or drowned valley: the other rivers that drain into Chesapeake Bay joined the Susquehanna as tributaries at the LGM due to sea-level fall.

[g] This is the sum of Mississippi River (Old River) discharge and drainage area, Mississippi River discharge and drainage area routed towards the Atchafalaya River, and the Atchafalaya-Red River system. Inclusion of the latter increases drainage area from $2.978 \times 10^6$ km$^3$ to $3.225 \times 10^6$ km$^2$ and discharge from 16,790 m$^3$ s$^{-1}$ to 18,430 m$^3$ s$^{-1}$.

[h] Before the installation of river control structures (i.e., preindustrial); all other values (including those on the same line) are modern unless indicated

[i] Before modern water withdrawals (i.e., preindustrial); these values are estimated based on the work of Gates et al. (2000) for the Rio Grande and Prairie and Callejo (2005) and Nowak (2011) for the Colorado River at Imperial Dam, which was summed with inputs from the Gila River (U.S. Bureau of Reclamation, 1952); see Cohen et al. (2001) for an earlier calculation of natural discharge to the Colorado River delta, and Schmidt et al. (2010) for a review of the 20th century human–river interaction that necessitated my approximation here. Modern values of water and sediment discharge are from Milliman and Farnsworth (2013).

[j] Modern discharge to the ocean.

[k] Modern discharge values are from Milliman and Farnsworth (2013); past (preindustrial) estimates and the estimated sediment yield are from other sources.

variable drainage basin area (Wickert et al., 2013), but outside of some information on provenance (Carlson et al., 2007a, 2009), can not independently indicate where the past drainage basin margins lie.

While detailed and well-dated geological constraints permit a quantitative approach to reconstruct past discharge (Licciardi et al., 1999; Carlson et al., 2007a; Carlson, 2009; Carlson et al., 2009; Obbink et al., 2010; Wickert et al., 2013), most interpretations are much more basic, simply showing when periods of high or low discharge may have existed due to changes in drainage basin area (e.g., Ridge, 1997), isotopic ratios (e.g., Andrews and Dunhill, 2004), and/or proglacial lake outlets (e.g., Teller and Leverington, 2004; Breckenridge, 2015). A quantitative data–model comparison would be limited to a minority of both the total modeled river history and the available geologic data. I therefore took the simplest approach to interpret these records, defining them as simply "high discharge" (light and dark gray on Figs. 11–16) or "low discharge" (white background on Figs. 11–16). Dark gray indicates times for which there is geological evidence for a distinct period high flow, and light gray indicates the time over which data-based inferences indicate that enhanced flow would have occurred and/or when discharge would have been higher but not as high as during the time indicated by the dark gray regions. For example, Mississippi River discharge increased when Lake Agassiz flooded into it (one dark gray band), but was overall higher than present from the LGM until ∼12.9 ka due to ice-sheet input and a larger drainage basin area (light gray and dark gray).

## 4 Results

The result of the computational work is two major products: synthetic drainage basin and discharge histories. The former are summarized in four sets of maps showing significant points in time since the LGM (Figs. 7–10), with the full time-series of each reconstruction provided as a supplementary video and in the data repository (Section 7). The latter are a set of reconstructed deglacial water discharges by drainage basin (Figs. 11–16), with Fig. 16 including the meltwater total discharge calculations of Licciardi et al. (1999) for comparison. These provide a spatially-distributed and quantified view of how rivers in North America have broadly changed over the past 20,000 years.

The counterpart to these computed histories are those that are driven by more direct interpretations of the data (past examples of this include those by Teller, 1990a; Licciardi et al., 1999; Carlson et al., 2007a; Carlson, 2009). I write, "counterpart" as opposed to "ground-truth" because these are also based on limited data and have varying degrees of quantitative rigor. Their key utility is that they offer a much simpler path from the raw data to the interpreted drainage basin extents and periods of high discharge, and one that can be more closely tied to geological fact. No age uncertainty is provided for these data-driven reconstructions: The maps display multiple events that coincide, assembled from disparate data sources, of which not all have well-quantified error. The periods of high and low discharge are marked with sharp lines even where chronological constraints are sparse, and

should be viewed as a current geological best estimate. Nevertheless, the large-scale drainage basins drawn on Figures 7–10, each of which represents an important time in deglacial history, display the current state of knowledge for that point in the geologic time scale, regardless of chronological error. Therefore, large-scale disagreements between the data-constrained drainage basin outlines and the model results at these key points in the geological time scale generally indicate a need to improve the model-based reconstructions. Mismatches between data-driven and model-driven discharge histories are less straightforward to attribute to model error, as they involve both drainage basin area and rates of ice-sheet melt (Section 3, last paragraph). These potential mismatches are discussed below.

## 5 Discussion

### 5.1 Comparison with geologic evidence for varying drainage basin areas and discharge

Each of the river systems in this study experienced high flow and low flow at different times. High flow could be due to meltwater inputs, drainage area expansion, or an increase in $P - ET$. The first two are often linked: ice-sheet advance into mid-continent drainage basins vastly increased their areas (Figs. 7–9). Figure 11 includes a brief description of each of these times, which is expanded upon in the following sections. The data-driven drainage histories (Section 3) through North America are updated from Wickert (2014), with the dates for ice advance and retreat compiled by Licciardi et al. (1999) still broadly consistent with the modern state of knowledge. For the following chronology, all radiocarbon ages were calibrated (or recalibrated) using IntCal13 (Reimer et al., 2013). Current geologic evidence, while incomplete and subject to revision, provides a good picture of the broad changes in drainage across North America since the LGM.

### 5.1.1 Drainage histories by river

The Mississippi, Hudson, and Susquehanna Rivers traded drainage inputs from the Great Lakes and their respective former ice lobes, resulting in a pattern of symmetric increases and decreases in discharge as ice advanced and retreated along the southern Laurentide margin. At the LGM, the drainage area of the Great Lakes was split between the Mississippi and Susquehanna Rivers (Flock, 1983; Ridge et al., 1991; Ridge, 1997; Licciardi et al., 1999; Reusser et al., 2006; Knox, 2007), with the Mississippi receiving meltwater from all but a portion of Lake Ontario. The Hudson River received some northern ice-sheet inputs (Wickert, 2014) as well as flow from the Delaware River due to glacial-isostatic tilting of the land surface (Stanford et al., 2016). At 19.9 ka, ice retreat associated with the Erie Interstade (Mörner and Dreimanis, 1973; Monaghan, 1990) opened the Mohawk valley in New York, which became the preferential pathway for all of the meltwater formerly sent to the Susquehanna River, as well as water from the Lake Erie and western Lake Ontario basins that formerly drained to the Mississippi (Ridge, 1997). In spite of the loss of Mississippi-routed meltwater inputs, deposits of the Kankakee Outwash Plain (Knox, 2007) indicate increased meltwater

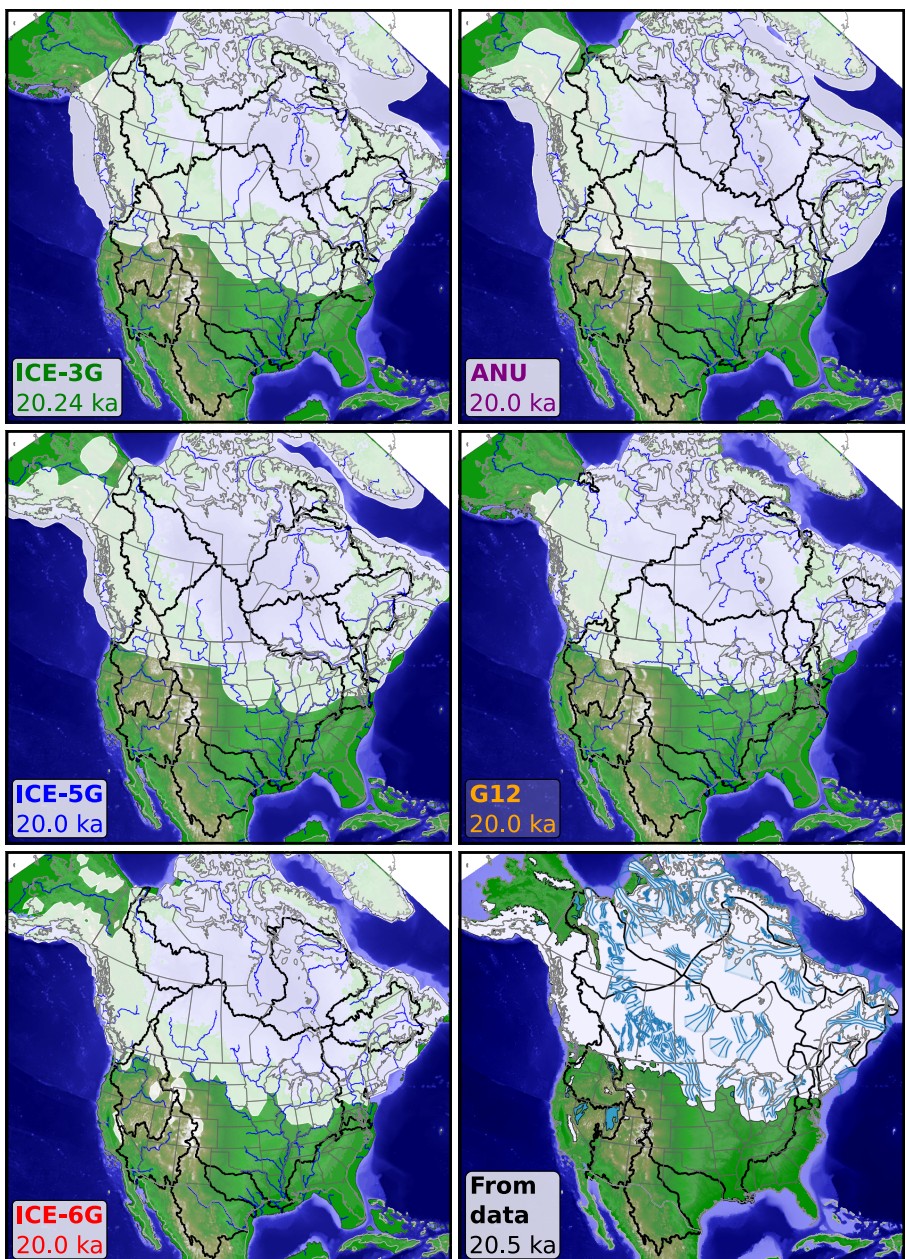

**Figure 7. End of the global LGM.** North American drainage basins at the LGM. Ice models are organized left to right, top to bottom, in the order of their publication age. Drainage basins are delimited in black. Blue lines on the model output panels trace corridors through which total water discharge (ice melt and meteoric) is $\geq 1000$ m$^3$ s$^{-1}$; these lines are based on raw outputs that are not adjusted based on modern or preindustrial discharge (Section 2.5, last paragraph). The data-driven compilation indicates evidence of past ice streams in blue (Margold et al., 2014). The model- and data-driven reconstructions show a generally consistent trend of an enlarged Mississippi River drainage basin, ice flow via Hudson Strait into the North Atlantic, and ice-sheet meltwater routed down the Hudson and/or Susquehanna Rivers.

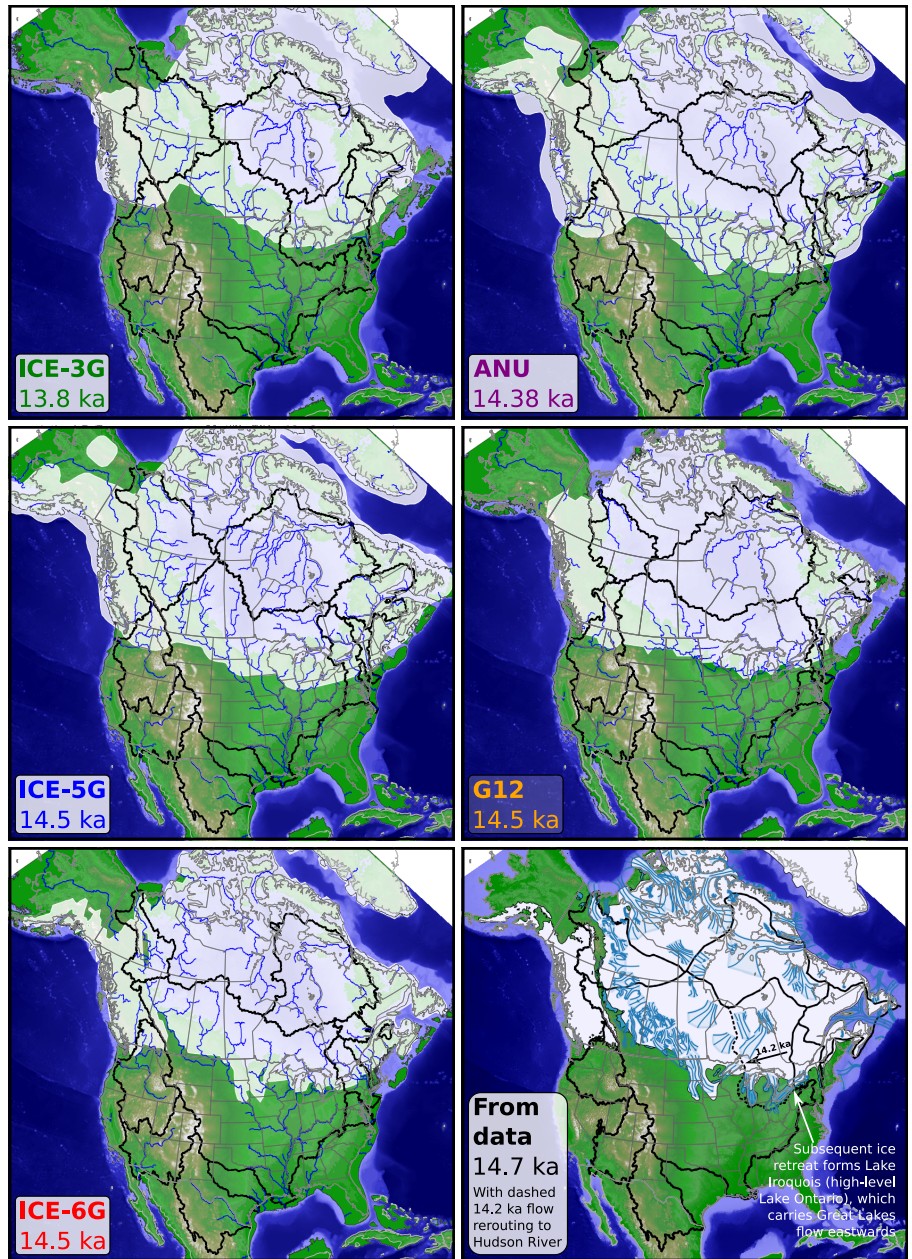

**Figure 8. Bølling-Allerød warm period and** (for all but ICE-3G and the data-driven reconstruction, for which the closest-available time-slice(s) was/were chosen) **Meltwater Pulse 1A.** The Bølling-Allerød was the major period of warming between earlier near-LGM conditions and later Younger Dryas cooling. The early part of the Bølling-Allerød, the Bølling warming, encompasses MWP-1A, a period of 14–18 m sea level rise between 14.65 and 14.31 ka (Deschamps et al., 2012). Collapse of the ice saddle between the Laurentide and Cordilleran Ice Sheets at this time, combined with generally enhanced melt rates due to the Bølling warming, could be the mechanism for rapid sea-level rise during MWP-1A (Gregoire et al., 2012; Gomez et al., 2015; Gregoire et al., 2016). Note the extension of the blue $\geq 1000$ m$^3$ s$^{-1}$ river lines for many of the ice models; this is representative of an increase in melt that increases river discharge.

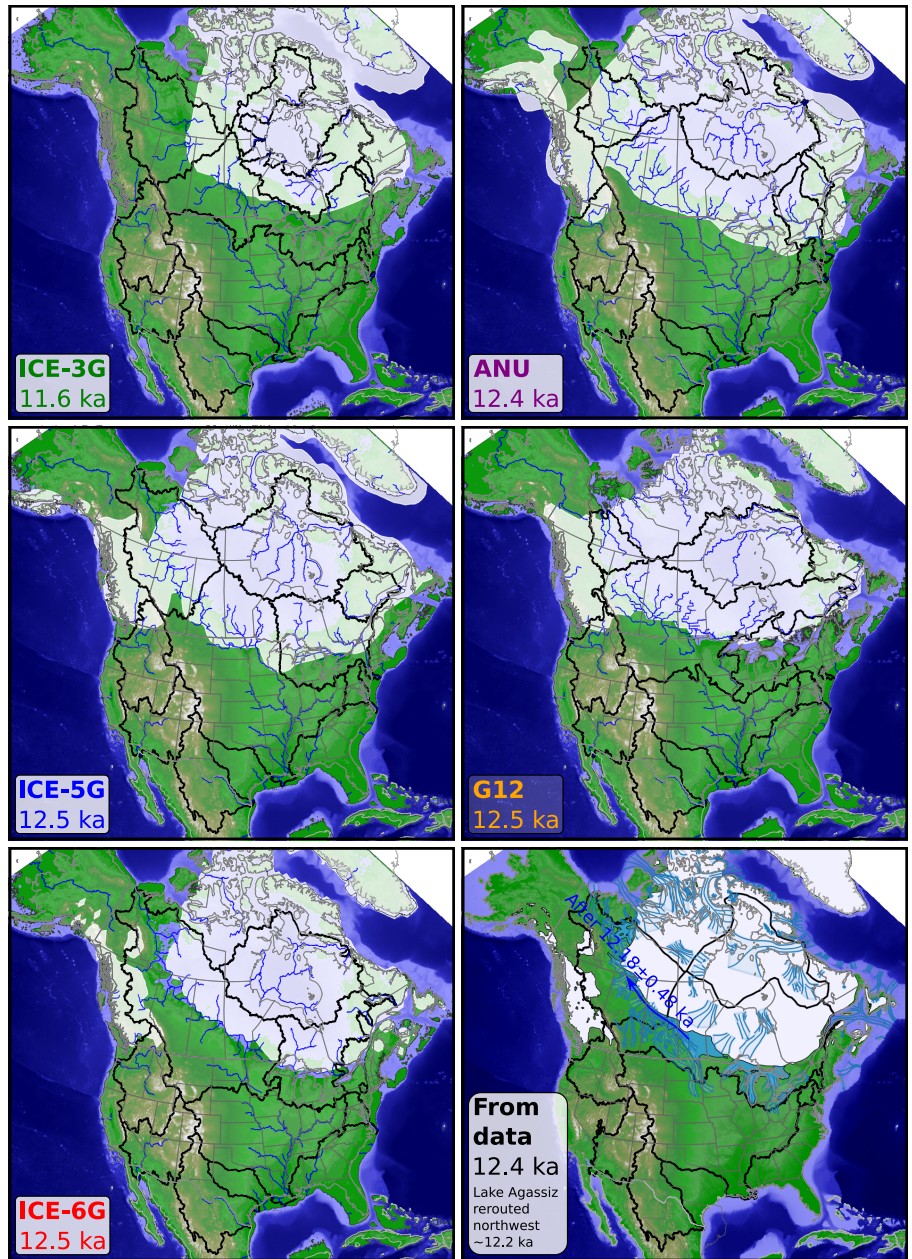

**Figure 9. Younger Dryas** (at the transition to the Holocene for ICE-3G). The Younger Dryas was a period of rapid cooling that corresponds with meltwater rerouting away from the Mississippi (e.g., Williams et al., 2012; Wickert et al., 2013) and enhanced meltwater output to the Mackenzie (Murton et al., 2010; Keigwin and Driscoll, 2014; Keigwin et al., 2016) and Saint Lawrence (e.g., Carlson et al., 2007a; Cronin et al., 2012) Rivers. Model predictions here vary widely, but all successful ice-sheet reconstructions should at least match the extensively-researched continental-scale drainage rerouting away from the Mississippi basin starting at 12.9 ka. Recent research indicates that outflow from Lake Agassiz, at the center of the continent, was first routed towards the Saint Lawrence and later (∼12.2 ka: Breckenridge, 2015) routed towards the Mackenzie River (Section 5.1.1).

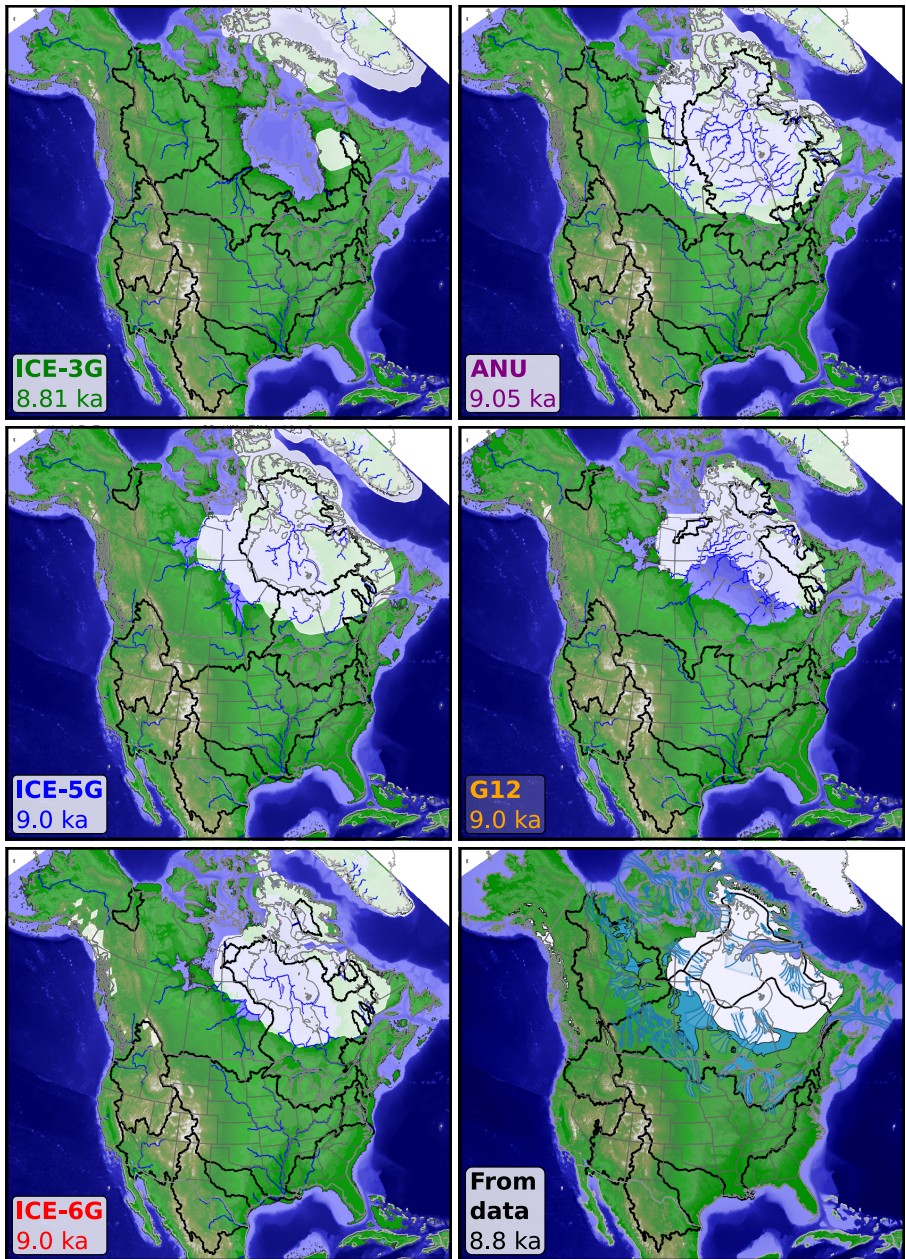

**Figure 10. Early Holocene.** The early Holocene was notable for its ice retreat and proglacial lake growth, which included enhanced flow down the Saint Lawrence River and abrupt drainage of proglacial lakes Agassiz and Ojibway into Hudson Bay. In G12, westerly flow around the ice dam in Hudson Bay could highlight a pathway for the 8.6 ka partial drainage of Lake Agassiz–Ojibway (Breckenridge et al., 2012). Many other rivers no longer had ice-sheet input. In ICE-5G, the Upper Missouri river occupies the preglacial Heart River valley in the Bismarck–Mandan area (Todd, 1914; Leonard, 1916; Bluemle, 2006) and flows into the James River, tributary to the Red River of the North, and away from the Mississippi (Section 5.1.3). This illustrates how sensitive low-gradient mid-continental river systems can be to subtle tilts of the land surface, especially when preexisting valleys and topographic slopes facilitate alternate flow paths.

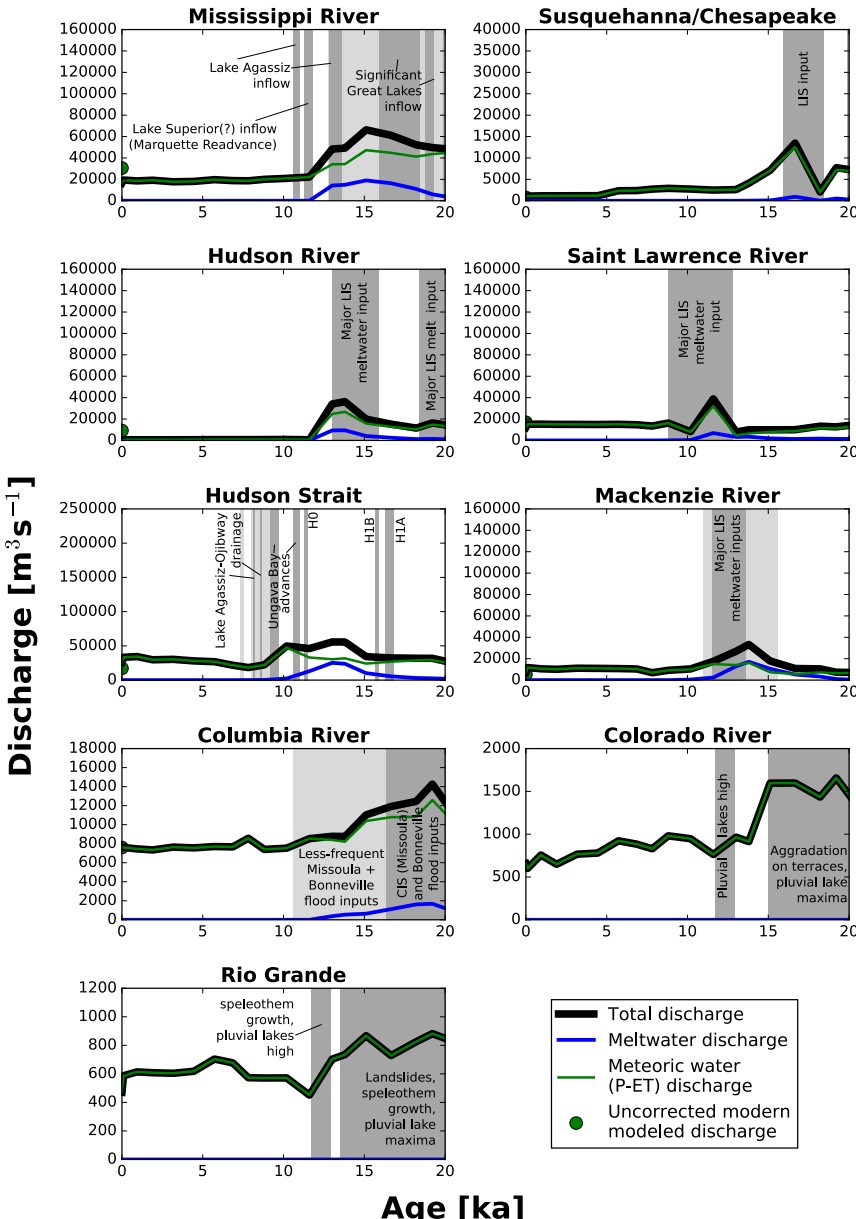

**Figure 11.** Computed discharge from ICE-3G/VM1 (Tushingham and Peltier, 1991), with gray shading at times of enhanced discharge based on the geologic record. Dark gray indicates a specific time of enhanced discharge, while light gray indicates inferred higher discharge from ice-sheet and landscape geometry or other factors, or, for the Columbia River, a time of enhanced discharge that is likely less-enhanced than discharge earlier in the record. These are annotated here with the following abbreviations: LIS – Laurentide Ice Sheet; H0, and H1B, and H1A – Heinrich Events 0, 1B, and 1A; CIS – Cordilleran Ice Sheet.

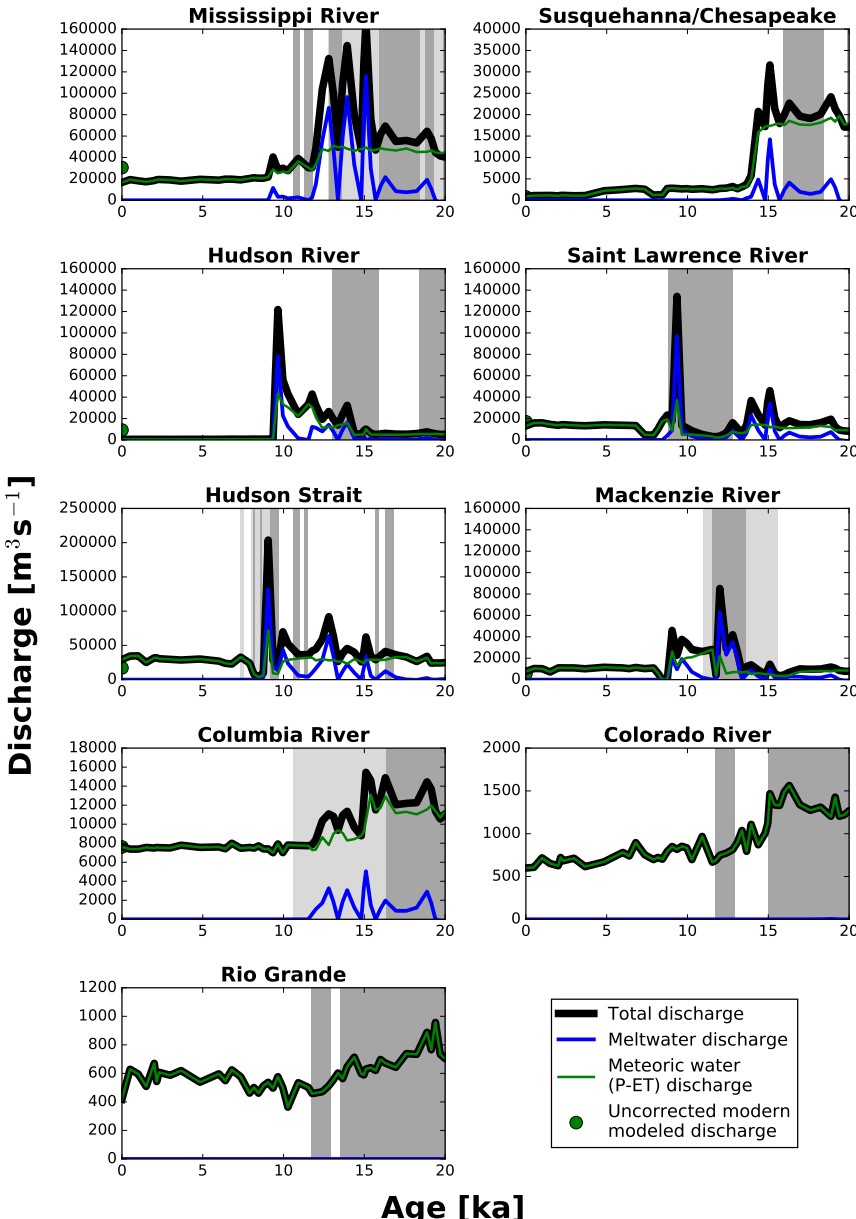

**Figure 12.** Computed discharge based on the ANU ice-sheet–solid-Earth model pair (Lambeck et al., 2002), compared to the geologic record of enhanced past discharge (gray shading: dark, direct record; light, inferred and/or less-enhanced discharge). Notes on the geologic record can be found in Fig. 11.

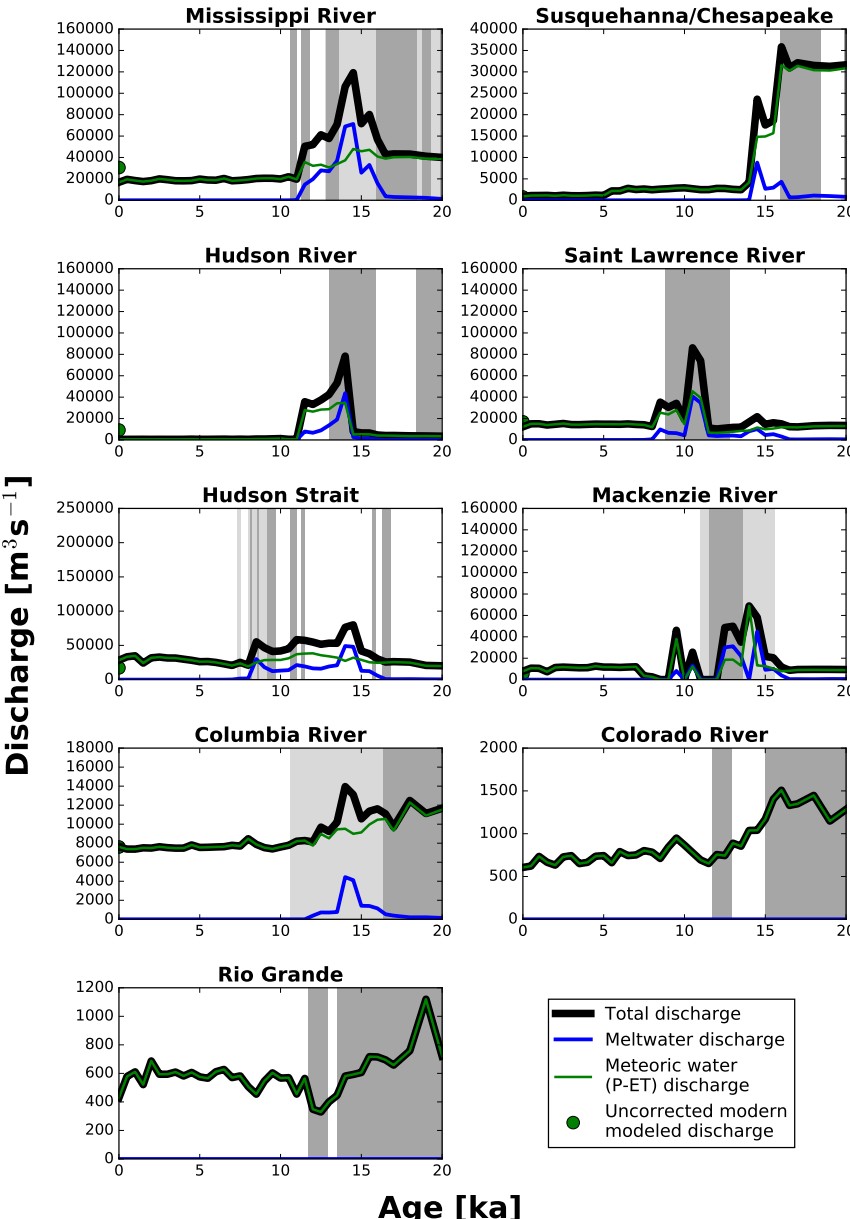

**Figure 13.** Computed discharge based on ICE-5G/VM2 (Peltier, 2004), compared to the geologic record of enhanced past discharge (gray shading: dark, direct record; light, inferred and/or less-enhanced discharge). Notes on the geologic record can be found in Fig. 11.

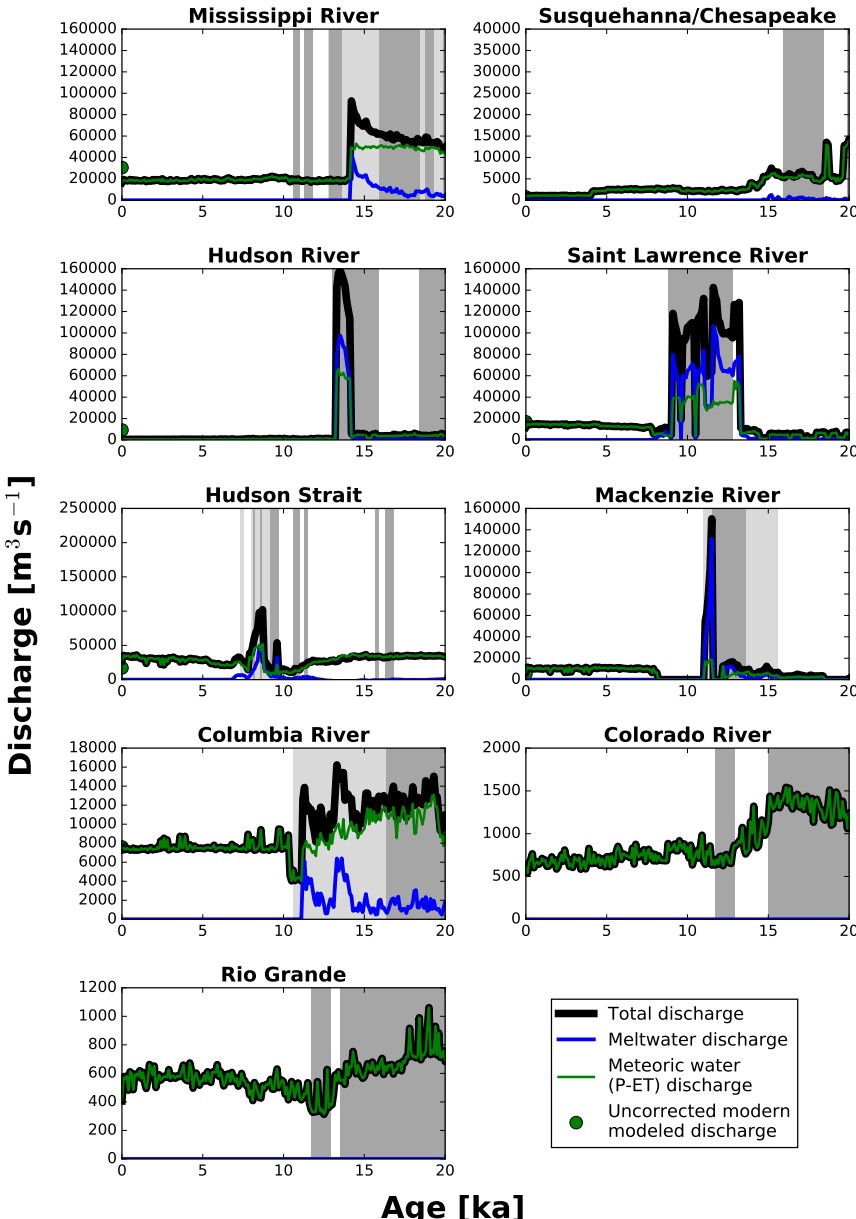

**Figure 14.** Computed discharge based on the G12/VM2 coupled ice-sheet–solid-Earth model (Gregoire et al., 2012; Peltier, 2004), compared to the geologic record of enhanced past discharge (gray shading: dark, direct record; light, inferred and/or less-enhanced discharge). Notes on the geologic record can be found in Fig. 11.

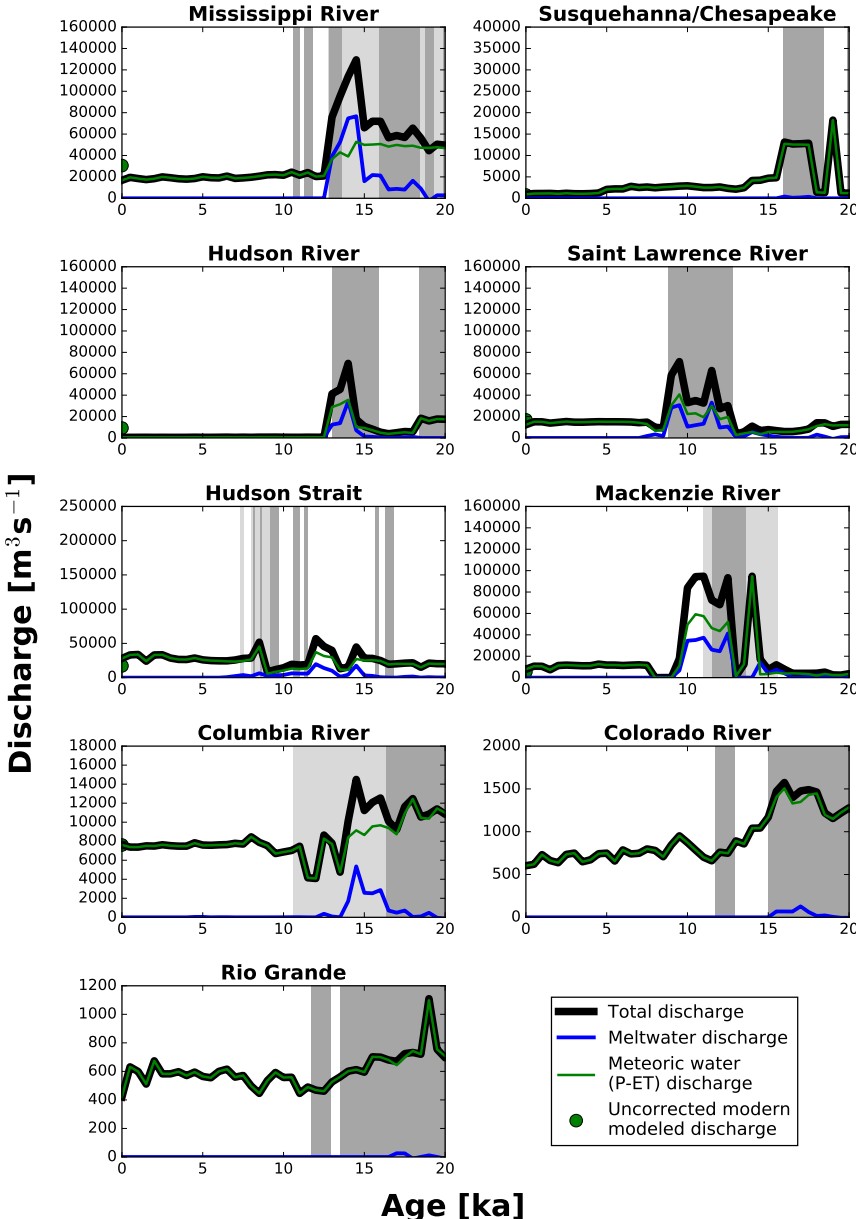

**Figure 15.** Computed discharge based on the ICE-6G/VM5a coupled ice-sheet–solid-Earth model (Argus et al., 2014; Peltier et al., 2015), compared to the geologic record of enhanced past discharge (gray shading: dark, direct record; light, inferred and/or less-enhanced discharge). Notes on the geologic record can be found in Fig. 11.

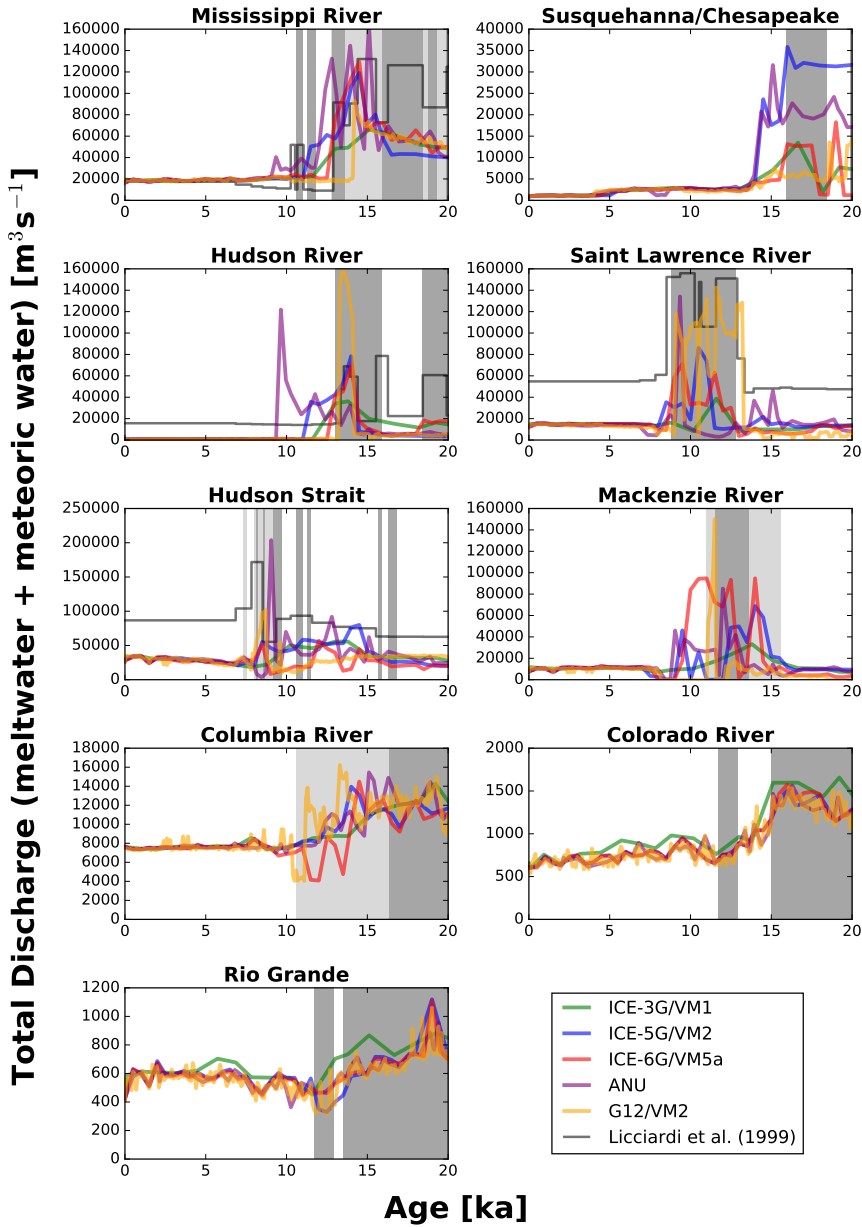

**Figure 16.** Total discharge from each river basin, represented on a single set of axes to compare the timing and absolute magnitudes of meltwater discharge among the models, with the geologic record of enhanced past discharge (gray shading: dark, direct record; light, inferred and/or less-enhanced discharge). Also shown here in darker gray lines on four panels are the data-driven discharge histories of Licciardi et al. (1999), with their original radiocarbon chronologies calibrated using IntCal13 (Reimer et al., 2013).

outflow to the Mississippi River from the Lake Michigan basin ~19.3–18.8 ka, with a major flood ~19 ka (Curry et al., 2014). Following the end of the Erie Interstade, at 18.4 ka (Licciardi et al., 1999; Reimer et al., 2013; Wickert, 2014), the Port Bruce Stade advance produced the Valley Heads Glaciation in New York, sending flow back from the Hudson River to the Susquehanna and Mississippi from 18.4–15.9 ka (see Reusser et al., 2004, 2006, for the Susquehanna terrace record). As the ice again retreated during the 15.9–15.6 ka Mackinaw Interstade, the Hudson River captured outflow from all of the Great Lakes except Superior, reducing Mississippi discharge and removing ice-sheet inputs from the Susquehanna for the final time. Ice advanced again starting at 15.6 ka towards its Port Huron Stade maximum, at which point only Lake Ontario flowed towards the Hudson River (via the Mohawk Valley) and all other Great Lakes sent their outflow to the Mississippi (Hansel and Mickelson, 1988; Licciardi et al., 1999; Obbink et al., 2010). At 14.2 ka, ice retreat during the Two Creeks Interstade rerouted outflow from all of the Great Lakes except for the western Lake Superior basin to the Hudson River (Broecker and Farrand, 1963; Ridge, 1997; Licciardi et al., 1999; Rech et al., 2012), though the Mississippi was augmented by increasing melt via the Des Moines Lobe, James Lobe, and Lake Agassiz (Patterson, 1997, 1998; Licciardi et al., 1999; Obbink et al., 2010; Sionneau et al., 2010). Outflow from Lake Michigan temporarily returned to the Mississippi during the Greatlakean Stade, 13.5–13.0 ka (Evenson et al., 1976; Hansel and Mickelson, 1988; Licciardi et al., 1999) before rapid and complete (though not final) abandonment of the Mississippi River meltwater outlet between 13 and 12.85 ka.

Much of this early history stems from the continental record of ice advance and retreat (e.g., Dreimanis and Goldthwait, 1973). As such, it is a good predictor of changes in drainage basin area, but is somewhat less effective at predicting past river discharge, especially when that discharge is strongly influenced by ice-sheet melt. Discharge inferences continue to be aided by the growing database of paleoceanographic proxy records (e.g., Keigwin et al., 2006; Obbink et al., 2010; Williams et al., 2012; Hillaire-Marcel et al., 2013; Taylor et al., 2014; Gibb et al., 2014; Gil et al., 2015), improved geochemical techniques to produce richer records that also reduce or quantify uncertainties (e.g., Vetter, 2013; Spero et al., 2015; Khider et al., 2015; Vázquez Riveiros et al., 2016), and continuing work to provide quantitative freshwater discharge estimates based on these data (Carlson et al., 2007a; Carlson, 2009; Obbink et al., 2010; Wickert et al., 2013). Particularly important are records of past water isotopes, past water temperatures, and geochemical signatures of provenance. In addition to these sediment-based methods, continental-scale constraints on ice-sheet melt can be informed by sea-level fingerprinting (e.g., Gomez et al., 2015; Liu et al., 2015). In this work, I follow Licciardi et al. (1999) in correlating the terrestrial record of ice-sheet advance through the Great Lakes with a large eastern Mississippi River drainage basin and therefore with periods of enhanced Mississippi River discharge.

Meltwater discharge down the Mississippi decreased sharply between 13.0 and 12.85 ka (Flower et al., 2004; Williams et al., 2012; Wickert et al., 2013), synchronous with ice retreat that opened

the Saint Lawrence River (Occhietti and Richard, 2003; Richard et al., 2005; Carlson et al., 2007a; Rayburn et al., 2011). At 12.9 ka the Lake Agassiz drainage subbasin was rerouted away from the Mississippi River. The the best current chronology indicates that water from the Lake Agassiz subbasin first flowed east to the newly-ice-free Saint Lawrence (Broecker et al., 1989; Carlson et al., 2007a; Carlson and Clark, 2012; Breckenridge, 2015; Levac et al., 2015; Leydet, 2016), though this rerouting did not produce any significant drawdown of the lake level. Starting at 12.18±0.48 ka, outflow from the Lake Agassiz subbasin was rerouted to the northwest towards the Mackenzie River (Andrews and Dunhill, 2004; Carlson et al., 2007a; Breckenridge, 2015). Two final periods of meltwater discharge down the Mississippi River occurred after the initial rerouting of Lake Agassiz flow. The first is recorded in a 11.8–11.3 ka $\delta^{18}$O spike in offshore sediments (Williams et al., 2012; Wickert, 2014), whose age range encompasses the ∼11.6–11.5 ka Marquette Readvance of ice in the Lake Superior basin (Lowell et al., 1999) that temporarily separated the Great Lakes–Saint Lawrence river system from Lake Agassiz (Breckenridge and Johnson, 2009) and may have sent meltwater from the Lake Superior basin back towards the Mississippi River. The second lasted ∼11.0 ka–10.6 ka (Fisher, 2003; Wickert, 2014), and was the final pulse of ice-sheet meltwater to flow down the Mississippi.

The Saint Lawrence continued to receive significant ice-sheet meltwater inputs via proglacial lakes until sometime between ∼8.6 ka, when the combined Lake Agassiz–Ojibway level dropped in a partial drainage event (Breckenridge et al., 2012), and 8.2 ka, when the ice saddle in Hudson Bay collapsed, causing a massive flood that resulted in sea-surface freshening and the brief cooling of the 8.2 ka event (Barber et al., 1999; Carlson et al., 2009; Roy et al., 2011; Hoffman et al., 2012; Gregoire et al., 2012; Breckenridge et al., 2012; Stroup et al., 2013). While this ended meltwater inputs to the Saint Lawrence River, meltwater from the Québec–Labrador dome flowed down the Manicouagan River and into the Gulf of Saint Lawrence until ∼7.8 ka (Occhietti et al., 2004).

The record of ice and meltwater discharge from Hudson Bay and Hudson Strait that predates the Agassiz–Ojibway flood comes primarily from ice-rafted debris in marine sediment cores. This is because Hudson Strait was an ice-covered marine-terminating margin of a major ice stream of the Laurentide Ice Sheet (Margold et al., 2014). The largest of the post-LGM ice-rafted debris events was Heinrich Event 1, which occurred in two pulses: H1A (16.8–16.3 ka) and H1B (15.9–15.7 ka) (Gil et al., 2015). Heinrich Event 0, formerly thought to have an age of 12.8–12.3 ka that corresponds to the Younger Dryas (Andrews and Tedesco, 1992; Clark et al., 2001), has been recently shown to correlate with early Holocene warming and melt from 11.5–11.3 ka (Pearce et al., 2015). Later ice-rafting events are related to ice advances of the Québec–Labrador dome into Ungava Bay (Kaufman and Miller, 1993; Metz et al., 2008; Rashid et al., 2014; Pearce, 2015; Jennings et al., 2015); these comprise the 11.0–10.6 ka Gold Cove advance (Pearce, 2015) and the 9.7–9.2 ka Noble Inlet advance (Jennings et al., 2015). Following these ice-rafting events, the merged Lake Agassiz–Ojibway released water into Hudson Bay in two major pulses. The first was the ∼8.6 ka partial drainage of

Lake Agassiz–Ojibway, identified by a low-water period in the varve record (Breckenridge et al., 2012). The second was the 8.2 ka complete drainage of Lake Agassiz–Ojibway (Barber et al., 1999; Breckenridge et al., 2012; Stroup et al., 2013) that is associated with collapse of the Laurentide

Ice Sheet saddle that connected the Keewatin and Québec–Labrador domes (Gregoire et al., 2012). High carbonate concentrations and low $\delta^{18}$O in sediments indicate consistent high discharge, even between these events, from the start of the Noble Inlet Advance at 9.7 ka until ~8.0 ka (Jennings et al., 2015).

After 8.2 ka, Hudson Bay continued to receive a smaller meltwater input from the ablating rem-

700 nant Laurentide ice caps (Carlson et al., 2008) that included one last ~200-year period period of enhanced discharge ~7.55–7.35 ka that appears in the $\delta^{18}$O record of Jennings et al. (2015). The total volume of the 7.5–6.8 ka final ablation of the Laurentide Ice Sheet – comprising the Keewatin, Québec–Labrador, and Baffin Island–Foxe ice domes – could have been up to 5 meters sea-level equivalent (SLE) (Carlson et al., 2007b; Jennings et al., 2015). Much of this melt would have flowed

into Hudson Bay, but would not have been significant: the maximum estimate (5 m SLE and all melt entering Hudson Bay) results in an average discharge enhancement of 82 m$^3$ s$^{-1}$, which is small compared to the 30,900 m$^3$ s$^{-1}$ total modern discharge (Table 1). Therefore, background melt outside of the $\delta^{18}$O excursion does not appear as a time of enhanced discharge on Figs. 11–16. Following final Laurentide Ice Sheet collapse, isostatic rebound with a time-averaged uplift rate of 1.3

710 cm yr$^{-1}$ (Lavoie et al., 2012) caused Hudson Bay to shrink to its modern size.

The main period of post-LGM high discharge down the Mackenzie River started ~15.6 ka with the retreat of the Mackenzie Lobe of the Laurentide Ice Sheet and associated meltwater production (Duk-Rodkin et al., 1994; Lemmen et al., 1994). Meltwater inputs increased at 13.6±0.2 ka (Smith, 1994), when glacial Lake McConnell formed in the Great Bear Lake Basin. Lake McConnell catas-

715 trophically flooded towards the Mackenzie River between ~13.6 and ~13.3 ka, possibly again at ~13.3 ka, and at ~13.1 ka. At ~13.1 ka, ice retreat rerouted Glacial Lake Peace, which formerly drained into the Missouri River (and hence, the Mississippi), towards the Mackenzie, significantly increasing the drainage area of the latter. At 12.18±0.48 ka, strandlines of Glacial Lake Agassiz indicate that it, too, started to flow to the north (Breckenridge, 2015). This age range is consistent

with the 13.0±0.2 to 11.7±0.1 bounds on the age of gravel deposits that overlie a regional erosion surface on the Mackenzie River delta, presumably the result of a large flood associated with drainage rearrangement (Murton et al., 2010; Carlson and Clark, 2012). It also postdates the youngest dated material from Glacial Lake Mackenzie (13.0±0.3 ka), which drained when The Ramparts – the largest bedrock gorge in the Mackenzie River – incised (Mackay and Mathews, 1973). This incision

may be related to drainage rerouting that increased inflow into Glacial Lake Mackenzie and caused it to overtop the bedrock sill that dammed it to the north. Enigmatically, low-$\delta^{18}$O meltwater inputs to the Beaufort Sea cease ~12.4 ka (Keigwin et al., 2016), which is within error of or predates terrestrial evidence of meltwater routing towards the Mackenzie (Murton et al., 2010; Breckenridge,

2015). Meltwater inputs to the Mackenzie River ended no later than 11 ka, when its eastern tribu-
taries were temporarily rerouted eastward due to a combination of ice retreat and glacial-isostatic
depression (see Section 5.1.3). Later northwestern drainage of Lake Agassiz during its high-water
Emerson Phase, 10.8–10.2 ka (Fisher, 2007), must therefore have flowed in a northerly course near
or along the retreating ice margin instead of entering the isostatically-constricted Mackenzie River
drainage basin.

The Columbia River has an offshore record of rapid turbidite deposition that stretches 36.1–
10.6 ka (Zuffa et al., 2000). Within this time span, large floods associated with glacial Lake Mis-
soula occurred 22.9–16.5 ka (Bretz, 1923, 1969; Waitt, 1980; Atwater, 1984; Benito and O'Connor,
2003; Lopes and Mix, 2009). Lake Bonneville routed a portion of flow from the Great Basin to the
Columbia River from 18.2±0.3 to 16.4±0.2 ka (Gilbert, 1890; Godsey et al., 2005, 2011; McGee
et al., 2012), including a massive <1-year flood at ∼17.5 or ∼18 ka that resulted from rapid ∼100-
meter incision of a natural dam of alluvium (Jarrett and Malde, 1987; Godsey et al., 2005, 2011;
Oviatt, 2015). The terrestrial record contains evidence for later, smaller outflows from glacial Lake
Missoula 15.1–12.6 ka (Hanson et al., 2012), and intermittent overflow inputs from Lake Bonneville
from 16.4±0.2 to $14.9^{+0.3}_{-0.6}$ ka (synthesis of Godsey et al., 2011; McGee et al., 2012).

Water discharge in Colorado River and the Rio Grande, while influenced by mountain glacier
retreat, responded primarily to the cooler and wetter regional climate during the LGM and Younger
Dryas that was caused by changes in continental-scale moisture transport (e.g., Kutzbach and Wright,
1985; He, 2011; Oster et al., 2015; Oster and Kelley, 2016). These cool, wet periods are marked
by high pluvial lake levels (e.g., Currey et al., 1990; Matsubara and Howard, 2009; McGee et al.,
2012; Oviatt, 2015), mountain glacier advances prior to the onset of the Bølling-Allerød (e.g., Owen
et al., 2003; Guido et al., 2007; Orme, 2008; Refsnider et al., 2008, 2009; Brugger, 2010), increased
landsliding in the Rio Grande basin between ∼21.2 and ∼14.5 ka (Reneau and Dethier, 1996),
deposition of valley fills in the Colorado River (Pederson et al., 2013), and speleothem growth (e.g.,
Polyak et al., 2004; Oster and Kelley, 2016). The warmer Bølling-Allerød and Holocene correspond
to records of mountain glacier retreat (Guido et al., 2007; Laabs et al., 2013; Munroe and Laabs,
2013) and strath terrace formation due to river incision (Anders et al., 2005; Cook et al., 2009). While
anomalous records, such as speleothem growth ∼14 ka during the Bølling-Allerød (Polyak et al.,
2004), complicate the picture of past climate in the southwest, the general pattern of climate change
closely mirrors that observed in the Greenland ice core records (Benson et al., 1997; Svensson et al.,
2008; Asmerom et al., 2010; Wagner et al., 2010).

### 5.1.2   Comparison of data and models for river discharge

In order to determine how well, broadly, these different models were able to reproduce our observed
record of discharge variability, I subdivided the computed discharges into events during known high-
flow periods (light and dark gray regions on Figs. 11–16) and those from the low-flow remainder of

the record. I first compared the mean discharges during known high- and low-flow periods (Fig. 17A). For the five ice-sheet models and nine river systems, only three times did the known low-flow periods correspond to an overall higher computed discharge: the Hudson River for the ANU model and Hudson Strait for ICE-3G and ICE-5G. The ice-physics-based G12 performed the best in generating discharges during known high-flow periods that were much higher than those during

known low-flow periods, with ICE-6G in second place and ICE-3G and ICE-5G tied for third. The strong performance of G12 may be because its input mass balance was driven by a coupled ocean–atmosphere GCM (FAMOUS: Smith et al., 2008), making its ice mass the result of a pre-computed hydrologic balance. I then generated histograms for the high-flow and low-flow segments of the model results, and performed Kolmogorov–Smirnov tests between the pairs of distributions for each

ice model and river system to test how different they are from one another (Fig. 17B). Here, the GIA-based ICE-6G and ICE-5G were second and third, respectively, to ICE-3G, though ICE-3G cannot be a correct ice-sheet reconstruction more generally because it does not include a large enough total ice mass. G12 reproduced many patterns of deglacial water discharge well, including the pulses of iceberg/meltwater output from Hudson Strait (Figure 14), but its goodness of fit is reduced by its

short (but intense) pulse of meltwater down the Mackenzie River, its lack of early meltwater delivery to the Hudson River, and a too-early switch of meltwater outflow away from the Mississippi River because its climate forcing and dynamical ice response caused the Québec–Labrador dome to retreat too early. For both of these metrics, the inconsistent time steps between ice-sheet reconstructions can affect the fits, as can be seen especially for the Rio Grande and Columbia River, which should

be the same for all runs as the same climate reconstruction was employed in all cases. Therefore, these catchments were removed from the above analysis, the broad strokes of which provide a useful template for comparison.

Across all model runs, Hudson Strait was the consistent site of the worst agreement between data and the models. It is possible that this is due to data-collection issues (e.g., few organics) and data

sparsity (few settlements and fewer GPS monuments) in this formerly fully ice-covered region, lead-ing to a lack of good regional constraints on ice-sheet thickness near the center of the Laurentide Ice Sheet. However, recent data sets have brought more clarity to the Hudson Strait record (e.g., Rashid et al., 2014; Gil et al., 2015; Pearce et al., 2015; Pearce, 2015; Jennings et al., 2015), fur-thering the argument that this mismatch could be because the models do not accurately represent the

dynamics of the marine-terminating Hudson Strait ice catchment (e.g., MacAyeal, 1993). An addi-tional complication of Hudson Bay/Strait is that its drainage area was smaller at the LGM (Fig. 7), but it had the potential to release a large amount of ice, as is observed in the Heinrich Events (e.g., Heinrich, 1988). A final potential issue is that Hudson Bay lies on the low-relief Canadian Shield in the zone of maximum glacial-isostatic deformation, meaning that its computed drainage area during

deglaciation is sensitive to GIA (Fig. 10). In summary, a proper comparison requires both long-term proxy discharge evidence and a mechanistic ice-sheet and GIA model that can explain the dynamics

of ice and solid Earth masses in Hudson Bay and Hudson Strait under a realistic climate forcing. The Glimmer–CISM (Rutt et al., 2009) runs of Gregoire et al. (2012) are able to reconstruct the general form of the pulsed meltwater releases seen in proxies (e.g., Jennings et al., 2015), putting more weight behind approaches that involve ice physics to reconstruct past ice sheets (e.g., Tarasov and Peltier, 2005, 2006; Tarasov et al., 2012; Gregoire et al., 2012; Wickert et al., 2013).

A comparison between these model results and data-driven drainage histories of Licciardi et al. (1999) (Fig. 16: Mississippi, Hudson, Saint Lawrence, Hudson Strait) includes both similarities and differences when compared to the river system reconstructions presented here. For the Mississippi River System, Licciardi et al. (1999) expect enhanced discharge between 18.4 and 16.3 ka and prior to 19.9 ka – that is, at times when all Great Lakes flow was routed down the Mississippi River. The reconstructions analyzed here exhibit enhanced discharge over the entire time prior to 16.3 ka, but with no shorter-term periods of more enhanced discharge (Fig. 16: Mississippi). The discharges predicted by Licciardi et al. (1999) are higher than any produced here, and this is because they neglect evapotranspiration. Periodicity due to drainage rearrangement, predicted by Licciardi et al. (1999) is seen in the $\delta^{18}O$ record of the Gulf of Mexico (Williams et al., 2012; Wickert, 2014) (albeit with somewhat different timing, especially earlier in the record), though this is more sensitive to the very negative $\delta^{18}O$ of meltwater discharge than to the overall freshwater discharge. The modeled Hudson River discharge does not exhibit the predicted Erie or Mackinaw Interstade periods of enhanced discharge in any of the computed discharge histories presented here. Compelling geological evidence that meltwater flowed through the Mohawk Valley into the Hudson River during the Erie Interstade (Ridge, 1997) makes this omission a point to improve in future reconstructions that include the southern and southeastern Laurentide Ice Sheet. The Saint Lawrence River constraints from Licciardi et al. (1999) are consistent with the flow routing of G12 (Gregoire et al., 2012) and ICE-6G (Argus et al., 2014; Peltier et al., 2015), even though G12 reroutes meltwater discharge from the Mississippi to the Saint Lawrence ∼1.1 ka too early (Wickert et al., 2013). The Hudson Strait Record is difficult to match because of its many episodic meltwater releases (e.g., Heinrich, 1988; Andrews and MacLean, 2003; Hemming, 2004; Gil et al., 2015; Jennings et al., 2015), and only the climate-forced and ice-physics-based G12 reproduces the general form of these events. These events are simplified or smoothed-over by Licciardi et al. (1999), but recent improvements in the age constraints on these iceberg and/or meltwater release events (e.g., Rashid et al., 2014; Gil et al., 2015; Pearce et al., 2015; Pearce, 2015; Jennings et al., 2015) set the stage for forthcoming advances in reconstructions of the northeastern core of the Laurentide Ice Sheet.

### 5.1.3 Comparison of data and models for drainage basin extents

Any model-based reconstruction that employs a realistically-shaped North American ice sheet complex will produce certain changes in continental-scale drainage. The most visible such pattern, consistent across all model-based reconstructions, is that the high Keewatin and Labrador domes of the

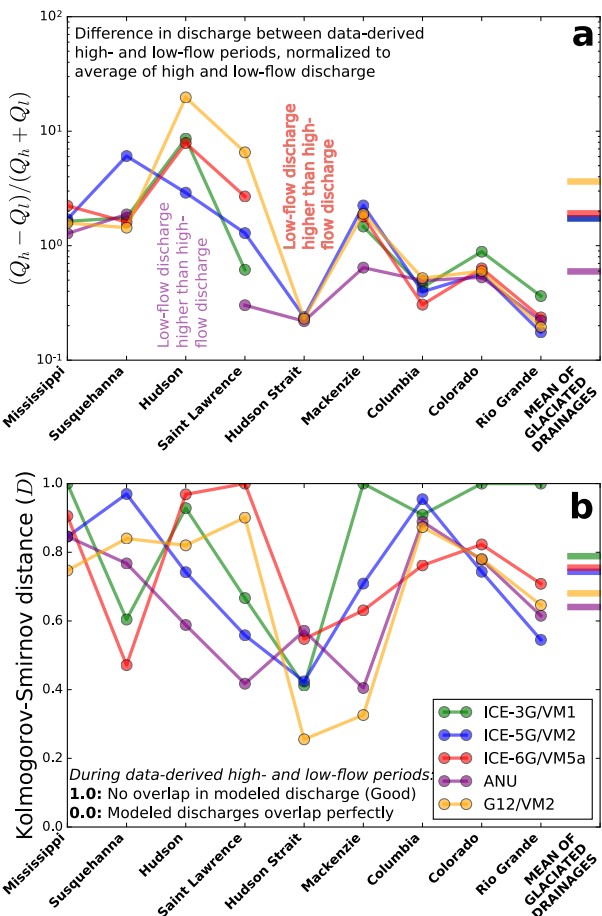

**Figure 17.** Computed discharges are classified as occurring at times of known high or low flow based on geologic record. Times of high flow in the geologic record are denoted by both light and dark gray in Figs. 11–16. **a.** Normalized difference in the modeled discharge between geologically-known high and low discharge times. The greatest mean normalized difference is seen in G12, the ice-physics-based model of Gregoire et al. (2012), and the least in the GIA-based model of Lambeck et al. (2002). **b.** The Kolmogorov–Smirnov test measures the distance between one-dimensional distributions; here I use it to compute a metric of difference between the computed discharge histograms for each river system during the generalized data-derived times of high and low discharge. A high distance value indicates a good match between the data and model. Here, the GIA-based ICE-3G and ICE-6G models perform the best, with poor matches only to Hudson Bay. The ice-physics-based G12 performs poorly, due in part to the too-early demise of its Québec–Labrador dome, and the GIA-based ANU model performs the worst.

Laurentide Ice Sheet and ice advance across the Great Lakes basins generated a new trans-continental drainage divide that added regions that currently flow into Hudson Bay and the Saint Lawrence River to the LGM Mississippi drainage basin (Fig. 7). The patterns of post-LGM drainage basin evolution, as well as regional-scale changes in drainage basin extents, vary more significantly among model-based reconstructions. Identifying differences between the model-driven and data-driven drainage basin reconstructions can guide efforts to produce a more accurate depiction of the last deglaciation.

At the LGM, ICE-5G and G12 have the most severe misfits with data. ICE-5G sends a large fraction of what data indicate should be Mississippi-River-bound drainage to the Hudson River, an artefact fixed in ICE-6G with its more realistic ice-dome structure. G12 at the LGM has Mackenzie River drainage routed towards the Yukon River, though time steps shortly before and after the LGM have the proper drainage routing. This is the result of its ice sheet briefly crossing a threshold that rerouted drainage. It is these thresholds that make drainage routing an especially useful discriminator to separate valid and invalid portions of ice-sheet reconstructions.

During Bølling-Allerød time, only ICE-6G and G12 properly show that ice retreated beyond the bounds of the Susquehanna River basin. All models simulate the Mississippi and Mackenzie catchments generally well, and close to what available data (e.g., Margold et al., 2014) would indicate.

During the early part of the Younger Dryas, the Mackenzie and Saint Lawrence Rivers both experienced periods of high discharge (de Vernal et al., 1996; Carlson et al., 2007a; Cronin et al., 2012; Murton et al., 2010; Keigwin and Driscoll, 2014; Keigwin et al., 2016). Debate continues as to whether Lake Agassiz flowed towards the Mackenzie River in the northwest (e.g., Murton et al., 2010; Keigwin and Driscoll, 2014; Keigwin et al., 2016), towards the Saint Lawrence River in the east (e.g., Carlson and Clark, 2012), or, as one group has proposed, became a closed basin at this time due to regional drying (Lowell et al., 2013). However, evidence currently indicates that Lake Agassiz drained east at the onset of the Younger Dryas (Carlson et al., 2007a; Rayburn et al., 2011; Carlson and Clark, 2012; Leydet, 2016), and then drained north at 12.18±0.48 ka (Breckenridge (2015), which is consistent with the data of Fisher et al. (2008) and Murton et al. (2010)). The drying hypothesis seems the least likely, as the model to simulate it (Lowell et al., 2013) creates a closed basin when at least two parameters seem unreasonable. These are: (1) evaporative loss that is generally >1.5 times today's observations, in spite of cooler temperatures and a longer lake-ice season, although katabatic winds may have increased evaporation somewhat over the open water (see Carlson and Clark, 2009, for more on the problem of too much evaporation), and (2) a temperature lapse rate that is 7.3 °C km$^{-1}$, outside of the error bars of the 5.1±1.2 °C km$^{-1}$ observed on modern ice sheets and ice caps Anderson et al. (2014), and resulting in a smaller melt-producing area of the Laurentide Ice Sheet. The drying hypothesis has received further critiques from Teller (2013) and Breckenridge (2015) based on field evidence, with strandline mapping by Breckenridge (2015) indicating that early eastward outflow from Lake Agassiz is plausible, and that the Moorhead Phase rapid lake-level drop (cf. Lepper et al., 2013) and associated meltwater outflow occurred dur-

ing northwesterly flow rerouting to the Mackenzie River. Regardless of where Lake Agassiz flowed, or whether it became a closed basin (the flow routing presented here does not allow closed basins to exist: see Section 2.3.2), isotopic data from the Gulf of Mexico show unequivocally that meltwater discharge to the Mississippi dropped precipitously ∼12.9 ka (Williams et al., 2012; Wickert et al., 2013). This means that the ICE-5G, ANU, and G12 reconstructions miss the timing of the most major drainage rearrangement in the North American deglaciation. In G12, this is due in part to the too-early retreat of the southeastern Laurentide margin. Overall, ICE-3G and ICE-6G perform the best at 12.5 ka, with the former sending Lake Agassiz meltwater to the Gulf of Saint Lawrence and the latter sending Lake Agassiz meltwater to the Mackenzie River and Beaufort Sea; In both ICE-3G and ICE-6G, the Mackenzie and Saint Lawrence received proximal meltwater inputs. Either routing at 12.5 ka is within error of the 12.18±0.48 ka date for drainage rerouting from the Saint Lawrence to the Mackenzie (Breckenridge, 2015), and is also consistent within error of the $12.3^{+0.3}_{-0.4}$ ka early peak in radiocarbon dates on roots from the Moorhead Phase low in the Lake Agassiz basin (Fisher et al., 2008).

Ice retreat during the terminal Pleistocene and early Holocene reveals excessive isostatic adjustment in some of the models. The overly-large isostatic depression produced by G12/VM2 causes the Great Lakes system to become a marine embayment following ice retreat, which leads to an erroneous 9.6 ka rerouting of Saint Lawrence drainage to Hudson Strait. Excess isostatic subsidence produced by ICE-5G/VM2, G12/VM2, and ICE-6G/VM5a also results in a north-flowing Upper Missouri River. While there is a plausible course for northerly flow from the Upper Missouri River into the preglacial Heart River at Bismarck, North Dakota (Todd, 1914; Leonard, 1916; Bluemle, 2006), at present no field studies indicate drainage rearrangement following the LGM.

The early Holocene in northern North America was characterized by retreat of the Laurentide Ice Sheet that led to expansion of proglacial lakes, setting the stage for the short-lived and rapid 8.2 ka cooling event when the ice dam separating Lake Agassiz–Ojibway and the sea collapsed and the lake rapidly drained into the ocean (Barber et al., 1999; Breckenridge et al., 2012; Gregoire et al., 2012). None of the models perform particularly well at this time, though some observations hold promise. ICE-3G has altogether too much ice retreat. ICE-5G, ICE-6G, and ANU have a drainage divide that splits the east–west Saint Lawrence drainage basin, which at this time extended west to the Canadian Rockies (Teller and Leverington, 2004; Breckenridge et al., 2012; Stroup et al., 2013). G12 has this continuous drainage basin, but instead of flowing into the Saint Lawrence River, water in its drainage reconstruction flows to the Arctic Ocean via a gap at the Manitoba–Saskatchewan border along the southwestern corner of the retreating ice sheet (Fig. 10). While this could be a route for the ∼8.6 ka flood associated with partial drainage of Lake Agassiz–Ojibway (Breckenridge et al., 2012, p. 53), it is not the long-term drainage route for these lakes through the Kinojévis spillway system and into the Ottawa River (Vincent and Hardy, 1977; Veillette, 1994).

A tiny early Holocene Mackenzie River basin appears in ICE-5G, ICE-6G, and G12, and the ANU model also produces a small Mackenzie drainage basin at 8.75 and 8.45 ka. In these models, rapid retreat of the western margin of the Laurentide Ice Sheet exposed an eastward-tilting isostatically-depressed land surface in the eastern portion of the modern Mackenzie River drainage basin. Flow from the eastern modern Mackenzie River drainage basin did not return to the Mackenzie River until significant isostatic rebound tilted the land surface to the west. New cores from the Beaufort Sea record increased $\delta^{18}$O between 11 and 9 ka when corrected for ice-sheet volume (Keigwin and Driscoll, 2014; Keigwin et al., 2016). This increase cannot be reconciled with changing climate, but would be consistent with a decrease in drainage basin area. ICE-5G, ICE-6G, G12, and the ANU model each include this reduced drainage basin area at different times (see supplementary videos). Therefore, the temporary reduction in Mackenzie River drainage area is sensitive to ice-sheet geometry and/or solid-Earth rheology, making it a useful constraint on ice-sheet reconstructions. Among the models studied here, G12 fits this timing the best by having a greatly reduced Holocene drainage basin area from 11.0 to 8.2 ka, although it also produced a small Mackenzie River drainage basin at earlier times.

## 5.2 Future directions: new ice-sheet reconstructions and coupling with hydrologic, climate, isotopic, archaeological, and sediment transport models and observations

The work presented here is a necessary starting point for future studies that investigate the interactions between deglacial surface-water hydrology and other components of the integrated Earth system. Combining these meltwater routing patterns with proglacial lakes and changing land cover will allow equations designed to predict sediment yield from large catchments to be employed for a continental-scale paleo-sediment-discharge reconstruction (following Overeem et al., 2005; Kettner and Syvitski, 2008; Pelletier, 2012; Cohen et al., 2013), with modern control provided by present-day sediment yields (Table 1). These sediment discharges can be compared with records of deposition (e.g., Andrews and Dunhill, 2004; Breckenridge, 2007; Rittenour et al., 2007; Williams et al., 2010) and geomorphic change (e.g., Dury, 1964; Reusser et al., 2006; Knox, 2007; Bettis et al., 2008; Anderson, 2015). The need to properly compute past lake and land cover motivates continued work with climate- and water-balance models (e.g., Collins et al., 2006; Matsubara and Howard, 2009; Liu et al., 2009; He, 2011; Blois et al., 2013; Fan et al., 2013; Ivanović et al., 2016a), which can in turn be used to improve past drainage basin and discharge reconstructions. These, together with paleogeographic reconstructions such as those presented here, can be used to reconstruct areas of archaeological interest, either as changes in shoreline positions and topography (Clark et al., 2014) or as wholesale landscape reconstructions that incorporate site-potential modeling (Fedje and Christensen, 1999; Mandryk et al., 2001; Monteleone, 2013; Monteleone et al., 2013; Dixon and Monteleone, 2014). On a global scale, several current flow-routing algorithms could be made global for better integration with ice-sheet, climate, and GIA models (Metz et al., 2011; Qin and Zhan,

2012; Braun and Willett, 2013; Huang and Lee, 2013; Schwanghart and Scherler, 2014), with the possibility to include high-resolution flow routing as part of a transient coupled GCM instead of an *a posteriori* analysis, as is presented here. Finally, high-resolution drainage routing schemes can connect models of past climate, ice sheets, and drainage routing to oxygen isotopes in sediment cores (Wickert et al., 2013) The increasingly complete collection of such records from the North American continent and continental margin (e.g., Hooke and Clausen, 1982; Remenda et al., 1994; Andrews et al., 1994; de Vernal et al., 1996; Birks et al., 2007; Carlson et al., 2007a, 2009; Breckenridge and Johnson, 2009; Lopes and Mix, 2009; Obbink et al., 2010; Brown, 2011; Hoffman et al., 2012; Williams et al., 2012; Gibb et al., 2014; Taylor et al., 2014; Ferguson and Jasechko, 2015; Hladyniuk and Longstaffe, 2016) is opening new possibilities in isotopic studies of whole-ice-sheet mass balance.

Pursuit of these targets does not preclude the search for a more representative ice-sheet reconstruction and better ways to integrate models and data. New reduced-complexity modeling rooted in ice-margin mapping and ice physics (Gowan et al., 2016a, b) spans the gap between the empirical and modeling approaches highlighted here. Surface energy balance calculations (Ullman et al., 2015) also elucidate which part of the ice sheet was melting, and when, providing a testable link to records of glacial meltwater input. Continued development of numerical ice-sheet models (e.g., Cornford et al., 2013; Le Morzadec et al., 2015) provides new avenues to reconstruct past ice-sheets with climate inputs from GCM ensembles (Gregoire et al., 2016) and route meltwater through an evolving deglacial drainage network (Tarasov and Peltier, 2006; Tarasov et al., 2012; Ivanović et al., 2014, 2016b). Such fully-coupled models, if run at a high enough spatial and temporal resolution, have the potential to recreate short-term flooding events that have left a strong mark in the geologic record of deglaciation (e.g., Kehew and Teller, 1994; Breckenridge, 2007; Murton et al., 2010). Climate and ice-sheet models may also be isotope-enabled for comparison with isotopic data from geological archives (Caley et al., 2014; Ferguson and Jasechko, 2015; Gasson et al., 2016). Each of these approaches presents new opportunities to test deglacial drainage patterns and converge towards a solution for the shape and evolution of the North American ice-sheet complex that integrates data from glacial, terrestrial, and marine systems.

## 6 Conclusions

Most ice-sheet reconstructions, including all those examined here, are built to fit sea-level records, glacial isostatic adjustment measurements, and/or moraine positions; some are built using ice-physics-based simulations. The models in this study were not calibrated to drainage basins or meltwater pathways, which leave behind geomorphic and stratigraphic markers that can be used to characterize the plausibility of hypothesized ice-thickness distributions and deglaciation histories.

Significant scatter in the data–model fit among the ice-sheet reconstructions tested leaves no clear "winner" among these models, though a few concluding remarks can be made. ICE-6G is based on GIA and ice margins, but also possesses an ice-dome structure that is consistent with geological evidence of ice-flow paths. Its generally good performance is likely tied to this combined geological and geophysical basis. G12, while only loosely calibrated to match the outlines of the North American ice sheet complex, nonetheless reproduces many changes in continental-scale drainage. Its basis in a climate-driven process-based numerical model helped it to generate periods of high and low river discharge that were largely in agreement with the geological record, and permitted tightly-spaced output time steps that enable better comparison between the reconstruction and the geologic record. Unexpectedly, the venerable ICE-3G performed the best in the Kolmogorov–Smirinov test for times of high and low discharge; this was due largely to its strong fit with the data-driven Mackenzie River record.

Beyond the focus on deglaciation and ice-sheet geometry, the dynamic deglacial drainage basins of North America are significant in their own right. The new paleohydrographs presented here provide new insights into the ice-age legacy imprinted on the major rivers of the continent and quantify the mid-20th-century observation that many North American rivers are "underfit" in their glacial-meltwater-carved valleys (Dury, 1964). These paleohydrographs and drainage basin changes provide the paleogeographic framework for new advances in paleohydrology and fluvial geomorphology.

The goal of this work is to move towards a self-consistent paleogeographic framework within which models and geologic records may be quantitatively compared to build new insights into past glacial systems and climate–ice-sheet interactions. When integrated more broadly into studies of North American deglaciation, these time-varying deglacial drainage basins and paleohydrographs are poised to improve reconstructions of past climate and ice-sheet geometry.

## 7  Code and model output availability

All computer code for this research is available on A. Wickert's GitHub repository, at https://github.com/awickert/ice-age-rivers, and a fork of this repository is available from the University of Minnesota Earth Surface Processes GitHub organization at https://github.com/umn-earth-surface/ice-age-rivers. The drainage basin and discharge histories developed for this work are provided there as well, and are deposited for long-term storage at the Data Repository for the University of Minnesota (DRUM), at http://hdl.handle.net/11299/182076. Complete time-series of maps of deglacial drainage such as those shown in Figures 7–10 are also available from DRUM. These images portray past elevations and sea level (background map), drainage basin extents (black outlines), ice extents (white filled contour), and rivers with discharge that is $\geq 1000$ m$^3$ s$^{-1}$ (blue lines). Movies created from these images are stored at DRUM and are also available as a supplement to this article. The compiled mean global precipitation and evapotranspiration data products may be downloaded from

https://github.com/awickert/global-precipitation-evapotranspiration. Software is available under the GNU General Public License v3, and climate data compilations, model outputs, and all other non-software products are licensed as Creative Commons Attribution-ShareAlike (CC-BY-SA 3.0).

*Acknowledgements.* Jerry Mitrovica provided GIA model outputs and Feng He supplied TraCE-21K GCM outputs, both of which were crucial to this work. CMAP Precipitation data were provided by the NOAA/OAR/ESRL PSD, Boulder, Colorado, USA, from their web site at http://www.esrl.noaa.gov/psd/. Ruža Ivanović, Lauren Gregoire, and Kelly Monteleone helped with stimulating discussion and ideas about where to take this work in the future. Conversations with Bob Anderson about glacial impacts on the morphology of large rivers inspired this work, and reviews from Andy Breckenridge and Lev Tarasov as well as comments from Anders Carlson improved the final publication. ADW was supported by the US Department of Defense through the National Defense Science & Engineering Graduate Fellowship Program, the US National Science Foundation Graduate Research Fellowship under Grant No. DGE 1144083, the Emmy Noether Programme of the Deutsche Forschungsgemeinschaft (DFG) through funds awarded to T. Schildgen under Grant No. SCHI 1241/1-1, and start-up funds from the University of Minnesota.

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
