# Peer review of "Reconstruction of North American drainage basins and river discharge since the Last Glacial Maximum"

_Earth Surface Dynamics, 2016_

## Referee Comment (RC1) · B. Andy (Referee) · 23 Mar 2016

General Comments

This is an outstanding first attempt to model North American late glacial freshwater discharge. I appreciate the side by side comparison that results from different ice sheet models, and a synthesis of the extensive literature concerned with estimating meltwater routing and discharge from North America. Licciardi et al (1999) is the oft-cited study to which this effort closely compares – but Licciardi et al is based on data interpretations versus the models used here. It could be useful to compare the Licciardi data to the output here and discuss the differences and similarities directly (perhaps in Fig. 15)?

The synthesis of the freshwater routing provided by this study is excellent, but there is a lot of uncertainty in the chronology of meltwater routing– especially with respect to dating. Agassiz routing is particularly contentious – but it's also the most intensively studied. Part of this challenge is attributed to the size of the basin, but I also think the more lines of evidence that are uncovered, the greater the potential for uncertainty (at least at first).

Another challenge I see when comparing model output and data is that the geologic record is often highlighted by short term (<100 yrs) flood events (e.g. Kankakee Torrent,  Champlain freshening event, Murton flood, 9.3 event, etc) – and this modeled output has to average discharge over longer periods of time. Some of the flood events must have resulted from short term release of meltwater stored in lakes, which this modeling attempt cannot resolve (perhaps in the future?) But I also wonder if melt rates may have changed suddenly at times due to climate (cf. pg. 11, 315-323). The varve records from Lake Agassiz, Hitchcock, Superior, and Ojibway clearly illustrate that varve thickness (which must correlate with discharge) could rapidly fluctuate on decadal scales – and the Hitchcock records shows this was at times the result of climate change. I would think that combined, these two factors (storage of meltwater in lakes, and climate change on 10-100 yr scales), may have had dramatic effects overall freshwater flux.

It'd love to see a future attempt to model $\delta^{18}O$ output for the modeled discharge.  Can these ice sheet models be combined with a dynamic oxygen isotope model for the LIS  - as well as a model for d18O precipitation? I wonder how the $\delta^{18}O$ composition of meltwater changed through time – prior attempts to estimate glacial meltwater discharge assume a static $\delta^{18}O$ composition for meltwater (or range of values). We see decreasing $\delta^{18}O$water values for Lake Agassiz and Minong, and these have been interpreted as evidence for increasing meltwater discharge – but is this valid? Perhaps meltwater was becoming progressively enriched in O-16 as the ice sheet decayed?  Long term, the best records to reconstruct freshwater discharge continue to be $\delta^{18}O$ records (both from the interior, and offshore), and this seems to be an important unknown to interpreting these records. Circulation dynamics and the influence of climate on the overflow of water from the large pro-glacial lakes could be another important complicating factor.

Specific and Technical comments

Abstract/Conclusions – for me examining the differences between the ice sheet models is an important contribution  - but there's no mention in the abstract or conclusions regarding how well the models compare with the synthesis provided. Your synthesis suggests the 6G model performs better than predecessors – and I wonder if this is worth drawing attention to?

Pg. 2(43) – $\delta^{18}O$ records can provide estimates of meltwater volume, e.g. Moore et al., 2000 (YD interval & outflow from the LIS, Paleoceanography 15(1), 4-18.) I think the treatment of the $\delta^{18}O$ composition of Lake Superior overflow and LIS is treated too simply in this model – but it's an excellent attempt to get at meltwater volume from geologic data.

Pg 2, line 48. "connective tissue" is an odd phrase and ensure is repeated. The sentence could be simplified.

Pg 3, lines 67-74, fig. 4b – it's odd to see native names given for rivers. Later on in the text, all the rivers are simply referred to by their English name. Perhaps just use the official US, Mexican, and Canadian names? - otherwise you are making errors of omission. For example, the Inuit name for the Mackenzie is omitted, as are native names for the Colorado and Columbia. My suspicions is that there were other native names for the Hudson and St. Lawrence.

Pg. 4 (102) – missing parenthesis

Pg. 5 (144-145) I'm confused by the idea that the St. Louis River is progressively capturing the Upper Mississippi R -  I don't see this is Dean and Philips (a challenging field guide to decipher), and haven't looked at van Hise and Leith 1911. Presumably this is at the divide near Floodwood? It seems like that in general southern basins would capture northern basins due to rebound?

Pg. 6 (182) – perhaps which recent models include a more massive LIS

(pg. 7, sec. 2.1.3) I wonder about the accuracy of the GIA models – the ice sheet models are visually easy to compare to the Dyke ice sheet reconstructions – but this isn't true for GIA, but there are clear flaws in GIA models for the mid-continent after these data are compared to strandline reconstructions (e.g. Lewis et al, GPQ, 2005). I don't have a suggestion, I just wonder how we can go about assessing and describing the spatio-temporal accuracy of the various GIA models.

Pg. 8 (255), add "in" after changes

Pg. 9 (263-265) I can image the 30 arcsecond grid may create problems for areas near drainage divides, where the potential sills have poorly approximated elevations? It seems like it would be a good idea to compare elevations in the DEM used to higher resolution topographic data – and modify the 30arcsecond DEM if necessary. I'm thinking about headwater areas – such as west of Lake Nipigon, or across potential NW outlet divides of Lake Agassiz.

Pg. 12 (351) consider using Caribou "sub-basin" rather than basin, as this is a sub-basin of the great Lake Superior basin.

Pg. 12 (359), "Our" is used? – consider "the"

Pg 15 (407) missing parenthesis after Figure 5

Pg. 15 (426) missing period before For

Pg. 18 (448) seems like the eskers and ice streams should only provide good evidence for ice divides for time periods when each were active.

Pg. 19 (481) perhaps cite the other discharge studies driven by direct interpretations of data

Pg. 30 (521) perhaps add Levac et al., 2015 (GPC 130: 47-65)

Pg. 30 (522) I'd remove the reference to Breckenridge, 2015. It looks like the NW outlet didn't open until water dropped from the Tintah, and the model I used for rebound would place after the onset of the YD. I'd argue the main Moorhead low did not occur until long after the onset of the YD.

Pg. 30 (528) Roy et al (2011) QSR 30: 682-692 would be a key publication for the timing of Ojibway drainage. As you note in line 539, our work shows there were two drainages – an initial partial drainage, and a second complete drainage sometime later. The first drainage is not accounted for in current models/hypotheses. I would guess that Lake Agassiz drained first in Manitoba, then either completely re-filled from closer/blockage of that drainage pathway, or the divide between Ojibway and Manitoba was covered by ice advance, re-filling Lake Ojibway. I'm curious if your models suggest other drainage possibilities (water routed from Agassiz to the Mackenzie via Wollaston-Athabasca?)

Pg. 31 (557) I'd change or add the reference, or modify the text. Fisher and Lowell (2012) specifically state that Agassiz could not have gone NW "until 10.6-10.1 ka cal BP)". IMO they are ignoring contrary evidence in Fisher et al. 2002 and Fisher and Souch 1998, and place too much confidence on the utility of basal radiocarbon ages to establish ice margins – but they don't suggest a 10.8 drainage in Fisher and Lowell (2012).

Pg. 34 (624) most favored hypothesis for NW routing is too generous. As noted earlier, until the Tintah can be directly dated, the strandlines suggest Agassiz did not drain NW until after the YD onset. I have unpublished sediment core data that suggests Agassiz drained east prior to routing NW – as others have argued, there was likely an early Moorhead low – caused by the opening of eastern routes – which was followed by a major Moorhead low when a NW path opened.

Pg. 34 (626) – correct the phrase "as the model simulate it"

Pg 34. (630) missing parenthesis around Anderson et al., 2014

Pg 35 (653) where is this gap to the southwest exactly? This could be an explanation for early (pre-8.2k) partial drainage of Lake Ojibway?

*Figures*

I might keep the scale of the axes consistent between the different model runs.

On the paleogeographic maps, only the "from data" figures include political boundaries – I find these useful when looking at the details between each map.

It might be useful to see total discharge to the oceans through time compared for all the models.

---

## Referee Comment (RC2) · L. Tarasov (Referee) · 3 Apr 2016

This is a worthwhile study comparing downslope drainage of GIA-based reconstructions (and one chronology from a glaciological model) against geological inferences. It is interesting to see how consistent or not GIA reconstructions are with inferred deglacial meltwater drainage.

The inclusion of only one glaciological model that lacks any data documented calibration and that is not self-consistent with the GIA model used to infer deglacial drainage in this study is a bit unfortunate.

Technically, the drainage computation is sound. However, as detailed below, more consideration needs to be given to uncertainties in the inferred "data-driven" reconstruction. This is especially the case wrt temporal uncertainties in the inferred high/low drainage. "~15.6 ka" has no clear interpretation.  I've also noted some errors in the inferred chronologies stemming from out of date references with old age models. I am not on top of the current litterature and as such there may be more such errors.

There are also some inaccurate statements and some poor referencing as detailed below.

I am curious why proglacial lakes were ignored. The GRASS solver could easily have computed these and would have enabled some quantitative data comparisons as opposed to the purely qualitative comparison currently.

Anway, once comments below are addressed, I would support publication in ESurf. The topic is appropriate for Esurf, figures and abstract are appropriate,...

** detailed comments

generating up to âM-^H¼4 km of high-albedo ice-surface topography (Kutzbach and Wright, 1985; Ull- man et al., 2014)
**Neither of those references are appropriate for a claim of "~4 km" elevation**
**(and high-albedo is obvious). Cite an appropriate primary source.**

but are evaluated using limited (Tarasov and Peltier, 2006) to no (all others) geologic evidence for past drainage patterns.
**Incorrect as worded, Tarasov et al, 2012 used the same geological strandline**
**data from 2006 as constraints.**

Reconstructions of past ice-sheet thickness have proliferated (e.g., Tushingham and Peltier, 1991; Lambeck et al., 2002; Peltier, 2004; Tarasov and Peltier, 2006; Tarasov et al., 2012; Gregoire et al., 2012; Argus et al., 2014; Peltier et al., 2015), ...  Instead, they are tested against terminal moraine positions and glacial isostatic adjustment,
**Incorrect. The models are either tuned or calibrated to these**
**constraints. This is different than "testing".**

**"glacial isostatic adjustment" what? That is a physical**
**process. Do you mean "records of"?**

**Also, Gregoire et al, 2012 was not tested nor tuned against**
**terminal moraine positions nor glacial isostatic adjustment records**
**contrary to what I'm inferring you are currently stating.**

with the latter being a response that is spread across hundreds of to
~1000 kilometers due to the high flexural rigidity of the lithosphere
**Incorrect scales. Yup, there is some smoothing to order 4**
**Lithospheric thickness (so order 400 km for a start), but I can also**
**get significantly different RSL response between 2 sites that are**
**less than 50 km apart or you can also look at RSL records to also**
**note such resolution sensitivity**

many studies do not include any well defined
picture of drainage basin evolution. This....
**With putting out this partial straw-man, why is there**
**no mention of the Tarasov and Peltier, 2006 that does**
**actively compute the self-consistent drainage chronology,**
**and that documents the change in drainage basins?**

In general, there exists a lack of recognition of the importance of
Pleistocene drainage rearrange- ment on river systems.
**You are creating another unsupported strawman, especially since you**
**subsequently cite more references that do recognize the changes**
**compared to those that don't. Yes there has been a problem to date**
**with most (but not all) GCMs using fixed present-day routing, but**
**there is a lot more to paleo science than GCM modelling.**

This ice-sheet surface is chosen to drive the flow-routing
calculations because this is what drives ice flow (cf. Cuffey and
Paterson, 2010),
**"drive" makes no sense. The ice sheet surface is used for flow**
**routing because you are extracting surface drainage. This is**
**independent of the physics of ice flow.**

Greenland Ice Sheet, where the subglacial topography is interpolated
from the etopo1 1-arcminute global topographic data set (Amante and
Eakins, 2009).
**I suggest you obtain a more current DEM for Greenland for any future**
**work where accuracy is critical**

H_i, ice-sheet thickness,...therefore are interpolated using an
iterative nearest-neighbor approach to remove stepwise discontinuities
that would otherwise introduce artifacts in the flow-routing
calculations.
**Please make the description clear and precise. How do you decide**
**where the ice margin is when downscaling from order 1 degree**
**resolution to 30 arc-second resolution?  This would be a critical**
**issue for getting drainage correct, especially when considering**
**switching between Mississippi, St. Lawrence and Arctic drainage for**
**Lake Agassiz region.**

Ice physics, on the other hand, are sensitive to the thermal evolution
of the ice-sheet, related thermomechanical effects on ice
rheology, and the ice basal conditions
**Thermal evolution is a derivative uncertainty as the physics of heat**
**transfer is well represented in current ice sheet models. The**
**primary uncertainty for paleo glaciological ice sheet modelling is**
**the climate forcing (which drives the thermal evolution).  For**
**regions with present-day ice cover, another primary uncertainty is**
**the basal conditions (topography, roughness, sediment cover,**
**geothermal heat flux).**

Therefore, I investigate both types of models.

**Hardly. You only examine one glaciological model and one that was**
**subject to few constraints.**

and matches geological evidence for the locations of the ice domes and
ice-sheet outlines
**Misleading: Geological evidence (ie the Dyke 2004 ice margin**
**chronology) for ice-sheet outlines is IMPOSED on ICE6G.**

GMSL was ~135 m lower than present
**Given different interpretations of what GMSL means (relative to center of**
**mass of earth, relative to present-day shorelines,..), you need to define**
**exactly what you mean (Note, Lambeck 2014 eustatic sealevel is neither of**
**the two definitions I provided above in the brackets)**

This gravitationally self-consistent sea level theory and its
numerical implementation are de- scribed by Mitrovica and Milne (2003)
and Kendall et al. (2005)
**Keen to stay out of the politics of the GIA community, but it is**
**scientific convention to also cite the originators of a theory and**
**not just a recent description of it.**

truncated at spherical harmonic degree and order 256. This satisfies
the Nyquist flexural wavenumber for the solid Earth models employed,
thereby allowing the solutions to be interpolated to an arbitrarily
high resolution
**Provide a reference for "Nyquist flexural wavenumber" (couldn't find**
**one on google, web of science,..).  Most readers will not understand**
**what this means and even more will not know it's value is**
**determined...**

VM5a is largely a simplification of VM2,
**Not really. Better to describe it as a modification of VM2 (yes it**
**has a simplified structure but it also has 10^21 Pa s viscosity**
**layer between the Lithosphere and mantle that is not present in VM2.**

I initiate the GIA modeling of G12 in equilibrium at the LGM
**Should mention that is a source of error, as there is no evidence to**
**support isostatic equilibrium for North America at LGM**

Our flow-routing calculations neglect geomorphic change for three
major reasons. First, signif- icant changes Lake Agassiz outflow
directions occurred as a precursor to spillway incision
**This approximation is reasonable since you are not modelling lake**
**levels and are therefore not making direct comparisons to**
**strandlines.**

Precipitation and evaporation inputs changed through time
(Fig. 2). These changes were calculated by Liu et al. (2009) and He
(2011) using a continuous LGM (22 ka) to present run of the National
Center for Atmospheric Research (NCAR) Community Climate System Model
version 3 (CCSM3) (Collins et al., 2006).
**Should mention that the TraCE-21K simulation used ICE-5G as a**
**boundary condition, and so will not be self-consistent with other**
**deglacial ice thickness reconstructions you are using.**

Flow routing and drainage basins can be calculated only on time steps
when the ice-sheet and GIA models provide surface topography (Section
2.3, below), but the ice-sheet contribution to runoff re- quires
ice-sheet thickness to be differenced, and thus should be most valid
for the times halfway between the ice thickness time-steps.
**There is no basis for the last claim when you have 500 year**
**timesteps (eg ICE5G). On that note, you need to explicitly provide**
**what timesteps were used**

This means that these models implicitly include the lakewater volume
within the ice mass.... as ice must be used in the models to represent
the surface loads of these large lakes in the geologic past....
**Incorrect and no easy choices here. Since most of the proglacial**
**lake sites lack proximal RSL constraints, there is little basis to**
**assume that the GIA models are implicitly taking pro-glacial lake**
**loads into account (except wrt global ice volume required to match**
**far-field constraints). Most proglacial lake regions are more**
**constrained in current GIA models by present-day vertical**
**velocities, but this has much poorer time resolution. Furthermore,**
**models such ICE-5G have their ice load spatial extent set to (though**
**with no clear documentation whether any discrepancies are allowed)**
**the Dyke et al 2004 ice margin chronology. This means they have no**
**load where there are proglacial lakes.**

**It would be best to repeat at least one model calculation with**
**inclusion of pro-glacial lake loads to quantify the uncertainty.**

models to match the ob-served ice-sheet geometry,
**"observed"? Sure wish we had observations of paleo ice-sheet geometry...**

While it is essential to ignore local depressions in the DEM
**Why? "Local depressions" = lakes = more data for comparision.**

This is an improvement over âM-^@M-^\hosingâM-^@M-^] experiments (e.g., Condron
and Winsor, 2012)
**This citation does not make sense for this context. Better to cite**
**PMIP III. Unlike PMIP III hosing that distributed meltwater fluxes**
**across a band of of the North Atlantic, Condron and Winsor for the**
**first time resolved what would happen (all be it for only a few**
**years) to discharge from major river outlets. If you are citing**
**Condron and Winsor as an example of why PMIP style hosing has no**
**geophysical basis, then you need to rewrite the sentence to make**
**this clear**

The fully-distributed GCM meltwater inputs create a more realistic
ice-age ocean
**A bit simplistic. It is unclear whether fully-distributed GCM**
**meltwater inputs can create a more realistic ice-age ocean in**
**current paleoclimate modelling AOGCMs since these models lack the**
**resolution to accurately resolve meltwater flux transports and**
**turbulent mixing**

I developed a set of data-driven drainage basin boundaries for each of
the study basins ... The second is the ice-sheet margin chronology of
Dyke et al....
**There are many significant uncertainties in this approach, that need**
**some tabulation For instance, the Dyke et al chronology has**
**significant temporal uncertainty (cf Tarasov et al, 2012), that**
**could significantly affect deglacial drainage chronologies.**

The second is the ice-sheet margin chronology of Dyke et al. (2003);
Dyke (2004), which, when lacking independent information, I use as an
approximate set of contours of ice-sheet thickness
**Does not make sense. How do you get from an ice margin chronology to**
**ice sheet thickness contours?**

As such, disagreements between these data-derived basins and those
derived from models, especially where the data-driven drainage basins
are tightly-constrained
**For such interpretation, you then need to provide uncertainty**
  estimates for the data driven approach

but by 19.9 ka,
**I'm assuming calendar before present? Needs to be clarified**

**Figures 6-9: "from data" panel: what is the age uncertainty?**

**Figures 10-15 the high/low/none inferred discharge shading needs**
**a visually representation of temporal uncertainties in the transitions**
**eg, with hatching or someother visual texturising**

18.2âM-^@M-^S17.7 ka (Rashid et al., 2003)
**You need update your citations, this is not a consensus estimate**

Heinrich Event 0, corresponding to the Younger Dryas,
was 12.8âM-^@M-^S12.3 ka (Clark et al., 2001)
**Again, this is an out of date termination estimate.**
**Current chronologies end it around 11.6 ka**

**Your age extraction from old references also ignores recent**
**refinements in ice core age chronologies and C14 calibration**

 started ~15.6 ka
**what does "~" mean? +/- 0.1 ka, +/- 1 ka or ?**

At 13.1 ka, ice retreat rerouted Glacial Lake Peace
**What dating scheme is going to give 100 year accuracy at 13.1 ka???**

I then generated histograms for the high-flow and low-flow segments of
the model results, and performed KolmogorovâM-^@M-^SSmirnov tests between the
pairs of distributions for each ice model and river system to test how
different they are from one another
**again how was temporal uncertainty taken into account?**

The ice-physics-based G12 performed the best in generat- ing
discharges during high-flow periods that were much higher than those
during low-flow periods.
**Again need to make clear what timesteps were used. Was the above due**
**to G12 being provided at higher temporal resolution or not?**

(1) evaporative loss that is generally >1.5 times todayâM-^@M-^Ys
observations, in spite of cooler temperatures and a longer lake-ice
season
**you ignore the likelihood of much higher mean wind speeds near an**
**ice margin and their resultant impact on evaporative loss. This**
**needs to be mentioned as a potentially offsetting factor to your**
**one-sided critique.**

---

## Editor Comment (EC1) · J. K. Hillier (Editor) · 4 Apr 2016

Dear Dr Wickert,

Your paper now has two thorough reviews. Both are encouraging. The least enthusiastic describes the study as 'worthwhile', and offers constructive criticism with substantial technical advice. As such, I strongly encourage you to submit a revised version of your paper, as usual fully taking into account and responding to the reviewers' comments.

All the best

John

---

## Author Response (AR1)

**Responses to referee comments on "Reconstruction of North American drainage basins and river discharge since the Last Glacial Maximum" by A. D. Wickert**

I would like to thank both reviewers for their thorough comments. Having a geologist and a geophysicist/ice-sheet modeler review this work covers much of the breadth of its projected readership, and both of your thorough questions and comments have lead to an improved revised version of the manuscript.

Furthermore, I received reviews during the revision process from Anders Carlson. While these are not official reviews, they have been helpful, and I have included these here and my responses to them for completeness. Due to a combination of these comments and my desire to then do a final proofreading, this document differs somewhat from my earlier response to reviewers.

– Andy Wickert (01 June 2016)

*Responses to Reviewer 1 (Breckenridge)*

General Comments

This is an outstanding first attempt to model North American late glacial freshwater discharge. I appreciate the side by side comparison that results from different ice sheet models, and a synthesis of the extensive literature concerned with estimating meltwater routing and discharge from North America. Licciardi et al (1999) is the oft-cited study to which this effort closely compares – but Licciardi et al is based on data interpretations versus the models used here. It could be useful to compare the Licciardi data to the output here and discuss the differences and similarities directly (perhaps in Fig. 15)?

*The Licciardi et al. (1999) estimates are now included in Fig. 15. You are right – it is instructive to have these side-by-side. I also added reference to Marshall and Clarke (1999) to invoke another former integrated ice-sheet—meltwater runoff model.*

*I have also addressed this comparison, and some general features of the calculations featured here, in a new paragraph as follows:*
*"A comparison between these model results and data-driven drainage histories of \citet{Licciardi1999} (Fig. \ref{fig:Q_all_rivers}: Mississippi, Hudson, Saint Lawrence, Hudson Strait) includes both similarities and differences when compared to the river system reconstructions presented here. For the Mississippi River System, \citet{Licciardi1999} expect enhanced discharge between 18.4 and 16.3 ka and prior to 19.9 ka -- that is, at times when all Great Lakes flow was routed down the Mississippi River. The reconstructions analyzed here exhibit enhanced discharge over the entire time prior to 16.3 ka, but with no shorter-term periods of more enhanced discharge (Fig. \ref{fig:Q_all_rivers}: Mississippi). The discharges predicted by \citet{Licciardi1999} are higher than any produced here, and this is because they neglect evapotranspiration. Periodicity due to drainage rearrangement, predicted by \citet{Licciardi1999} is seen (albeit with different timing) in the $\delta^{18}$O record of the Gulf of Mexico \citep{Williams2012, Wickert2014}, though this is more sensitive to the very negative $\delta^{18}$O of meltwater discharge than to the overall freshwater discharge. The modeled Hudson River discharge does not exhibit the predicted Erie or Mackinaw Interstade periods of enhanced*

*discharge in any of the computed discharge histories presented here. Compelling geological evidence that meltwater flowed through the Mohawk Valley into the Hudson River during the Erie Interstade \citep{Ridge1997} makes this omission a point to improve in future reconstructions of the southeastern Laurentide Ice Sheet. The Saint Lawrence River constraints from \citet{Licciardi1999} remain consistent in this work with the flow-routed G12 model \citet{Gregoire2012} and ICE-6G \citep{Argus2014, Peltier2015} providing good fits to these constraints in spite of G12 rerouting meltwater discharge from the Mississippi to the Saint Lawrence $\sim$1.1 ka too early \citep{Wickert2013nature}. The Hudson Strait Record is difficult to match because of its many episodic meltwater releases \citep[e.g.,][]{Heinrich1988, Andrews2003, Hemming2004, Gil2015, Jennings2015}, and only the climate-forced and ice-physics-based G12 reproduces the general form of these events. These events are simplified or smoothed-over by \citet{Licciardi1999}, but recent improvements in the age constraints on these iceberg and/or meltwater release events \citep[e.g.,][] {Rashid2014, Gil2015, Pearce2015, Pearce2015comment, Jennings2015} set the stage for forthcoming advances in reconstructions of the northeastern core of the Laurentide Ice Sheet."*

The synthesis of the freshwater routing provided by this study is excellent, but there is a lot of uncertainty in the chronology of meltwater routing– especially with respect to dating. Agassiz routing is particularly contentious – but it's also the most intensively studied. Part of this challenge is attributed to the size of the basin, but I also think the more lines of evidence that are uncovered, the greater the potential for uncertainty (at least at first).

*Yes: with one data point, the result seems clear. With maybe 10, not so obvious; with maybe 100, trends appear and we are able to piece together the answer. So hopefully we can move towards having enough data to put together better reconstructions.*

*I added this sentence prefacing the data setting, "Current geologic evidence, while incomplete and subject to revision, provides a good picture of the broad changes in drainage across North America since the Last Glacial Maximum."*

Another challenge I see when comparing model output and data is that the geologic record is often highlighted by short term (<100 yrs) flood events (e.g. Kankakee Torrent, Champlain freshening event, Murton flood, 9.3 event, etc) – and this modeled output has to average discharge over longer periods of time. Some of the flood events must have resulted from short term release of meltwater stored in lakes, which this modeling attempt cannot resolve (perhaps in the future?) But I also wonder if melt rates may have changed suddenly at times due to climate (cf. pg. 11, 315-323). The varve records from Lake Agassiz, Hitchcock, Superior, and Ojibway clearly illustrate that varve thickness (which must correlate with discharge) could rapidly fluctuate on decadal scales – and the Hitchcock records shows this was at times the result of climate change. I would think that combined, these two factors (storage of meltwater in lakes, and climate change on 10-100 yr scales), may have had dramatic effects overall freshwater flux.

*The model time-steps are 100 to ~1000 years, meaning that the model results will average over these events. In my experience, those events that create a strong geomorphic record are related to large past events (e.g., flooding) but do not persist over enough time to be important on the scale of total meltwater release. The upside is that the increase in application of numerical modeling means that time steps will become shorter in the future, with the possibility to investigate short-term patterns of*

*change… though my prediction is that for the time being, we will be left with evaluating these events on a case-by-case basis.*

*I have added a sentence to a new paragraph on time steps that reads,*
*"This means that the model outputs will generally average over short-term geologic events that are responsible for creating some of the major spillways and deposits \citep[e.g.,][]{Kehew1994, Breckenridge2007, Murton2010}."*

*And this one to a new paragraph on numerical modeling,*
*"Such fully-coupled models, if run at a high enough spatial and temporal resolution, have the potential to recreate short-term flooding events that have left a strong mark in the geologic record of deglaciation \citep[e.g.,][]{Kehew1994, Breckenridge2007, Murton2010}."*

It'd love to see a future attempt to model d18 O output for the modeled discharge. Can these ice sheet models be combined with a dynamic oxygen isotope model for the LIS - as well as a model for d18O precipitation? I wonder how the d18 O composition of meltwater changed through time – prior attempts to estimate glacial meltwater discharge assume a static d18 O composition for meltwater (or range of values). We see decreasing d18 Owater values for Lake Agassiz and Minong, and these have been interpreted as evidence for increasing meltwater discharge – but is this valid? Perhaps meltwater was becoming progressively enriched in O-16 as the ice sheet decayed? Long term, the best records to reconstruct freshwater discharge continue to be d18 O records (both from the interior, and offshore), and this seems to be an important unknown to interpreting these records. Circulation dynamics and the influence of climate on the overflow of water from the large pro-glacial lakes could be another important complicating factor.

*Well, you've anticipated the general track! I wrote a paper in 2013 in which I do this in a simplified way for only the Mississippi drainage. A few colleagues and I have a proposal in review to integrate isotope-enabled ice-sheet models and the isotopic records for all of North America from the LGM to present. Not sure if we'll get the funding, but in general – yes, agreed!*

*On the track of isotopes in lakes, I have now seen a new paper on Lake Ontario d18O, and have included that reference, as well as your review Great Lakes oxygen isotope papers. (I usually cite original sources, but the list was getting very long for the end of a discussion section.)*

*From a more general standpoint, I have added a new paragraph about future directions in the ice-sheet reconstruction realm:*
*"Pursuit of these targets does not preclude the search for a more representative ice-sheet reconstruction and better ways to integrate models and data. New reduced-complexity modeling rooted in ice-margin mapping and ice physics \citep{Gowan2016, Gowan2016gmd} spans the gap between the empirical and modeling approaches highlighted here. Surface energy balance calculations \citep{Ullman2015} also elucidate which part of the ice sheet was melting, and when -- providing a testable link to records of glacial meltwater input. Continued development of numerical ice-sheet models \citep[e.g.,][]{Cornford2013, LeMorzadec2015} provides new avenues to reconstruct past ice-sheets that may be coupled directly with deglacial drainage evolution \citep{Tarasov2006, Tarasov2012}. Such fully-coupled models, if run at a high enough spatial and temporal resolution, have the potential to recreate short-term flooding events that have left a strong mark in the geologic*

*record of deglaciation \citep[e.g.,][]{Kehew1994, Breckenridge2007, Murton2010}. Climate and ice-sheet models may also be isotope-enabled for comparison with isotopic data from geological archives \citep{Caley2014, Ferguson2015, Gasson2016}. Each of these approaches presents new opportunities to test deglacial drainage patterns and converge towards a solution for the shape and evolution of the North American ice-sheet complex that integrates data from glacial, fluvial, and marine systems."*

Specific and Technical comments

Abstract/Conclusions – for me examining the differences between the ice sheet models is an important contribution - but there's no mention in the abstract or conclusions regarding how well the models compare with the synthesis provided. Your synthesis suggests the 6G model performs better than predecessors – and I wonder if this is worth drawing attention to?

*On my first edits in response to reviewer comments, I did add mention of ICE-6G in several places. However, after I added the comments from both reviewers and from Anders Carlson to my data-driven time-series of high discharge periods, ICE-6G did not seem so obviously the best choice – and indeed no model did. Therefore, in the end, no model is mentioned in the abstract as being exceptional, and I have added the following paragraph to the conclusions:*

*Now added to the conclusions:*
*"Significant scatter in the data--model fit among the ice-sheet reconstructions tested leaves no clear ``winner'' among these models. Nevertheless, ICE-6G and G12 each perform well enough, and for distinct reasons, to be noted here. ICE-6G is based on GIA and ice margins, but also posesses an ice-dome structure that is consistent with geological evidence of ice-flow paths. Its generally good performance is likely tied to this combined geological and geophysical basis. G12, while only loosely calibrated to match the outlines of the North American Ice Sheet complex, nonetheless reproduces many changes in continental-scale drainage. Its basis in a process-based numerical model enhanced the separation between known high- and low-flow periods, as both the G12 ice sheet and the geological record were driven by climate forcings, and permitted tightly-spaced output time-steps that allow better comparision between the reconstruction and the geologic record."*

*In reading over my abstract, I realized that I did not state which drainage basins were analyzed, so I have added that there.*

Pg. 2(43) – d18 O records can provide estimates of meltwater volume, e.g. Moore et al., 2000 (YD interval & outflow from the LIS, Paleoceanography 15(1), 4-18.) I think the treatment of the d18 O composition of Lake Superior overflow and LIS is treated too simply in this model – but it's an excellent attempt to get at meltwater volume from geologic data.

*True. I think I was thinking of just viewing the raw data when I said that they didn't give volume, because they certainly can! Added:*
*"Geologic data provide a timeline of meltwater delivery to one basin or another \citep[e.g.,][]{Ridge1997, Licciardi1999, Clark2001, Flower2004, Knox2007, Rittenour2007, Carlson2009GRL, Breckenridge2009, Murton2010, Hoffman2012, Williams2012, Breckenridge2015}, with some types of*

*data, such as oxygen isotopes, being able to be analyzed to produce reconstructions of meltwater discharge \citep{Moore2000, Carlson2007, Carlson2009, Obbink2010, Wickert2013nature}."*

Pg 2, line 48. "connective tissue" is an odd phrase and ensure is repeated. The sentence could be simplified.

*That did look strange. I removed everything after the comma, which solves both issues.*

Pg 3, lines 67-74, fig. 4b – it's odd to see native names given for rivers. Later on in the text, all the rivers are simply referred to by their English name. Perhaps just use the official US, Mexican, and Canadian names? - otherwise you are making errors of omission. For example, the Inuit name for the Mackenzie is omitted, as are native names for the Colorado and Columbia. My suspicions is that there were other native names for the Hudson and St. Lawrence.

*It is true that these are errors of omission, but there is something that doesn't sit properly with me when we use only the European names for places that people named long before.*

*I have decided to remove the other names from the text (using only the primary English-language names), but keep them and augment them on Figure 4b, following your note that they are incomplete. I have retained the note that I wrote for the Figure 4b caption that indicates that the list of languages is incomplete.*

*The short version, I suppose, is to say that not perfectly including all non-European-American names for the rivers is better than discounting the names that were used before Euro-Americans arrived.*

*I include the following text in the figure caption,*
*"Names are also given in European non-English languages where they are common, alongside a non-exhaustive set of common names in local languages."*

Pg. 4 (102) – missing parenthesis

*Thanks!*

Pg. 5 (144-145) I'm confused by the idea that the St. Louis River is progressively capturing the Upper Mississippi R - I don't see this is Dean and Philips (a challenging field guide to decipher), and haven't looked at van Hise and Leith 1911. Presumably this is at the divide near Floodwood? It seems like that in general southern basins would capture northern basins due to rebound?

*I'm glad you mentioned to take another look at this – since the study was one just looking at maps in 1911, and hasn't been followed up (maybe something for an undergraduate student project?), I have added some weasel words, "...is believed to have progressively captured Upper Mississippi drainage near Savanna Portage...".*

*I haven't gotten my hands on an original copy of Dean and Philips either, but the salient parts are reproduced in a Minnesota Historical Society article from 1932: http://www.jstor.org/stable/20161011?seq=1#page_scan_tab_contents*

*In case you cannot access the article, I've copied the text excerpt and map below:*

The streams of the Duluth escarpment descend very steeply to Lake Superior; few of them head more than 4 or 5 miles from Lake Superior . . . the greatest distance being 12 to 14 miles, in contrast with lengths of 30 to 75 miles on the north and northeast shores of Lake Superior. Many of them have as steep an average grade as 150 to 250 feet to the mile . . . the general average being 80 to 160 feet to the mile. No one of these rather tumultuous streams has cut a significantly deep valley in the face of the escarpment and most of them have only cut short gorges with small rapids and waterfalls.

Quite in contrast with these steep-graded, rapidly falling streams of the escarpment are the leisurely flowing streams of the plateau surface above. The Cloquet, the upper St. Louis, and various other rivers have an average slope of about 8 or 10 feet to the mile. It is well established that a rapidly flowing stream with a steep grade is able to deepen its valley rapidly and to extend its headwater area so that it encroaches upon the area drained by an adjacent leisurely flowing stream . . . capturing and diverting the latter or some portion of its headwaters. Stream captures or piracies, as they are called, of this kind are common. We should expect, then, that in the course of stream development for a great length of time several of the swiftly flowing streams of the escarpment would have extended their headwaters back to the region drained by the leisurely flowing streams of the plateau surface and captured part or all of these drainage systems. The fact that many of the large streams have not done so is evidence of their youth.

The largest stream in the region, however, seems to have already done just what would be expected . . . and it is natural that the largest stream should be able to do this first. St. Louis River, cutting back at a point near the end of the escarpment where it is rather low, has been able to extend its headwater region northwestward until it has captured the southwestward-flowing Cloquet and the southwestward-flowing stream that forms the present headwaters of the St. Louis itself. These captured streams had been a part of the leisurely drainage system of the plateau surface, and, it seems certain, were within the Mississippi basin. . . . Indeed, a large valley extending southwestward from the town of Floodwood, where the St. Louis now turns abruptly to the southeast, indicates that this is probably the latest elbow of capture at which the piratical St. Louis

[Figure]

PRESENT DRAINAGE SYSTEMS OF THE ST. LOUIS AND MISSISSIPPI
RIVER HEADWATERS

[Figure]

ANCIENT DRAINAGE SYSTEMS OF THE ST. LOUIS AND MISSISSIPPI
RIVER HEADWATERS
[Maps redrawn from Van Hise and Leith, *Geology of the*

Pg. 6 (182) – perhaps which recent models include a more massive LIS

*"While it does not include enough ice to reproduce the full magnitude of the observed LGM global sea level fall \citep{Austermann2013, Lambeck2014}, it provides a counterpoint to the more recent ice-sheet reconstructions, all of which include a much more massive North American ice-sheet complex (Figure \ref{fig:Vice_Qice}).*"

(pg. 7, sec. 2.1.3) I wonder about the accuracy of the GIA models – the ice sheet models are visually easyto compare to the Dyke ice sheet reconstructions – but this isn't true for GIA, but there are clear flaws in GIA models for the mid-continent after these data are compared to strandline reconstructions (e.g. Lewis et al, GPQ, 2005). I don't have a suggestion, I just wonder how we can go about assessing and describing the spatio-temporal accuracy of the various GIA models.

*There are also inconsistencies (and consistencies) with GPS data, GRACE, paleoshorelines, …*

*The GIA response computed by the combination of the ice sheet reconstructions and their respective solid Earth models is generally not calibrated to lake data from the mid-continent. This is currently on my mind and to-do this, and has been for some time. It is beyond the scope of this current work, but one study that does incorporate them is brand-new: Gowan et al. (QSR: 2016) http://www.sciencedirect.com/science/article/pii/S0277379116300646*

*I have been talking with Evan Gowan about trying to use his model for all of North America, and he considers this to be a priority as well; I think his approach of applying a bit of the physics but keeping it fairly simple is a good way to take a first stab.*

*Of course, Tarasov (the other reviewer) has considered lake records as well for his ice sheet and GIA model.*

Pg. 8 (255), add "in" after changes

*Oops! Thanks!*

Pg. 9 (263-265) I can image the 30 arcsecond grid may create problems for areas near drainage divides, where the potential sills have poorly approximated elevations? It seems like it would be a good idea to compare elevations in the DEM used to higher resolution topographic data – and modify the 30arcsecond DEM if necessary. I'm thinking about headwater areas – such as west of Lake Nipigon, or across potential NW outlet divides of Lake Agassiz.

*For this work, at least, I think that the relief around drainage divides has not been so important – in a fairly low-relief landscape in which drainage is rearranged by km-thick ice masses and ~100's of meters of GIA, topographic variability is relatively small. Further, modifying a 1-km-scale continent-wide DEM by hand would be incredibly time-consuming and somewhat subjective, so I would prefer to find an automated way to do this, if possible.*

*Developing a method for modifying the continental-scale DEM to be hydrologically correct as it were modified over geologic time, on the other hand, could be an approach to take in the future.*

*Another option for future work would be to just do these calculations at 3 arcsecond resolution; the whole deglacial cycle currently runs overnight on a desktop computer, so if this could be parallelized onto 100 processors, it would be the same runtime. This would require interpolation (and therefore smoothing) over the ocean basins, but I guess that this could be an acceptable price to pay for precision over the continents.*

*And finally, just to add some context – most such drainage calculations are run at 0.5-1 degree resolution. So this is already a lot better than the old standard.*

Pg. 12 (351) consider using Caribou "sub-basin" rather than basin, as this is a sub-basin of the great Lake Superior basin.

*That makes sense! Changed.*

Pg. 12 (359), "Our" is used? – consider "the"

*Done.*

Pg 15 (407) missing parenthesis after Figure 5

*Fixed.*

Pg. 15 (426) missing period before For

*Thanks, fixed.*

Pg. 18 (448) seems like the eskers and ice streams should only provide good evidence for ice divides for time periods when each were active.

*Yes; hence the following admission-of-approximation sentence, "As there are few chronological controls on when these ice-streams were active, ice divides are drawn between diverging ice-streams."*

*I have also added "While this is a potential source of error..." immediately after this.*

Pg. 19 (481) perhaps cite the other discharge studies driven by direct interpretations of data

*Added four citations*

Pg. 30 (521) perhaps add Levac et al., 2015 (GPC 130: 47-65)

*I hadn't seen that paper – added a reference to it.*

Pg. 30 (522) I'd remove the reference to Breckenridge, 2015. It looks like the NW outlet didn't open until water dropped from the Tintah, and the model I used for rebound would place after the onset of the YD. I'd argue the main Moorhead low did not occur until long after the onset of the YD.

*I messed this up. I have removed the reference, and added in a sentence after my critiques of the drying hypothesis, as follows:*
*"The drying hypothesis has received further critiques from \citet{Teller2013} and \citet{Breckenridge2015} based on field evidence, with strandline mapping by \citet{Breckenridge2015} indicating that the level of Lake Agassiz dropped when it flowed to the northwest, though current evidence suggests that Lake Agassiz drawdown postdates the onset of the Younger Dryas."*

Pg. 30 (528) Roy et al (2011) QSR 30: 682-692 would be a key publication for the timing of Ojibway drainage. As you note in line 539, our work shows there were two drainages – an initial partial drainage, and a second complete drainage sometime later. The first drainage is not accounted for in current models/hypotheses. I would guess that Lake Agassiz drained first in Manitoba, then either completely re-filled from closer/blockage of that drainage pathway, or the divide between Ojibway and Manitoba was covered by ice advance, re-filling Lake Ojibway. I'm curious if your models suggest other drainage possibilities (water routed from Agassiz to the Mackenzie via Wollaston-Athabasca?)

*It would be a key publication! I've added it now.*

*That is a good question – where did the first drainage go? I've attached a figure from right before this time (next page). From this, it would seem that a drainage towards the NW (but not the Mackenzie River), via the Manitoba–Saskatchewan northern border region, could be plausible along the very subsided portion of the ice-sheet margin.*

*The following now appears in the paper; let me know if there is anything else of yours that I should cite related to this possibility:*
*"G12 has this continuous drainage basin, but instead of flowing into the Saint Lawrence River, water in its drainage reconstruction flows to the Arctic Ocean via a gap at the Manitoba--Saskatchewan border along the southwest corner of the retreating ice-sheet (Fig. \ref{fig:mapsEH}). While this could be a route for the $\sim$8.6 ka flood associated with partial drainage of Lake Agassiz--Ojibway \citep{Breckenridge2015}, it is not the long-term drainage route for these lakes through the Kinoj{\'e}vis spillway system and into the Ottawa River \citep{Vincent1977, Veillette1994}."*

[Figure]

08.8 ka

*G12*

Pg. 31 (557) I'd change or add the reference, or modify the text. Fisher and Lowell (2012) specifically state that Agassiz could not have gone NW "until 10.6-10.1 ka cal BP)". IMO they are ignoring contrary evidence in Fisher et al. 2002 and Fisher and Souch 1998, and place too much confidence on the utility of basal radiocarbon ages to establish ice margins – but they don't suggest a 10.8 drainage in Fisher and Lowell (2012).

*I tried to find why I had written this, and I see that this was a mistake made while writing my dissertation that propagated. I have now removed this, and instead mention the age range of Agassiz rerouting from your 2015 work.*

Pg. 34 (624) most favored hypothesis for NW routing is too generous. As noted earlier, until the Tintah can be directly dated, the strandlines suggest Agassiz did not drain NW until after the YD onset. I have unpublished sediment core data that suggests Agassiz drained east prior to routing NW – as others have argued, there was likely an early Moorhead low – caused by the opening of eastern routes – which was followed by a major Moorhead low when a NW path opened.

*Yes, I was sloppy here. Revised to, "Currently, evidence indidcates that Lake Agassiz first drained east at the onset of the Younger Dryas \citep{Carlson2007, Rayburn2011, CarlsonClark2012} and then drained north at 12.18$\pm$0.48 ka \citep{Breckenridge2015}."*

Pg. 34 (626) – correct the phrase "as the model simulate it"

*"as the model to simulate it"*

Pg 34. (630) missing parenthesis around Anderson et al., 2014

*Fixed.*

Pg 35 (653) where is this gap to the southwest exactly? This could be an explanation for early (pre-8.2k) partial drainage of Lake Ojibway?

*Good question! I have rewritten some awkward phrasing and answered it in the paragraph in the paper, which includes the sentence quoted above on the ~8.6 ka partial drainage of Lake Agassiz—Ojibway (I put the names together to remind myself and others that they were conjoined at this time; let me know if this is not correct). The whole paragraph is below, but the important part towards your question here is un-italicized.*
*"The early Holocene in northern North America was characterized by retreat of the Laurentide Ice Sheet that led to expansion of proglacial lakes, setting the stage for the short-lived and rapid 8.2 ka cooling event \citep{Breckenridge2012}. G12 produced a saddle collapse that led to this flood \citep{Gregoire2012}, though with a somewhat different spatial pattern than what has been mapped \citep{Dyke2003, Dyke2004}. None of the models perform particularly well here. ICE-3G has altogether too much ice retreat. ICE-5G, ICE-6G, and ANU have a drainage divide that splits the east--west Saint Lawrence drainage basin, which at this time extended west to the Canadian Rockies*

*\citep{Teller2004, Breckenridge2012, Stroup2013}*. G12 has this continuous drainage basin, but instead of flowing into the Saint Lawrence River, water in its drainage reconstruction flows to the Arctic Ocean via a gap at the Manitoba--Saskatchewan border along the southwest corner of the retreating ice-sheet (Fig. \ref{fig:mapsEH}). While this could be a route for the $\sim$8.6 ka flood associated with partial drainage of Lake Agassiz--Ojibway \citep{Breckenridge2015}, it is not the long-term drainage route for these lakes through the Kinoj{\'e}vis spillway system and into the Ottawa River \citep{Vincent1977, Veillette1994}. *It is notable that ICE-5G, ICE-6G, and G12 have a tiny early Holocene Mackenzie River basin, and the ANU model also produces a small Mackenzie drainge basin at 8.75 and 8.45 ka. In these models, rapid retreat of the western margin of the Laurnetide Ice Sheet exposed an eastward-tilting isostatically-depressed land surface in the eastern portion of the modern Mackenzie River drainage basin, which drained to the Mackenzie River only after significant isostatic rebound. New cores from the Beaufort Sea record increased $\delta^{18}$O between 11 and 9 ka (Keigwin et al., manuscript in prep.) that cannot be reconciled with changing climate, but would be consistent with a decrease in drainage basin area. Each model output presented here (see supplementary videos) includes this reduced drainage basin area at different times. Therefore, the temporary reduction in Mackenzie River drainage area is sensitive to ice-sheet geometry and/or solid-Earth rheology, making it a useful constraint on ice-sheet reconstructions. G12 fit this timing the best of the models studied here by having a reduced Holocene drainage basin area from 11.0 to 8.2 ka, though it also exhibited a limited basin area at certain times before 11.5 ka."*

*To make this a bit clearer, here is a gorgeous annotated screenshot:*

[Figure]

*Thinking about this in light of your earlier question about the drainage of Lake Ojibway/Agassiz ~8.6 ka, this model would agree with your intuition.*

Figures

I might keep the scale of the axes consistent between the different model runs.

*Good point! I had meant to do this, but had not copied over the lines of code that do this to the proper section of the script. Done now.*

On the paleogeographic maps, only the "from data" figures include political boundaries – I find these useful when looking at the details between each map.

*These are now on all maps.*

It might be useful to see total discharge to the oceans through time compared for all the models.

*Done – the new figure and its caption are below. I have also included the global estimate of Lambeck et al. (2014) for reference.*

[Figure]

Ice volume histories over the North American computational region for each of the examined ice-sheet reconstructions (colors), along with the full global ice-volume estimate of \citet{Lambeck2014} (gray) shifted for easier comparison to match the same ice volume as ICE-6G in the North American region at the modern time-step. \textbf{a.} Cumulative change in ice volume within the computational domain, which here contains the entire North American ice-sheet complex and part of Greenland. These ice volumes are represented in meters sea-level equivalent (SLE), assuming a constant ocean surface area of 3.62$\times$10$^8$ km$^2$ \citep[after][]{MarshallClarke1999}. \textbf{b.} Total meltwater discharge over the computational domain for each ice-sheet reconstruction. ICE-5G and ICE-6G have a noticeable peak during Meltwater Pulse 1A (MWP-1A), 14.65--14.31 ka \citet{Deschamps2012}. Though the time-scale of their numerical model is shifted, \citet{Gregoire2012} also consider their meltwater peak, due to collapse of the ice saddle between the Cordilleran and Laurentide ice sheets, to represent MWP-1A. The vertical axis is in Sverdrups, where 1 Sv is 10$^6$ m$^3$ s$^{-1}$.

*Responses to Reviewer 2 (Tarasov)*

This is a worthwhile study comparing downslope drainage of GIA−based
reconstructions (and one chronology from a glaciological model)
against geological inferences. It is interesting to see how consistent
or not GIA reconstructions are with inferred deglacial meltwater
drainage.

The inclusion of only one glaciological model that lacks any data
documented calibration and that is not self−consistent with the GIA
model used to infer deglacial drainage in this study is a bit
unfortunate.

*It is – and I had considered asking you for your models as well, but you already computed drainage in
your work. In hindsight, I should have asked you if I could check yours with the method developed
here.*

Technically, the drainage computation is sound. However, as detailed
below, more consideration needs to be given to uncertainties in the
inferred "data−driven" reconstruction. This is especially the case wrt
temporal uncertainties in the inferred high/low drainage. "~15.6 ka"
has no clear interpretation. I've also noted some errors in the
inferred chronologies stemming from out of date references with old
age models. I am not on top of the current litterature and as such
there may be more such errors.

*There were some out-of-date references – thank you for the notes on these. I have checked through
many of the others as well. Largely, the poor age control and lack of quantitative error is simply the
state of the data and available literature.*

There are also some inaccurate statements and some poor referencing as
detailed below.

I am curious why proglacial lakes were ignored. The GRASS solver could
easily have computed these and would have enabled some quantitative data
comparisons as opposed to the purely qualitative comparison currently.

*The GRASS solver cannot compute proglacial lakes; see further notes below. Agreed that the lakes are
a great target to hit, and one that I am keen to see done in a more quantitative comparison in the
future.*

Anway, once comments below are addressed, I would support publication in ESurf.
The topic is appropriate for Esurf, figures and abstract are appropriate,...

** detailed comments

generating up to âM−^H1⁄44 km of high−albedo ice−surface topography (Kutzbach
and Wright, 1985; Ull− man et al., 2014)
**Neither of those references are appropriate for a claim of "~4 km" elevation**
**(and high−albedo is obvious). Cite an appropriate primary source.**

*OK; changed to primary sources and an in-review integrative paper (EGU discussions).*

but are evaluated using limited (Tarasov and Peltier, 2006) to no (all
others) geologic evidence for past drainage patterns.
**Incorrect as worded, Tarasov et al, 2012 used the same geological strandline**
**data from 2006 as constraints.**

*My mistake. Fixed.*

Reconstructions of past ice−sheet thickness have proliferated (e.g.,
Tushingham and Peltier, 1991; Lambeck et al., 2002; Peltier, 2004;
Tarasov and Peltier, 2006; Tarasov et al., 2012; Gregoire et al.,
2012; Argus et al., 2014; Peltier et al., 2015), ... Instead, they
are tested against terminal moraine positions and glacial isostatic
adjustment,
**Incorrect. The models are either tuned or calibrated to these**
**constraints. This is different than "testing".**

*Reworded.*

**"glacial isostatic adjustment" what? That is a physical**
**process. Do you mean "records of"?**

*Yes; changed.*

**Also, Gregoire et al, 2012 was not tested nor tuned against**
**terminal moraine positions nor glacial isostatic adjustment records**
**contrary to what I'm inferring you are currently stating.**

*Gregoire et al. (2012) was not formally tuned to terminal moraine positions, but was built to match the
ice outlines. I have separated this as follows:*
*All but the G12 model \citet{Gregoire2012} were tuned or calibrated terminal moraine positions and
records of glacial isostatic adjustment, with the latter being a response that is spread across hundreds
of to $\sim$1000 kilometers due to the high flexural rigidity of the lithosphere, especially in the
Canadian Shield provence that was covered by the ice-sheet \citep{Kirby2009, Tesauro2012}. G12
\citep{Gregoire2012} was built to roughly match ice-sheet outlines and evolve more freely.*

with the latter being a response that is spread across hundreds of to
~1000 kilometers due to the high flexural rigidity of the lithosphere
**Incorrect scales. Yup, there is some smoothing to order 4**
**Lithospheric thickness (so order 400 km for a start), but I can also**
**get significantly different RSL response between 2 sites that are**
**less than 50 km apart or you can also look at RSL records to also**
**note such resolution sensitivity**

*Correct scales, but not precise enough wording. Updated to:*

*"...with the latter being a response with a characteristic two-dimensional flexural half-wavelength of 500--850 km over the high-flexural-rigidity Canadian Shield provence that was covered by the ice sheet \citep{VeningMeinesz1931, Kirby2009, Tesauro2012}."*

many studies do not include any well defined
picture of drainage basin evolution. This....
**With putting out this partial straw−man, why is there**
**no mention of the Tarasov and Peltier, 2006 that does**
**actively compute the self−consistent drainage chronology,**
**and that documents the change in drainage basins?**

*Tarasov and Peltier (2006) was included in the list here of those that "did it right". I think that the problem is that your figures are colored where meltwater flowed, but do not give full drainage basin boundaries. This removes some of the information that a geologist might be interested in (where is the ice divide, on the ice?) and therefore is a need for a more illustrative work (i.e. this).*

*As an aside, I think that too few geologists have made note of and/or read your work, so hopefully this paper will provide another portal to approach paleodrainage work.*

In general, there exists a lack of recognition of the importance of
Pleistocene drainage rearrangement on river systems.
**You are creating another unsupported strawman, especially since you**
**subsequently cite more references that do recognize the changes**
**compared to those that don't. Yes there has been a problem to date**
**with most (but not all) GCMs using fixed present−day routing, but**
**there is a lot more to paleo science than GCM modelling.**

*There were more references for the "we did it right" side. That's because I didn't want to miss anyone in that crowd. As you are in that crowd, it might surprise you that there is a broad blank spot in glacial-stage drainage for many geologists. (I'm happy as well about the GCM's coming through, though.)*

*My point here is that, if the papers that did it right presented the data in a way that were easier for*

*geologists in related fields to digest, that these continued mistakes wouldn't happen. So once again, the goal of this paper is in part communicating this to a broader community.*

*I have added some more to the "not getting it, or at least not entirely" category such that there are now more papers on the "not getting it" side.*

This ice−sheet surface is chosen to drive the flow−routing
calculations because this is what drives ice flow (cf. Cuffey and
Paterson, 2010),
**"drive" makes no sense. The ice sheet surface is used for flow**
**routing because you are extracting surface drainage. This is**
**independent of the physics of ice flow.**

*Rephrased:This ice-sheet surface is chosen for the flow-routing calculations because this is what generates the driving stress for ice flow \citep[cf.][]{Cuffey2010}.*

*I am extracting surface drainage, yes. But this is not surface melt (i.e. flow). This is ice flow direction and water flow direction, calculated at once. This only gets a bit messy with subglacial water near the ice margins, but at this point, the drainage direction is already determined, and this error is only a factor of (rho_water / rho_ice)*

Greenland Ice Sheet, where the subglacial topography is interpolated
from the etopo1 1−arcminute global topographic data set (Amante and
Eakins, 2009).
**I suggest you obtain a more current DEM for Greenland for any future**
**work where accuracy is critical**

*I agree; I kept the Amante and Eakins map to match my 2013 work and because Greenland just happens to be inside the rectangle I clipped around North America (not actually discussing it).*

H_i, ice−sheet thickness,...therefore are interpolated using an
iterative nearest−neighbor approach to remove stepwise discontinuities
that would otherwise introduce artifacts in the flow−routing
calculations.
**Please make the description clear and precise. How do you decide**
**where the ice margin is when downscaling from order 1 degree**
**resolution to 30 arc−second resolution? This would be a critical**
**issue for getting drainage correct, especially when considering**
**switching between Mississippi, St. Lawrence and Arctic drainage for**
**Lake Agassiz region.**

*Addressed and updated to:*
*$H\_i$, ice-sheet thickness, is provided by pre-generated ice-sheet reconstructions. These are produced at a coarser resolution than the 30-arcsecond flow-routing grid, and therefore are interpolated using*

*an iterative nearest-neighbor approach to remove stepwise discontinuities. This smoothing works successively from 1/3 the native resolution to 30 arcseconds, and is necessary because some coarse ice-sheet reconstructions contain multi-cell interior plateaus and several-hundred-meter cliffs at their termini that would otherwise introduce artifacts in the flow-routing calculations. For a 1-degree grid, this requires 5 iterations and leads to smoothing over a half-degree distance ($\sim$40--50 km) outside of a given cell. While this somewhat smears the ice-sheet boundary (see, e.g., ``ICE-6G'' vs. ``From data'' in Figures \ref{fig:mapsLGM} and \ref{fig:mapsBA}), it does not significantly change its fit to the published ice-margin chronology \citep{Dyke2003, Dyke2004}.*

Ice physics, on the other hand, are sensitive to the thermal evolution
of the ice−sheet, related thermomechanical effects on ice
rheology, and the ice basal conditions
**Thermal evolution is a derivative uncertainty as the physics of heat**
**transfer is well represented in current ice sheet models. The**
**primary uncertainty for paleo glaciological ice sheet modelling is**
**the climate forcing (which drives the thermal evolution). For**
**regions with present−day ice cover, another primary uncertainty is**
**the basal conditions (topography, roughness, sediment cover,**
**geothermal heat flux).**

*Updated to, "Ice physics, on the other hand, are most sensitive to the paleoclimate forcing and basal conditions (e.g., topography, roughness, sedimentary cover, geothermal heat flux) \citep{Tarasov2006, Cuffey2010}."*

I investigate both types of models.
**Hardly. You only examine one glaciological model and one that was**
**subject to few constraints.**

*Clarified to, "Here, I investigate four geophysically-based ice-sheet reconstruction \citep{Tushingham1991, Lambeck2002, Peltier2004, Argus2014, Peltier2015} and one ice-physics based reconstruction \citep{Gregoire2012}, with the latter being allowed to evolve more freely than the tuned model reconstructions of, e.g., \citet{Tarasov2006} and \citet{Tarasov2012}."*

and matches geological evidence for the locations of the ice domes and
ice−sheet outlines
**Misleading: Geological evidence (ie the Dyke 2004 ice margin**
**chronology) for ice−sheet outlines is IMPOSED on ICE6G.**

*Good point! Changed to, "and was built to match geologic evidence..."*

GMSL was ~135 m lower than present
**Given different interpretations of what GMSL means (relative to center of**
**mass of earth, relative to present−day shorelines,..), you need to define**

**exactly what you mean (Note, Lambeck 2014 eustatic sealevel is neither of**
**the two definitions I provided above in the brackets)**

*Lambeck et al. (2014) use ice-volume-equivalent sea-level; I have clarified this point at all places where it appeared. "GMSL" is no longer used because it became unwieldy due to the extra explanation. And, following a closer look at Lambeck's sea-level chronology, I have updated this to 130-135 m, which is the range of LGM values in the supplement provided with the paper*

This gravitationally self−consistent sea level theory and its
numerical implementation are de− scribed by Mitrovica and Milne (2003)
and Kendall et al. (2005)
**Keen to stay out of the politics of the GIA community, but it is**
**scientific convention to also cite the originators of a theory and**
**not just a recent description of it.**

*My aim here was not to ignore the other work on GIA (Farrell and Clark, Peltier, …) but to say that it is this particular theory and its implementation that I use here. To clarify this, I have written, "The gravitationally self-consistent sea level theory applied here and its numerical implementation..."*

truncated at spherical harmonic degree and order 256. This satisfies
the Nyquist flexural wavenumber for the solid Earth models employed,
thereby allowing the solutions to be interpolated to an arbitrarily
high resolution
**Provide a reference for "Nyquist flexural wavenumber" (couldn't find**
**one on google, web of science,..). Most readers will not understand**
**what this means and even more will not know it's value is**
**determined…**

*I didn't realize that I was the first to describe it in this way − I am simply using the Nyquist frequency, except as a wavenumber. I have clarified as follows: "This satisfies the Nyquist flexural wavenumber for the solid Earth models employed, meaning that the wavelength of the spectral representation is always less than half of the flexural wavelength \citep{VeningMeinesz1931} of the lithosphere in the solid Earth model. Satisfaction of this condition..."*

VM5a is largely a simplification of VM2,
**Not really. Better to describe it as a modification of VM2 (yes it**
**has a simplified structure but it also has 10^21 Pa s viscosity**
**layer between the Lithosphere and mantle that is not present in VM2.**

*Good catch − I was only looking deeper. In fact, I don't know what I was thinking when I wrote this section − I completely left out VM1. Now the paragraph is much longer and more complete.*

I initiate the GIA modeling of G12 in equilibrium at the LGM

**Should mention that is a source of error, as there is no evidence to**
**support isostatic equilibrium for North America at LGM**

*True. I added the following, "This is a potential source of error: while the sea-level records of \citet{Lambeck2014} indicate that global ice mass was grew only slightly between 30 and 20 ka, providing enough time to approach isostatic equilibrium, this is only ~2 $e$-folding time-scales based on data from the {\AA}ngerman River and Hudson Bay \citep{Mitrovica2004, Nordman2015}. Therefore, it is very likely that the North American Ice Sheet complex was not yet in isostatic equilibrium at the LGM."*

Our flow−routing calculations neglect geomorphic change for three
major reasons. First, significant changes Lake Agassiz outflow
directions occurred as a precursor to spillway incision
**This approximation is reasonable since you are not modelling lake**
**levels and are therefore not making direct comparisons to**
**strandlines.**

*Sounds good.*

Precipitation and evaporation inputs changed through time
(Fig. 2). These changes were calculated by Liu et al. (2009) and He
(2011) using a continuous LGM (22 ka) to present run of the National
Center for Atmospheric Research (NCAR) Community Climate System Model
version 3 (CCSM3) (Collins et al., 2006).
**Should mention that the TraCE−21K simulation used ICE−5G as a**
**boundary condition, and so will not be self−consistent with other**
**deglacial ice thickness reconstructions you are using.**

*Important − yes. Added, "Therefore, the only fully self-consistent calculations presented here are from the ICE-5G/VM2 reconstruction."*

Flow routing and drainage basins can be calculated only on time steps
when the ice−sheet and GIA models provide surface topography (Section
2.3, below), but the ice−sheet contribution to runoff re− quires
ice−sheet thickness to be differenced, and thus should be most valid
for the times halfway between the ice thickness time−steps.
**There is no basis for the last claim when you have 500 year**
**timesteps (eg ICE5G). On that note, you need to explicitly provide**
**what timesteps were used**

*To the first point, I think this is a simple miscommunication. I have removed the awkward wording that you object to; all I was saying is that a central differencing approximation is better than differencing and placing that difference at some other point between the new time steps.*

*To the second point, I have provided a new paragraph that explains this:*
*"Time steps vary by model. ICE-3G \citep{Tushingham1991} and ANU \citep{Lambeck2002} use irregular time steps separated by a $\sim$1000 years (ICE-3G) and $\sim$500 years (ANU). Time steps for ICE-5G are 1000 years from 32 to 17 ka, and 500 years thereafter \citep{Peltier2004}. All time steps for ICE-6G \citep{Argus2014, Peltier2015} are 500 years. Model outputs from \citet{Gregoire2012} were provided at 10-year intervals, with GIA computed at uniform 100-year time-steps \citep{Wickert2013nature}, which are the same as those used here for the flow routing calculations. This means that the model outputs will generally average over short-term geologic events that are responsible for creating some of the major spillways and deposits \citep[e.g.,][]{Kehew1994, Breckenridge2007, Murton2010}."*

This means that these models implicitly include the lakewater volume
within the ice mass.... as ice must be used in the models to represent
the surface loads of these large lakes in the geologic past....
**Incorrect and no easy choices here. Since most of the proglacial**
**lake sites lack proximal RSL constraints, there is little basis to**
**assume that the GIA models are implicitly taking pro−glacial lake**
**loads into account (except wrt global ice volume required to match**
**far−field constraints). Most proglacial lake regions are more**
**constrained in current GIA models by present−day vertical**
**velocities, but this has much poorer time resolution. Furthermore,**
**models such ICE−5G have their ice load spatial extent set to (though**
**with no clear documentation whether any discrepancies are allowed)**
**the Dyke et al 2004 ice margin chronology. This means they have no**
**load where there are proglacial lakes.**
**It would be best to repeat at least one model calculation with**
**inclusion of pro−glacial lake loads to quantify the uncertainty.**

*I agree in spirit, but don't think that this is possible with the ice-sheet reconstructions as they currently exist. As you note, the lakes must take global ice-volume equivalent sea level. Therefore, that lake water volume* must *be in the ice sheets. To include the lakes would therefore be to ignore mass balance. As my goal here is to evaluate existing models, I will not add the lakes for this work. However, I am 100% on board with the spirit in which this comment is written, and in fact, have plans to include proglacial lakes in future analyses. So stay tuned...*

models to match the observed ice−sheet geometry,
**"observed"? Sure wish we had observations of paleo ice−sheet geometry…**

*Wishful thinking! Changed to, "match the observed record of ice-marginal geometry".*

While it is essential to ignore local depressions in the DEM
**Why? "Local depressions" = lakes = more data for comparision.**

*A perfect question for an age-old problem! I have added the following paragraph for clarification and*

*changed the one after it as follows:*

*"The ability of the \citet{Metz2011} algorithm to route water across local depressions also prevents both natural and artificial (human-built or DEM artifact) highs and lows in the landscape from stopping the flow routing. This is an advantage because flow-routing calculations without this feature fail unless they are performed on a hydrologically-corrected DEM (i.e., one in which the elevation is altered to produce the known drainage basins). Using a hydrologically-corrected DEM is a potential source of error in reconstructions of the past because its actual terrain has been artificially changed \citep[e.g.,][]{Grimaldi2007} in a way that may produce unexpected results when combined with surface deformation and ice-sheet thickness.*

*In spite of these advantages, the \citet{Metz2011} method causes internally drained..."*

This is an improvement over âM−^@M−^\hosingâM−^@M−^] experiments (e.g., Condron
and Winsor, 2012)
**This citation does not make sense for this context. Better to cite**
**PMIP III. Unlike PMIP III hosing that distributed meltwater fluxes**
**across a band of of the North Atlantic, Condron and Winsor for the**
**first time resolved what would happen (all be it for only a few**
**years) to discharge from major river outlets. If you are citing**
**Condron and Winsor as an example of why PMIP style hosing has no**
**geophysical basis, then you need to rewrite the sentence to make**
**this clear**

*I'm not sure where I miscommunicated, but I have reworded the paragraph. To further elaborate. Condron and Windsor (2012) changed a patch of the polar ocean and watched what would happen – an excellent intuition-builder! What I am writing about is computing all discharges to the ocean and then sending these to the ocean – a different task.*

*The updated paragraph reads,*

*"These methods also allow us to produce grids of meltwater discharge to the coast that are fully-distributed in space and time. These have been converted to GCM-input-appropriate spatial resolutions and used in preliminary work to connect ice melt and global ocean circulation and climate change \citep{Ivanovic2014AGU}. This is an improvement over ``hosing'' experiments \citep[e.g.,][] {Meissner2006, Liu2012, Condron2012}, in which meltwater additions are prescribed to a patch of ocean in a way that need not be connected to patterns of continental runoff. These ``hosing'' experiments provide very useful intuition for the climate impacts of a major meltwater input event. Fully-distributed GCM meltwater inputs that are tied to paleodrainage histories, when coupled with models that can appropriately handle point sources of fresh water, create a more realistic ice-age ocean that is better-conditioned to respond to changes in meltwater inputs and can be used to connect models of ice-sheet history with their likely impacts on global climate. Code to produce these GCM inputs is also provided in the Github repository (\url{https://github.com/umn-earth-surface/ice-age-rivers})."*

The fully−distributed GCM meltwater inputs create a more realistic
ice−age ocean
**A bit simplistic. It is unclear whether fully−distributed GCM**
**meltwater inputs can create a more realistic ice−age ocean in**
**current paleoclimate modelling AOGCMs since these models lack the**
**resolution to accurately resolve meltwater flux transports and**
**turbulent mixing**

*Qualified to, "when coupled with models that can appropriately handle point sources of fresh water, create a more realistic ice-age ocean"*

I developed a set of data−driven drainage basin boundaries for each of
the study basins ... The second is the ice−sheet margin chronology of
Dyke et al....
**There are many significant uncertainties in this approach, that need**
**some tabulation For instance, the Dyke et al chronology has**
**significant temporal uncertainty (cf Tarasov et al, 2012), that**
**could significantly affect deglacial drainage chronologies.**

*Added, "\citet{Tarasov2012} and \citet{Gowan2013} note that uncertainties exist in this deglaciation chronology"*

The second is the ice−sheet margin chronology of Dyke et al. (2003);
Dyke (2004), which, when lacking independent information, I use as an
approximate set of contours of ice−sheet thickness
**Does not make sense. How do you get from an ice margin chronology to**
**ice sheet thickness contours?**

*The last part of the ice-sheet to melt is probably close to the zone of maximum thickness of ice. Of course, the highest portion of the ice sheet can migrate, but it is not a bad first-order approximation, and in any case, is corroborated by the records of ice-streaming by Margold et al.*

As such, disagreements between these data−derived basins and those
derived from models, especially where the data−driven drainage basins
are tightly−constrained
**For such interpretation, you then need to provide uncertainty**
**estimates for the data driven approach**

*I have changed it and the whole paragraph. Uncertainties in the data-driven approach are not quantitatively constrained, but provide a rules-based starting point for thinking about how to look at the data and form simple conclusions. Drainage basin areas are robust based on the terrestrial record, but discharges are less so. I think that the recent work by Gowan et al. (2015, 2016) is the way forward in this regard.*

*The paragraph now reads:*

*"The counterpart to these computed histories are those that are driven by more direct interpretations of the data \citep[e.g.,][]{Teller1990P, Licciardi1999, Carlson2007, Carlson2009}. I write, ``counterpart'' as opposed to ``ground-truth'' because these are also to some extent interpretations, based on limited data. Their key utility is that they offer a much simpler path from the raw data to the interpreted drainage basin extents and periods of high discharge, and one that can be more closely tied to geological fact. No age uncertainty is provided for these data-driven reconstructions: The maps display multiple events that coincide, assembled from disparate data sources, of which not all have well-quantified error. The periods of high and low discharge are marked with sharp lines even where chronological constraints are sparse, and should be viewed as a current geological best estimate. Nevertheless, the large-scale drainage basins noted on Figures \ref{fig:mapsLGM}--\ref{fig:mapsEH}, each of which displays an important time in deglacial history, display the current state of knowledge for that point in the geologic time scale, regardless of chronological error. Therefore, large-scale disagreements between the data-constrained drainage basin outlines and the model results at these key points in the geological time scale generally indicate a need to improve the model-based reconstructions. Mismatches between data-driven and model-driven discharge histories are less straightforward to attribute to model error, as they involves both drainage basin area and rates of ice-sheet melt, with the latter being less easily quantified by geological data. These potential mismatches are discussed below."*

but by 19.9 ka,
**I'm assuming calendar before present? Needs to be clarified**

*Yes, and good catch. All ages calendar, especially because not all chronological control is 14C-based. I have now included, "All radiocarbon ages were recalibrated to calendar years BP using IntCal13 \citep{Reimer2013}." early in the paper.*

**Figures 6−9: "from data" panel: what is the age uncertainty?**

*The "from data" panel includes all events occurring during this time, each of which has its own specific age uncertainty, as does the ice margin. These figures are a reconstruction in a very geological sense: no quantified errors, lots of correlation. Why? Lots of unreported error, lots of different data types, data sources for which error is difficult to quantify (how long did the ice stay at these hills based on a handful of dates that may each have their own systematic errors), …*

*Therefore, I have made the changes noted two comments above.*

**Figures 10−15 the high/low/none inferred discharge shading needs**
**a visually representation of temporal uncertainties in the transitions**
**eg, with hatching or someother visual texturising**

*As before, this would be possible in a perfect world, but with the limited age control, we often just know that "X happened between A and B", and more likely between [subset of ages]. I think that it is better to leave these as they are. However, I do agree that your point is valid, and we do know that "X*

*happened during the Younger Dryas". So, as noted above, I have changed the respective paragraph to account for this.*

18.2âM−^@M−^S17.7 ka (Rashid et al., 2003)
**You need update your citations, this is not a consensus estimate**

*I can see two reasons for this comment, both of which lead to improvements in the text:*
*1. That "Heinrich Event" is canonically used for the timing of IRD release, typically shorter than the climate impacts (which also use the "Heinrich Event" terminology).*
*2. That the reference is, in fact, old!*
*(1) makes it hard to find a "consensus" estimate (it bounces around based on field), but led me to check that the IRD definition of Heinrich Events is in the text (it is).*
*(2) Pushed me to find a newer reference (Gil et al., 2015). While Carlson comments on the Younger Dryas part, Gil et al. (2015) show that their two-phased IRD pulse from H1 on the Laurentian Fan is consistent with near-European records. As such, I have updated the dates of H1 to:*
*H1A: 16.8-16.3 ka*
*H1B: 15.9-15.7 ka*
*(these are calibrated with IntCal13)*

Heinrich Event 0, corresponding to the Younger Dryas,
was 12.8âM−^@M−^S12.3 ka (Clark et al., 2001)
**Again, this is an out of date termination estimate.**
**Current chronologies end it around 11.6 ka**
**Your age extraction from old references also ignores recent**
**refinements in ice core age chronologies and C14 calibration**

*To the first two lines: this is great to be aware of. I was never satisfied with the Hudson Bay chronology from the Younger Dryas through the early Holocene, and now see that numerous papers on it came out in 2015, after I had ended my literature review.*

*The points about 14C calibration are incorrect: I am using IntCal13 and recalibrating all of the ages from old papers accordingly. To make this clear, I have added the line, "All radiocarbon ages were recalibrated to calendar years BP using IntCal13 \citep{Reimer2013}."*

started ~15.6 ka
**what does "~" mean? +/− 0.1 ka, +/− 1 ka or ?**

*"~" means that age control is weak, old, and from a data-poor region. The age is 14C without reported error, because it is an amalgamation of data sources. I have added a comment to this effect.*

At 13.1 ka, ice retreat rerouted Glacial Lake Peace
**What dating scheme is going to give 100 year accuracy at 13.1 ka???**

*Revised to ~13.1. I know it is unsatisfactory, but no good error estimates exist for these ages.*

*Though to answer the question you pose in a more general sense: varved sediments with absolute age control in nearby layers. For example, the Laacher See Tephra in Europe.*

I then generated histograms for the high−flow and low−flow segments of
the model results, and performed KolmogorovâM−^@M−^SSmirnov tests between the
pairs of distributions for each ice model and river system to test how
different they are from one another
**again how was temporal uncertainty taken into account?**

*Temporal uncertainty was not taken into account, but would provide a data–model mismatch if it were on either side. Yes, the geological constraints are also imperfect. This is not a formal data—model integration exercise, but rather is intended to highlight where the two are more or less in agreement. To that extent, I think that it successfully separated the more successful and less successful models.*

The ice−physics−based G12 performed the best in generat− ing
discharges during high−flow periods that were much higher than those
during low−flow periods.
**Again need to make clear what timesteps were used. Was the above due**
**to G12 being provided at higher temporal resolution or not?**

*Included the time steps as noted above. It is not clear whether finer time steps help or hurt: these values are means across larger swaths of the post-glacial time. One indication that finer time steps can hurt is the lower match with data for the time-step sparse ICE-3G and the time-step rich G12 for the non-glacial rivers, which should be the same in both.*

*I have added the following sentence to address this,*
*"For both of these metrics, the inconsistent time steps between ice-sheet reconstructions can affect the fits, as can be seen especially for the Rio Grande and Columbia River, which should be the same for all runs as the same climate reconstruction was employed in all cases. Therefore, these catchments were removed from the above analysis, the broad strokes of which provide a useful template for comparison."*

*(1) evaporative loss that is generally >1.5 times todayâM−^@M−^Ys observations, in spite of cooler temperatures and a longer lake−ice season*
**you ignore the likelihood of much higher mean wind speeds near an**
**ice margin and their resultant impact on evaporative loss. This**
**needs to be mentioned as a potentially offsetting factor to your**
**one−sided critique.**

*Katabatic winds are certain, but there would be no topographic funneling of such winds. I have added text that the winds would be present. The critique was one-sided because Lowell et al. (2013) showed*

*that Lake Agassiz could be a closed basin if they picked parameters outside reasonable limits, and concluded that their modeling supported their hypothesis that lake-level lowered due to evaporation. I have trouble understanding this sort of conclusion.   The two parameters that I mention are not the only two that are problematic: they also had to choose a low PDD factor, low runoff, low precipitation, large lake area (contributing to evaporation)… more or less, every knob that could be turned to make Agassiz a closed basin, was turned. So plausible? Yes. Likely? I think not.*

**Responses to Carlson (comments received via email):**

Hi Andy, so saw your paper in revision on line at the EGU journal. Cool stuff. So, had a couple of points/references you should include to be complete. I hope you can add these! Also, there are a few errors in the references that I highlight below. Just FYI, we have now 10Be ages from the eastern outlet that show it opened at the start of the YD and so Agassiz went east. Then the Breckenridge calculations support our old idea that Agassiz went north late in the YD.
Cheers
Anders

Line 22 - add Ullman et al. 2014  Climate of the Past reference
... it's already there.

   Line 41 - add Carlson & Clark 2012 Reviews of Geophysics reference
Right! Your review is a very useful article. Added.

   Line 44 - add reference to Licciardi et al. (1999, AGU monograph); Clark et al. (2001, Science); Carlson et al. (2007, PNAS); Carlson (2009, QSR); Carlson et al. (2009, GRL); Obbink et al. (2010, Geology); Carlson & Clark (2012 RoG) - and each of these papers did calculate discharge - and add Hoffman et al. (2012, GRL) who did not calculate discharge…
I've already modified this to include some of these references (in the version you saw, I was focusing more on site-specific references). Somehow I had completely forgotten about the Obbink paper!

   Line 54 - this isn't full accurate, Ullman et al. (2014, CoP) and (2015, Nat. Geosci) did alter drainage basins.
Re: "no well-defined picture", Ullman et al. do show ice divides, but not the drainage basin boundaries: some of these divides bridge sectors that go into the same drainage basin eventually, and their figure does not connect their work to the continental drainage patterns. I have therefore added the Ullman et al. references to the previous line -- on studies that acknowledge the change without making the change prominent. I've changed the wording to "clear picture of..." from well-defined, because Ullman et al. is well-defined, it's just not placed in line with the rest of the continent.

   Line 60 - see above comment; also Liu et al. (2009, Science; 2012, PNAS) and He et al.

(2013, Nature) did not change drainage patterns

I've added a new line towards the end that states,

"Paleoclimate general circulation model reconstructions of past climate \citep[e.g.,]{Liu2009, Liu2012, He2013} maintain static drainge basins in spite of climate sensitivity on patterns of meltwater routing, and this has just begun to be addressed \citep{Ivanovic2015}."

Line 90 - ref Licciardi et al. 1999 AGU monograph

Right. I think in these cases that I just felt I was citing it too much / all over, but it should be added.

Line 250 - ref Dutton et al. (2015, Science)

I've added the Dutton ref., but their 6-9 m limit is not based on Hay et al. (2014), whose study is as far as I know, the current best estimate, and Dutton et al. (2015) is a review. I've indicated which estimate I use in the paper.

Line 319 - ref Clark et al. 2001 Science

Done

Line 395 - ref Meissner & Clark (2006, GRL) and also Liu et al. (2012, PNAS) who did not do hosing, but rather point source additions of freshwater

I had the Meissner & Clark ref from before, and wish I would have thought of it earlier -- their point that the short-lived floods don't matter too much to the oceans/climate is one that I have to consistently make/debate at Quaternary meetings. I'll keep it close at hand.

Line 460 - ref Carlson (2009, QSR) and Obbink et al. (2010, Geology)

The references I had there were for original data, so I added your GRL paper and the Obbink paper, but not these 2009 QSR one. That one could have gone into the "paleohydrograph" portion, but is a single point, and so not a hydrograph.

Line 462 - Carlson et al. (2007, PNAS) and (2009, GRL) were able to determine basin area based on geochemical provenance

Added, though it doesn't give full marginal geometry -- but it is a very good hint at it if you know something more about the landscape and past ice sheets

Line 465 - Carlson et al. (2007, PNAS); Carlson (2009, QSR); Carlson et al. (2009, GRL); Obbink et al. (2010, Geology) did calculate discharge directly

Yes, and cited. Though most studies still aren't as quantitative, so I have separate spots for both.

**Line 515 - ref Obbink et al. (2010, Geology) which documented this switch.**

I had the ~14.6 ka from the terrestrial record -- good to see it in the marine, too. Really unfortunate that I hadn't read the Obbink paper! I added Hansel and Mickelson as another reference to be more thorough.

**Line 518 - ref Obbink et al. (2010, Geology)**

Done

**Line 522 - Breckenridge actually calculated northward routing at 12.2 ka - after the start of the YD - correct**

Corrected to 12.18$\pm$0.48 ka, with new text. Leydet's thesis cited here too. This paragraph now reads:

Meltwater routing down the Mississippi lasted until $\sim$12.9 ka \citep{Williams2012, Wickert2013nature}, when its last major ice-sheet contributor, the Lake Agassiz drainage basin, was most likely rerouted first east to the newly-ice-free Saint Lawrence \citep{Broecker1989, Carlson2007, CarlsonClark2012, Levac2015, Leydet2016}, and then was routed towards the Mackenzie River at 12.18$\pm$0.48 ka, based on strandline data \citep{Breckenridge2015} that are consistent with the 13.0$\pm$0.2 to 11.7$\pm$0.1 bounds placed on the northward rerouting by bouldery deposits on the Mackenzie River delta, presumably placed there during a large flood associated with drainage rearrangement. Starting just prior to Lake Agassiz rerouting, at $\sim$13.2 ka, a significant amount of meltwater was routed to the Gulf of Saint Lawrence \citep{Carlson2007, Rayburn2011}, which continued to receive significant ice-sheet meltwater inputs via proglacial lakes until sometime between $\sim$8.6 ka, the potential subglacial flood from the combined Lake Agassiz-Ojibway into Hudson Bay \citep{Breckenridge2012}, and 8.2 ka, when the ice saddle in Hudson Bay collapsed, causing the 8.2 ka event flood \citep{Barber1999, Gregoire2012, Roy2011, Breckenridge2012, Stroup2013}. While this was the last meltwater input to the Saint Lawrence River, meltwater from the Qu\'{e}bec-- Labrador dome flowed down the Manicouagan River and into the Gulf of Saint Lawrence until $\sim$7.8 ka \citep{Occhietti2004}.

**Line 536 - reference Andrews & Tedesco (1992, Geology)**

Forgot the original here! Done. Although this has been updated to "formerly thought to be" after recent work that updates its chronology:

Heinrich Event 0, formerly thought to correspond to the Younger Dryas (12.8--12.3 ka \citep{Andrews1992, Clark2001}), has been recently shown to correlate with early-Holocene warming and melt 11.5--11.3 ka \citep{Pearce2015}.

> Line 541 - reference Carlson et al. (2008, Nature Geosci)

Much better than the more generic Dyke refs here. Switched out.

> Line 561 - reference Lopez & Mix (2009, Geology) that reconstruct salinity/discharge of the Columbia River.

Haven't seen this reference before -- it complements the others and the story stays pretty much the same. Thanks for the heads-up.

> Line 624 - see Carlson et al. (2009, QSR) on the ridiculousness of this no-outlet hypothesis.

OK.

> Line 625 - this is wrong - northward routing, like Carlson et al. (2007, PNAS) argued, is late in the Younger Dryas, which is also what Breckenridge (2015) concludes. Actually, the data and model driven approaches agree quite well. Correct.

Already corrected.

> Line 629 - see Carlson et al. (2009, QSR), who used a regional climate/lake model - should reference

Referenced now -- and good to see that you and I came to the same conclusion -- though it's not clear whether or not you included ET in the rest of the basin (I'm guessing you did for the RCM inputs).

> Line 642 - see Carlson & Clark (2012, RoG) page 46 paragraph 173, who show the Murton et al. (2010) record to be misinterpreted and their conclusions fly in the face of the most basic laws of stratigraphy. Their conclusions are fundamentally wrong.

I have noted that their dates are consistent with Breckenridge's.

> Line 666 - ref Carlson et al. (2007, PNAS); Carlson (2009, QSR); Carlson et al. (2009, GRL); Obbink et al. (2010, Geology)

I'm trying to keep these lists of references from being a litany -- so had just three representative pubs here with "e.g.,"

> Line 679 - add ref to Carlson et al. (2007, PNAS), Carlson (2009, QSR) and Obbink et al. (2010, Geology)

You and collaborators did the oxygen isotope part, but not the drainage routing component, so can't add these here. (e.g., in your 2009 QSR paper, you used the Licciardi et al. (1999) value and discounted transpiration.) I find that key pieces to reconstructing past discharge include (1) having a good meteoric and ice-melt input from a time-evolving drainage basin, and (2) taking these calculations all the way to latest Holocene records to ensure that I am consistent with a scenario in which I can substitute modern

river discharge and d18O.

Line 681 - add refs Carlson et al. (2007, PNAS); Carlson (2009, QSR); Carlson et al. (2009, GRL); Obbink et al. (2010, Geology); Hoffman et al. (2012, GRL)

Added those papers that supplied data: C2007, C2009-GRL, O2010, H2012. I also added Lopes and Mix here.

Carlson also suggested a few additional reference inclusions, some of which I had made already; all added references are appropriate, and many had been omitted earlier to prevent the lists of references from becoming too unwieldy.

---

## Author Response (AR2)

**Response**

These revisions are the result of myself requesting to have my paper returned for some corrections to be made. These corrections are mentioned in the "general notes" section below, and affect in a small way most figures of the drainage basin and discharge histories, as well as some of the text. There has been no major change. As there is no point-by-point response to be made, I have not attached a version of the manuscript with differences here, though I do note responses to the helpful comments made by the copyeditors.

– ADW, 6 October 2016

**General notes on paper:**

When I requested to have more time on this paper, it is because the "river mouth regions" were not optimized. They are created by hand, and are now the best I can make them. This has resulted in slightly improved river routing. During this time, I have also gone through the papper twice, cleaned up the English, tried to tighten or add detail to sections where needed, and added new references where I found them useful. All of these changes have been "minor" in that they have not altered the substance of the work -- but I hope that they have made it a bit easier to read and digest and connect to the broader literature.

**Responses to comments from proofreaders**

CE1: "Flexurally" is a form of "flexural" that is commonly used in scientific articles, so I think that the sentence is understandable and should stay as it was.

CE2: Changed to, "Any series of operations within GRASS GIS may be scripted, and the Python..."

CE3: This should be "GitHub" on all occurrances; thank you for catching this. I have fixed it.

CE4: "seems unlikely" changed to "is glaciologically implausible"

CE5: deleted "it"

CE6: "To threshold" is a common verb form, e.g., in image analysis, at least in USA... but I see the point to make this more readable. We can replace this and the last sentence of the paragraph by, "...resulution. Any negative evapotranspiration rates generated by the interpolation were set to 0."

CE7: Here, "threshold" is the best term, and at least in my US English is an acceptable verb.

CE8: The matching ) goes after 11-16. Thank you for catching this!

TS1 & TS5: Uploaded to UMN DRUM; URL provided in paper and DOI will be forthcoming (will have to add it during copyediting)

TS2: Double-checked.

TS3 & TS4: This is an in prep. paper for which I have access to the data. This is therefore not in the reference list. If this can be indicated as "in prep", that will work. If not, the reference can be changed to the following conference abstract:

Keigwin, L. D., and N. W. Driscoll (2014), Deglacial floods in the Beaufort Sea, in AGU Fall Meeting Abstracts, B7.

--> Update: they submitted the article, so I have cited it as "submitted". I have still added the abstract, for completeness.

TS6-: Notes only as needed. Otherwise fixed as requested.

TS9: No pages; data set referenced here

TS11: Can't add much more here, but added the section in the inline reference ("section 8.2.1")

TS12: It's software -- nothing more to add here.

TS28: No more information to provide on this book (though I do now provide middle initials!)

34: Is a book; all pages.